# HRCHY-CytoCommunity identifies hierarchical tissue organization in cell-type spatial maps

Runzhi Xie[1,4], Zekun Wang[1,4], Jianrui Liu[1], Han Xu [1], Yafei Xu[1], Jiadong Lin [2,3] ✉, Yuxuan Hu [1] ✉ & Lin Gao [1] ✉

Tissues are organized through the assembly of diverse cell types into multi-cellular structures that exhibit hierarchical spatial organization. We present HRCHY-CytoCommunity, a graph neural network framework for identifying multi-level tissue structures directly from cell-type annotated spatial maps. It integrates differentiable graph pooling, adaptive edge pruning, and consistency and balance regularization in an end-to-end model, simultaneously inferring robust structures across multiple scales while preserving complete cellular coverage and fully nested relationships. The framework also supports cross-sample hierarchy alignment via cell-type enrichment-based clustering. Benchmarking on diverse spatial omics datasets, HRCHY-CytoCommunity outperforms existing hierarchical and non-hierarchical methods in identifying both coarse-grained tissue compartments and fine-grained cellular neighborhoods. Applied to a breast cancer cohort with clinical outcomes, the framework enables hierarchical prognostic stratification of patients and reveals survival-associated spatial patterns. HRCHY-CytoCommunity represents a general and scalable tool for deciphering tissue organization from single cells to multicellular modules, and ultimately to intact tissues and organs.

The rapid advancement of spatial omics technologies has significantly advanced our understanding of tissue spatial organization[1]. High-throughput, sequencing-based spatial transcriptomics technologies, such as Visium[2,3], Stereo-seq[4], and Slide-seq[5,6], provide transcriptome-wide gene expression profiling while preserving spatial context. Imaging-based spatial transcriptomics and proteomics technologies like MERFISH[7], CODEX[8], IMC[9], and MIBI-TOF[10], enable highly multi-plexed, single-cell-resolution measurement of targeted RNA or protein markers. To systematically decipher functional tissue organization based on these spatial omics data, researchers have introduced the concepts of cellular neighborhoods and spatial domains, which serve as multicellular modules where distinct cell types interact and colla-borate to support tissue functions[8,11,12]. This has spurred the

development of numerous computational methods for identifying such structural modules[13–24].

Many tissues exhibit inherent hierarchical organization, wherein large structures contain smaller, functionally specialized sub-structures. This pattern is evident in the layered architecture of the brain, the zonated morphology of lymphoid organs such as the spleen, and the compartmentalized ecosystem of tumors[25–27]. For example, some tumor tissues can be coarsely partitioned into neoplastic, immune, and stromal compartments[28,29], which can be further sub-divided into finer neighborhoods enriched with specific cell types, such as B cells and T cells, granulocytes, or fibroblasts[30]. These hier-archical patterns carry biological and clinical significance. As an example for illustrating the role of coarse-grained tissue structures, in

[1]School of Computer Science and Technology, Xidian University, Xi'an, Shaanxi, China. [2]School of Automation Science and Engineering, Faculty of Electronic and Information Engineering, Xi'an Jiaotong University, Xi'an, Shaanxi, China. [3]MOE Key Lab for Intelligent Networks & Networks Security, Faculty of Electronic and Information Engineering, Xi'an Jiaotong University, Xi'an, Shaanxi, China. [4]These authors contributed equally: Runzhi Xie, Zekun Wang.
✉e-mail: jdlin@xjtu.edu.cn; huyuxuan@xidian.edu.cn; lgao@mail.xidian.edu.cn

triple-negative breast cancer (TNBC), spatial segregation between neoplastic and immune compartments is associated with favorable outcomes[10]. As a representative fine-grained multicellular module, the presence of B and T cell-enriched tertiary lymphoid structure serves as a positive prognostic marker across multiple cancer types[31–33]. Uncovering such a multi-level organization is therefore essential for understanding how tissues assemble from individual cells into functional modules.

Despite its biological importance, few computational methods are designed specifically to identify hierarchical tissue structures. The current state-of-the-art method, NeST[34], identifies nested hierarchies in spatial transcriptomics data by detecting co-expression hotspots, which are defined as groups of spatially colocalized cells that co-express a set of genes. However, NeST relies on gene expression features, limiting its performance on data with sparse molecular measurements, such as spatial proteomics datasets[8,9]. Moreover, the structures identified by NeST often fail to cover all cells in the sample and may lack a clearly nested relationship across scales. For instance, cells within the same fine-grained structure can be assigned to different coarse-grained structures, complicating biological interpretation. Additionally, NeST treats spatially disconnected hotspots as distinct tissue structures, potentially impeding the discovery of structures with spatially discontinuous distribution.

Non-hierarchical methods[15,20–23], though not originally designed for hierarchical analysis, can be repurposed to identify multi-scale tissue structures through *post hoc* adjustment of clustering resolution. However, this strategy requires each hierarchical level to be trained and identified independently, often resulting in a lack of nested relationships across scales. Moreover, the absence of joint optimization or constraints between coarse- and fine-grained representations may compromise biological interpretability and reduce the reliability of the resulting hierarchical structures.

To overcome these limitations, we developed HRCHY-CytoCommunity, a hierarchical extension of the CytoCommunity framework[16], that identifies multi-scale tissue organization in cell-type spatial maps. Unlike conventional multi-step approaches, HRCHY-CytoCommunity employs a unified end-to-end learning framework that simultaneously infers tissue structures at multiple levels while preserving hierarchical consistency through joint optimization. By leveraging differentiable graph pooling, adaptive edge pruning, and consistency and balance regularization during training, the model identifies robust and fully nested tissue structures that encompass all cells and reflect biologically meaningful spatial hierarchies. This joint learning strategy enables bidirectional information flow across scales, promoting the identification of structurally consistent and functionally relevant tissue architectures. The model also supports cross-sample integration through cell-type enrichment-based clustering, aligning hierarchical structures across samples for comparative analysis.

In this study, we focused on a two-level hierarchy, defining coarse-grained structures as tissue compartments (TCs) and fine-grained structures as cellular neighborhoods (CNs). We demonstrated the performance of HRCHY-CytoCommunity across a wide range of simulated and real spatial omics datasets spanning diverse tissues, technologies, and modalities. Benchmarking results showed that our method outperformed existing hierarchical and non-hierarchical approaches in reconstructing biologically meaningful spatial hierarchies. Finally, using a clinical breast cancer cohort, we illustrated how these hierarchical structures can serve as multi-scale prognostic indicators, highlighting the potential of HRCHY-CytoCommunity to support translational research.

## Results

### Overview of HRCHY-CytoCommunity

HRCHY-CytoCommunity is an unsupervised graph neural network framework designed to identify hierarchical multicellular structures from single-cell spatial maps that integrate both cell type and spatial location information. The framework comprises a base module for soft hierarchical tissue structure assignment and a consistency and balance regularization module to ensure robust identification. Figure 1 illustrates the identification process of tissue structures at two hierarchical levels.

In the base module (Fig. 1a; Methods), HRCHY-CytoCommunity begins by constructing a K-nearest-neighbor (KNN)-based cell-cell proximity graph, where nodes represent cells and edges connect spatially adjacent cells. Each node is equipped with an attribute vector encoding cell type information. The model then applies a graph convolution layer with ReLU activation to generate node embeddings, which are subsequently transformed through a fully-connected layer with Softmax activation into soft assignment vectors for fine-grained tissue structures (i.e., CNs). These vectors represent the probability of each cell belonging to a specific CN.

HRCHY-CytoCommunity applies a differentiable graph pooling layer to coarsen the original graph into a new completed graph, where each pooled node corresponds to a fine-grained CN. The edge weights in this coarsened graph reflect inter-CN connectivity strength. To prevent the over-smoothing effect from message passing during graph convolution, an adaptive edge-pruning step is applied on the coarsened graph before performing a second graph convolution and a fully-connected layer to produce soft assignment vectors for coarse-grained tissue structures (i.e., TCs). These vectors represent the probability of each CN belonging to a specific TC. The learning process is guided by two graph minimum cut (MinCut)-based loss functions[35], separately optimizing fine-grained CN and coarse-grained TC assignments.

The second module (Fig. 1b; Methods) introduces both consistency and balance regularization to enhance the accuracy and robustness of hierarchical tissue structure identification. During training, multiple perturbed versions of the cell-cell proximity graph are generated by randomly dropping node features (DropNode). These perturbed graphs are fed into the base module to produce soft fine-grained CN and coarse-grained TC assignment matrices. Consistency regularization is applied to enforce agreement among the resulting assignments from different perturbed graphs, while balance regularization based on entropy prevents cluster collapse. The total loss function combines the base MinCut loss with the two regularization terms. Final robust hierarchical structures are obtained by performing hard assignment on the stable soft assignment matrices.

Since HRCHY-CytoCommunity operates as an unsupervised learning framework optimized for single-sample analysis, an optional cross-sample tissue structure alignment module is included (Supplementary Fig. 1; Methods). This module computes cell-type enrichment scores for each identified tissue structure across all samples, concatenates them into a composite feature matrix, and applies clustering to align structures into a unified set. This allows the identification of shared architectural patterns while preserving sample-specific multicellular structures.

For comprehensive evaluation, we compared HRCHY-CytoCommunity not only with NeST[34], a method specifically designed for identifying hierarchical tissue structures, but also with six widely-used non-hierarchical tissue structure identification methods. This comparison was conducted to assess whether explicit hierarchical modeling yields more biologically meaningful results than repeatedly adjusting the resolution of flat structural identification methods to infer multi-level organization. The methods include five deep learning-based approaches (CellCharter[21], NicheCompass[20], GraphST[15], SpaGCN[22], and SpaSEG[23]) as well as the seminal spatial domain detection method (Giotto Suite[24]) (Supplementary Data 1).

### Performance evaluation on coarse-grained TC identification

To evaluate the ability of HRCHY-CytoCommunity in identifying coarse-grained tissue structures, we first applied it to an imaging-

**a   Soft hierarchical tissue structure assignment (base module)**

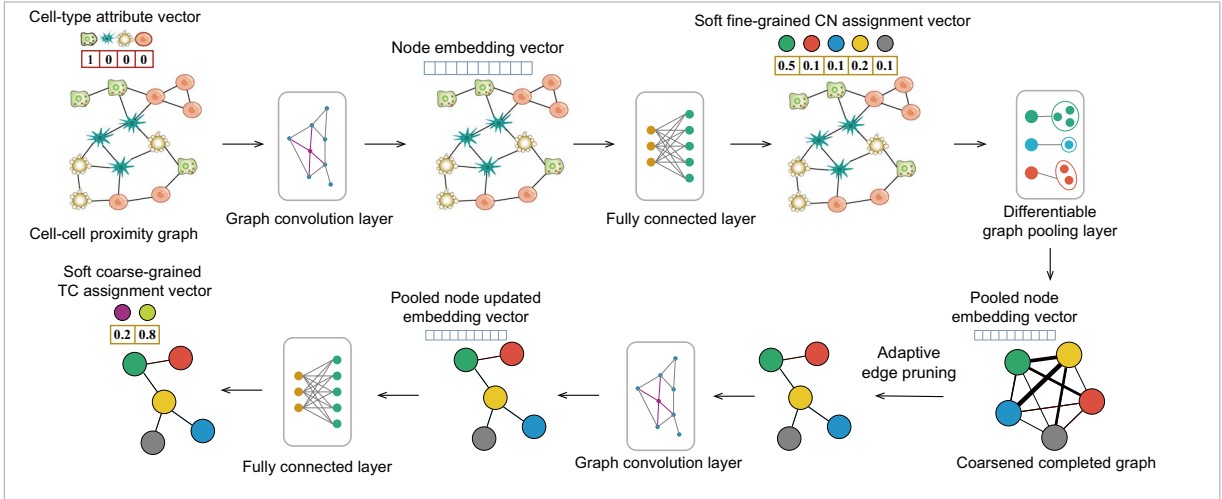

**b   Robust hierarchical tissue structure identification**

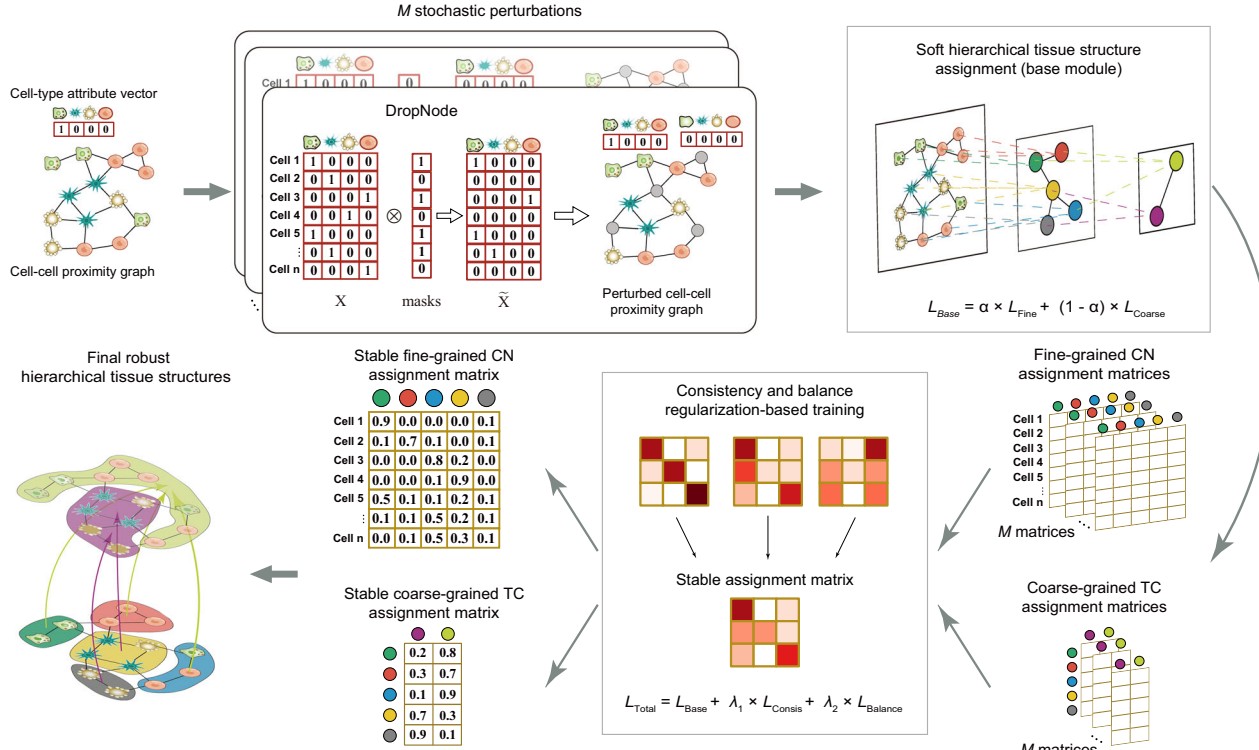

**Fig. 1 | Overview of the HRCHY-CytoCommunity framework.** The framework formulates hierarchical tissue structure identification as a hierarchical community detection problem on a cell-cell proximity graph. HRCHY-CytoCommunity includes two modules: **a** a soft hierarchical structure assignment base module and **b** a robust hierarchical structure identification module. **a** The base module begins with a cell-cell proximity graph, where nodes represent cells, and their attributes are one-hot encoded cell-type vectors. A graph convolution layer followed by a fully-connected layer transforms node attribute vectors into soft fine-grained cellular neighborhood (CN) assignment vectors. A differentiable graph pooling layer is then used to generate a coarsene,d completed graph, where each pooled node represents a CN. Adaptive edge pruning is applied to mitigate over-smoothing. Another graph convolution and fully-connected layers process the pooled node embeddings to produce soft coarse-grained tissue compartment (TC)

assignment vectors. The entire process is optimized by a graph minimum cut (MinCut)-based loss function, denoted as $L_{Base}$. **b** To enhance the accuracy and robustness of hierarchical tissue structure identification, $M$ perturbed cell-cell proximity graphs are generated via DropNode (random masking of node attribute vectors). Each graph is processed by the base module to produce soft fine-grained CN and coarse-grained TC assignment matrices. Consistency and balance regularization during training enforces stability across these $M$ assignments. The total loss, denoted as $L_{Total}$, combines the base MinCut loss ($L_{Base}$) with a consistency regularization term ($L_{Consis}$, which minimizes divergence across $M$ assignments) and a balance regularization term ($L_{Balance}$, which prevents cluster collapse). Finally, this module yields stable assignment matrices, from which final robust hierarchical tissue structures are derived.

based spatial proteomics dataset of mouse spleen generated through the Co-Detection by Indexing (CODEX) technology[36]. This dataset comprises three healthy mouse spleen samples (BALBc-1, BALBc-2, and BALBc-3) with an average of 81,760 cells per sample. Each sample

was annotated into 27 distinct cell types (Fig. 2a, left column) and two major functional compartments: the red pulp and the lymphoid compartment, the latter comprising the periarteriolar lymphoid sheath, B-zone, and marginal zone[37–39]. These manually annotated

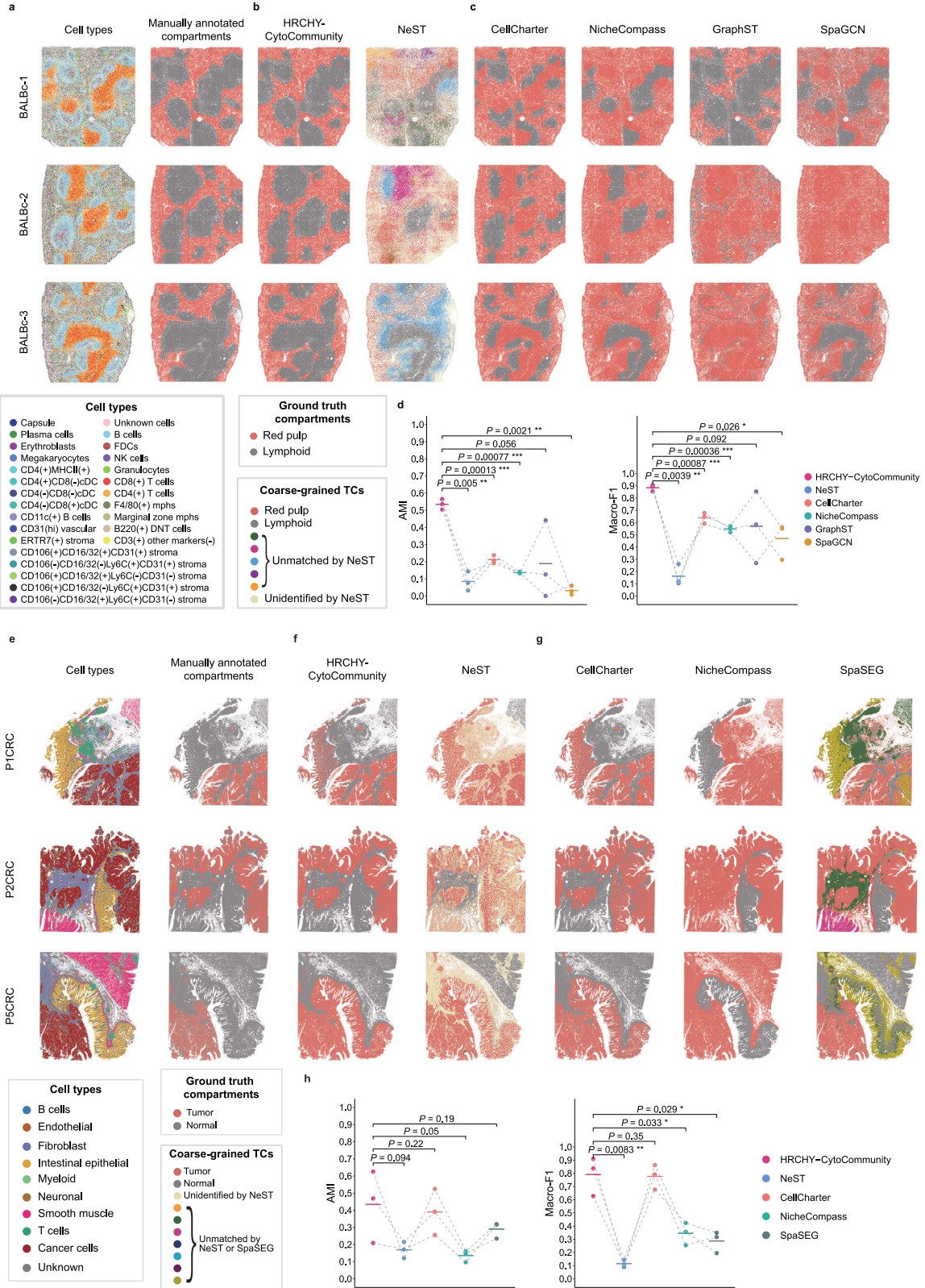

compartments served as the gold standard for the coarse-grained TCs (Fig. 2a, right column). To ensure a rigorous and fair comparison, HRCHY-CytoCommunity was evaluated against NeST and four non-hierarchical tissue structure identification methods (Supplementary Data 1). However, Giotto Suite was excluded from this analysis due to its computational limitations in handling large-scale spatial omics datasets. SpaSEG was also excluded from benchmarking in this

dataset because it is not applicable for spatial proteomics data (Supplementary Note 7).

Using a cluster stability-based criterion to determine the optimal numbers of coarse-grained TCs and fine-grained CNs (Methods and Supplementary Fig. 2a), HRCHY-CytoCommunity identified two TCs and seven CNs. Across all samples, the two TCs corresponded accurately to the red pulp and the lymphoid compartment. Notably,

**Fig. 2 | Performance evaluation of HRCHY-CytoCommunity on coarse-grained TC identification. a** Spatial maps of healthy mouse spleen samples generated by CODEX technology. Cells are colored by cell types (left) and manually annotated compartments (right), which serve as the ground truth for coarse-grained TCs. **b** Coarse-grained TCs identified by HRCHY-CytoCommunity (left) and NeST (right). **c** Coarse-grained TCs identified by non-hierarchical methods, including CellChar-ter, NicheCompass, GraphST, and SpaGCN. **d** Adjusted Mutual Information (AMI) and Macro-F1 scores calculated using manual compartment annotations. **e** Spatial maps of human CRC samples (8 μm bins) generated by Visium HD technology. Bins are colored by cell types (left) and manually annotated coarse-grained structures (right). **f** Coarse-grained TCs identified by HRCHY-CytoCommunity (left) and NeST (right). **g** Coarse-grained TCs identified by non-hierarchical methods, including CellCharter, NicheCompass, and SpaSEG. **h** AMI and Macro-F1 scores calculated using manual compartment annotations. In both **d** and **h**, each point corresponds to the performance on an individual sample, with horizontal bars indicating the mean performance across n = 3 samples. Points from the same sample are connected by grey dashed lines. *P*-values were calculated using one-sided paired *t*-tests. *, *P*-value < 0.05; **, *P*-value < 0.01; ***, *P*-value < 0.001. Unidentified indicates cells not assigned to any TC. Unmatched denotes TCs that could not be aligned with manual annotations. Source data are provided as a Source Data file.

although the lymphoid compartment is a spatially discontinuous structure, HRCHY-CytoCommunity successfully captured its distribution (Fig. 2b, left column). In contrast, NeST identified hierarchical structures starting from single-gene hotspots[34], thus some cells were not assigned to any TC (labeled as unidentified). Moreover, NeST often splits the spatially discontinuous lymphoid compartment into multiple distinct TCs (Fig. 2b, right column). Non-hierarchical tissue structure identification methods also exhibited limited accuracy in delineating coarse-grained TCs. For example, CellCharter captured only B cell-enriched regions within the lymphoid compartment (Fig. 2c, first column), while NicheCompass and SpaGCN detected only T cell (CD4+ and CD8+)-enriched areas of the lymphoid compartment (Fig. 2c, second and fourth columns). GraphST correctly identified TC distributions in sample BALBc-1 but failed to distinguish the lymphoid compartment in the other two samples (Fig. 2c, third column). To quantitatively evaluate performance, we measured the concordance between identified TCs and manually annotated compartments using adjusted mutual information (AMI) and Macro-F1 scores (Methods). The results demonstrated that HRCHY-CytoCommunity achieved significantly higher AMI and Macro-F1 scores than most state-of-the-art methods (one-sided paired *t*-test *P*-values < 0.05), while exhibiting performance comparable to GraphST (Fig. 2d).

To further evaluate the ability of HRCHY-CytoCommunity to identify coarse-grained tissue structures in sequencing-based spatial omics data, we applied it to a spatial transcriptomic dataset of human colorectal cancer (CRC) generated through the Visium HD technology[3], which profiles 18,085 genes (Supplementary Data 1). This dataset comprises three samples (named as P1CRC, P2CRC, and P5CRC). On average, each sample contained 420,502 bins at 8 μm resolution covering 10 major cell types (Fig. 2e, left column) and two major compartments: the tumor and the normal tissue compartment, serving as the gold standard for coarse-grained TCs (Fig. 2e, right column). For benchmarking, HRCHY-CytoCommunity was compared with NeST and three non-hierarchical spatial domain detection methods (Supplementary Data 1). Due to memory constraints associated with processing large-scale spatial omics data, GraphST, SpaGCN, and Giotto Suite were excluded from the analysis (Supplementary Note 7).

Based on the cluster stability criterion, HRCHY-CytoCommunity identified two coarse-grained TCs and ten fine-grained CNs (Supplementary Fig. 3a). In sample P2CRC, HRCHY-CytoCommunity accurately distinguished tumor and normal tissue compartments. Similarly, in samples P1CRC and P5CRC, the two compartments were clearly delineated, though with moderate inclusion of intestinal epithelial-enriched regions within the tumor compartment (Fig. 2f, left column). In contrast, NeST misclassified the majority of spatial bins as tumor regions and left most normal tissue regions unassigned (labeled as unidentified), limiting the interpretability of the resulting tissue structures (Fig. 2f, right column). Among the non-hierarchical methods, CellCharter exhibited competitive overall performance relative to HRCHY-CytoCommunity, but it misclassified epithelial-enriched regions as tumor compartments in sample P2CRC (Fig. 2g, first column). Meanwhile, NicheCompass incorrectly assigned large numbers of immune and stromal cells to the tumor compartment, while SpaSEG accurately identified the tumor compartments but split the normal

compartments into multiple distinct TCs (Fig. 2g, second and third columns). Quantitative evaluation confirmed that HRCHY-CytoCommunity outperformed other methods, showing a statistically significant improvement in Macro-F1 score (one-sided paired *t*-test, *P*-values < 0.05) and a performance level comparable to CellCharter. On the AMI metric, HRCHY-CytoCommunity achieved a higher average score, albeit without statistical significance (Fig. 2h).

We further investigated the biological relevance of the fine-grained CNs identified by HRCHY-CytoCommunity in these two datasets. In the absence of ground-truth labels for fine-grained structures, HRCHY-CytoCommunity consistently uncovered spatially coherent and biologically interpretable CNs that aligned with established anatomical and functional structures (Supplementary Note 1). For instance, in the mouse spleen, CNs accurately corresponded to anatomical regions such as the B-cell zone and the periarteriolar lymphoid sheath, while in the human CRC, CNs represented distinct tumor-core and tumor-edge regions along with immune cell infiltrations.

In summary, HRCHY-CytoCommunity outperforms existing state-of-the-art hierarchical and non-hierarchical tissue structure identification methods in identifying coarse-grained TCs across evaluated datasets. Moreover, the fine-grained CNs detected by our method show stronger correspondence to known biological structures, demonstrating its ability to reveal meaningful multi-scale spatial organizations in complex tissues.

## Performance evaluation on fine-grained CN identification

To further assess the capability of HRCHY-CytoCommunity in identifying fine-grained CNs, we first applied it to an imaging-based spatial transcriptomics dataset of the healthy mouse hypothalamic preoptic region. The dataset was generated using the Multiplexed Error-Robust Fluorescence in situ Hybridization (MERFISH) technology[7], profiling 155 genes. It comprises five samples representing distinct brain regions, named as Bregma-0.04, Bregma-0.14, Bregma+0.06, Bregma +0.16, and Bregma+0.26 based on their relative distances to the bregma. Each sample contains multiple small and symmetric hypothalamic nuclei regions, which were delineated in the original study through manual inspection of well-characterized brain histology[7,40]. We previously manually mapped these nuclei onto single-cell spatial maps that served as the ground truth for fine-grained CN evaluation (Fig. 3a). For benchmarking, HRCHY-CytoCommunity was compared with NeST and six non-hierarchical spatial domain detection methods (Supplementary Data 1).

Using the cluster stability criterion, HRCHY-CytoCommunity identified 12 fine-grained CNs and two coarse-grained TCs (Supplementary Fig. 4a). This model successfully identified multiple symmetric CNs that corresponded to established hypothalamic nuclei (Fig. 3b, left column). For instance, BNST (Bed Nucleus of the Stria Terminalis), LPO (Lateral Preoptic Area), and MPA (Medial Preoptic Area) were consistently identified across all five samples. Relatively larger nuclei, such as ACA (Anterior Commissure, Anterior Part) and MPN (Medial Preoptic Nucleus), were detected in most samples, while relatively smaller nuclei, including PS (Parastrial Nucleus), Pe (Periventricular Nucleus), and PaAP (Paraventricular Nucleus) were identified in several samples. Our method also captured sample-specific

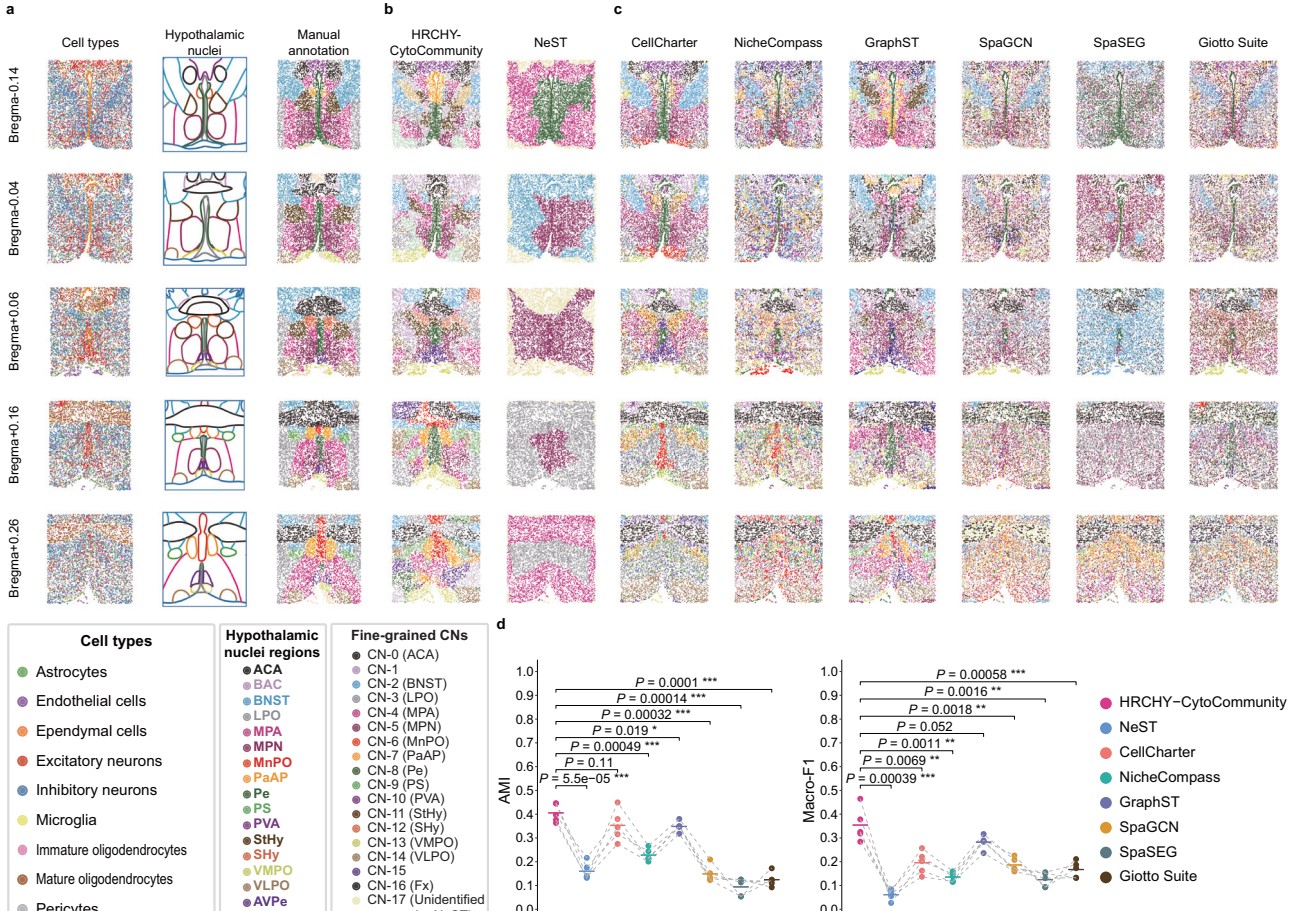

**Fig. 3 | Performance evaluation of HRCHY-CytoCommunity on fine-grained CN identification. a** Spatial maps of mouse hypothalamic preoptic region samples generated by MERFISH technology. Cells are colored by cell types (left) and manually annotated hypothalamic nuclei (right), which were generated based on the outlines of hypothalamic nuclei (middle) and serve as the ground truth for fine-grained CNs. **b** Fine-grained CNs identified by HRCHY-CytoCommunity (left) and NeST (right). **c** Fine-grained CNs identified by non-hierarchical methods, including CellCharter, NicheCompass, GraphST, SpaGCN, SpaSEG, and Giotto Suite.

**d** Adjusted Mutual Information (AMI) and Macro-F1 scores calculated using manually annotated hypothalamic nuclei. Each point corresponds to the performance on an individual sample, with horizontal bars indicating the mean performance across n = 5 samples. Points from the same sample are connected by grey dashed lines. P-values were calculated using one-sided paired t-tests. *, P-value < 0.05; **, P-value < 0.01; ***, P-value < 0.001. Unidentified indicates cells not assigned to any CN. Source data are provided as a Source Data file.

structures, such as PVA (Paraventricular Nucleus, Anterior Part) and Fx (Fornix) in the Bregma-0.14 sample and SHy (Suprachiasmatic nucleus shell) in the Bregma+0.06 sample. In contrast, NeST identified only one or two CNs per sample (Fig. 3b, right column), showing poor correspondence to the manual annotations. This limitation likely stems from NeST's reliance on gene expression features without pre-specifying the number of tissue structures and its inability to identify spatially discontinuous parts of the same nuclear structure. Comparison with six additional non-hierarchical spatial domain detection methods (Supplementary Data 1) revealed that while most methods preserved the general symmetry of identified CNs, their results often contained intermixed CNs, leading to suboptimal performance (Fig. 3c). Among them, GraphST and CellCharter performed relatively well. GraphST correctly identified relatively large CNs such as BNST, but intermixed CN boundaries reduced its accuracy. CellCharter achieved superior identification of BNST, MPA, and MPN in one sample (Bregma-0.14). However, these methods consistently failed to capture smaller regions like PS and PaAP across other samples, resulting in overall inferior performance compared to HRCHY-CytoCommunity. Quantitative evaluation demonstrated that HRCHY-CytoCommunity significantly outperformed NeST (one-sided paired t-test, P-values < 0.001) and all other non-hierarchical methods (one-sided paired t-test,

P-values < 0.05) in both AMI and Macro-F1 metrics, with the exception of CellCharter on the AMI score and GraphST on the Macro-F1 score, where the differences were not statistically significant (Fig. 3d).

To further evaluate the performance of HRCHY-CytoCommunity on sequencing-based spatial transcriptomics data, we applied it to a mouse intracerebral hemorrhage Stereo-seq dataset, which profiled 23,625 genes. This analysis focused on nine representative samples, with an average of 81,831 cells per sample, encompassing 27 molecularly defined regions that served as the ground truth for CNs (Supplementary Fig. 5a). HRCHY-CytoCommunity was compared with NeST and four non-hierarchical spatial domain detection methods for benchmarking (Supplementary Data 1). Giotto Suite and SpaSEG were excluded due to excessive memory requirements (Supplementary Note 7).

HRCHY-CytoCommunity identified 20 fine-grained CNs and three coarse-grained TCs (Supplementary Fig. 6a), successfully reconstructing symmetric CNs throughout the mouse brain. Specifically, our method accurately resolved biologically meaningful CNs, including cortical layers 1 to 6, olfactory areas (OLF), meninges, and pallidum. In contrast, CNs identified by NeST cannot correspond to known molecularly defined regions in most samples (Supplementary Fig. 5b). Among the non-hierarchical methods, while NicheCompass produced

the most competitive results, benefiting from its cell-cell communication modeling, HRCHY-CytoCommunity demonstrated superior sensitivity in detecting subtle structures such as OLF in the Naive-1 sample and islands of Calleja in the D1-3 sample (Supplementary Fig. 5b, left column), which were not well captured by NicheCompass (Supplementary Fig. 5c, second column). This enhanced sensitivity may arise from HRCHY-CytoCommunity's use of cell-type annotations as input features, enabling better discrimination of closely related cell subtypes, though this design occasionally resulted in over-segmentation. Quantitatively, HRCHY-CytoCommunity achieved significantly higher AMI and Macro-F1 scores than NeST (one-sided paired $t$-test, $P$-values < 1E-7) and other non-hierarchical methods (one-sided paired $t$-test, $P$-values < 0.001), with performance comparable to NicheCompass (Supplementary Fig. 5d).

We further examined the coarse-grained TCs identified by HRCHY-CytoCommunity in these two datasets. The analysis revealed that our method consistently detected biologically meaningful TCs across multiple samples, including distinct compartments enriched for neuronal subtypes (e.g., excitatory and inhibitory neurons) and non-neuronal cells in the hypothalamic data, as well as anatomical regions such as the brain stem, cortex, and striatum in the intracerebral hemorrhage data (Supplementary Note 2; Supplementary Figs. 4c and 6c). These findings confirm that HRCHY-CytoCommunity robustly reconstructs hierarchically organized tissue structures that correspond to biologically relevant domains, demonstrating its applicability across both imaging-based and sequencing-based spatial transcriptomics platforms.

## Cross-platform generalization of HRCHY-CytoCommunity to low-resolution spatial transcriptomics

To evaluate the generalization ability of HRCHY-CytoCommunity on spot-based spatial transcriptomics data lacking single-cell-resolution cell-type annotations, we applied it to two representative datasets: a mouse hippocampus Slide-seq V2 dataset and a human breast cancer Visium V1 dataset (Supplementary Data 1). For each dataset, we first inferred the cell-type composition of each spot using RCTD[41], a state-of-the-art deconvolution method (Supplementary Figs. 7a, 8a). The inferred cell-type fractions were used as node attributes to construct a spot-spot proximity graph, which was then input into HRCHY-CytoCommunity.

We first evaluated the method on the mouse hippocampus Slide-seq V2 data (10 μm-spot resolution), using the annotated Allen Mouse Brain Atlas[42] as the ground-truth reference (Fig. 4a, b, left panels). HRCHY-CytoCommunity identified three coarse-grained TCs and 11 fine-grained CNs (Supplementary Fig. 7b), showing strong alignment with the reference atlas. Specifically, at the coarse-grained level, the identified TCs accurately corresponded to major anatomical structures, including the cortical region, hippocampus, and brain stem (Fig. 4a, middle panel). At the fine-grained level, the method successfully recovered substructures, such as CA1-CA3 regions, dentate gyrus, corpus callosum, and distinct cortical layers (Fig. 4b, middle panel). In contrast, although NeST detected several CNs that matched the reference atlas (e.g., CA1-CA3 regions, dentate gyrus, and corpus callosum), it produced small and fragmented substructures rather than large-scale coherent TCs at the coarse-grained level. Moreover, NeST left a considerable proportion of spots unassigned (labeled as unidentified) for both TCs and CNs, limiting the interpretability of the hierarchical tissue organization (Fig. 4a, b, right panels).

We next applied HRCHY-CytoCommunity to the human breast cancer Visium V1 data (55 μm-spot resolution), using manual annotations of hierarchical tissue structures from Xu et al.[43] as ground-truth reference (Fig. 4c, d, left panels). HRCHY-CytoCommunity identified two coarse-grained TCs and nine fine-grained CNs (Supplementary Fig. 8b). The method accurately distinguished tumor and non-tumor compartments (Fig. 4c, middle panel), whereas NeST incorrectly partitioned discontinuous tumor regions into multiple fragmented TCs

(Fig. 4c, right panel). At the fine-grained level, both methods identified CNs consistent with manual annotations, such as DCIS/LCIS_1, DCIS/LCIS_3, and IDC_4, but NeST left the Healthy_1 region unassigned, reducing its interpretability (Fig. 4d, middle and right panels). Notably, both methods reclassified the manually annotated IDC_3 tumor region as a CN (e.g., CN-8 in our results) belonging to the non-tumor compartment (Fig. 4d, red arrowhead). Cell-type enrichment analysis revealed significant co-enrichment of B-cells and T-cells within this CN (Methods; one-sided Wilcoxon rank-sum test, $P$-values < 0.001; Fig. 4e). Differential expression analysis ($|\log_2 FC| > 0.25$) further indicated strong humoral immune activity in this region, characterized by up-regulation of immunoglobulin constant region genes (e.g., *IGHG1/3/4*, *IGHA1*, *IGHM*, *IGKC*, *IGLC1/2/3* and *JCHAIN*), consistent with a plasma cell-enriched microenvironment[44–46] (Fig. 4f). These results suggest that the original manual annotation of IDC_3 as tumor tissue was likely inaccurate, and that HRCHY-CytoCommunity more reliably captured the underlying biologically relevant tissue organization.

In summary, HRCHY-CytoCommunity generalizes effectively across low-resolution spatial transcriptomics platforms, consistently identifying biologically meaningful hierarchical tissue structures even in the absence of high-quality single-cell-resolution cell-type annotations. In contrast to NeST, which often produces fragmented substructures and leaves many spots unassigned, our method captures both large- and small-scale spatially coherent tissue structures and achieves complete cellular coverage.

## Cross-sample hierarchical tissue structures in triple-negative breast cancer

To further evaluate whether the hierarchical tissue structures identified by HRCHY-CytoCommunity can capture inter-patient heterogeneity in multicellular organization within tumor tissues, we applied our method to a triple-negative breast cancer (TNBC) dataset generated using the Multiplexed Ion Beam Imaging by Time-Of-Flight (MIBI-TOF) technology[10] (Supplementary Data 1). This dataset consists of 15 patients with compartmentalized tumors, characterized by clear spatial segregation between immune and neoplastic cell populations. Our analysis had two main objectives: (1) to assess whether HRCHY-CytoCommunity-identified coarse-grained TCs accurately correspond to immune and neoplastic cell-dominated regions compared to NeST; and (2) to investigate whether fine-grained CNs reflect inter-patient heterogeneity even among tumors broadly classified as compartmentalized at the macroscopic level.

HRCHY-CytoCommunity identified two TCs and 14 fine-grained CNs across the dataset (Supplementary Fig. 9a). In most samples, our method successfully delineated one TC dominated by immune cells and another by neoplastic cells, consistent with the spatial distribution patterns of cell types expected in compartmentalized tumors. In contrast, NeST identified only a single TC in multiple samples (e.g., patients 9, 10, and 16), likely due to limitations in feature availability preventing simultaneous identification of both compartments (Fig. 5a; Supplementary Fig. 9b). In patient 28, where immune and neoplastic cells are spatially contiguous, both methods effectively identified the TCs. However, in patient 32, where the tumor exhibits spatially discontinuous immune and neoplastic cell populations, HRCHY-CytoCommunity correctly separated them into distinct TCs, whereas NeST partitioned them into five distinct TCs (Fig. 5a, fourth row). To quantitatively evaluate TC identification performance, we calculated the proportion of neoplastic and immune cells that were correctly assigned to neoplastic and immune cell-dominated TCs. HRCHY-CytoCommunity showed significantly better performance across the 15 patients than NeST (one-sided paired $t$-test, $P$-value = 0.024; Fig. 5b).

While compartmentalized tumors can be broadly categorized into immune and neoplastic TCs, fine-grained CN analysis can reveal deeper heterogeneity within these tumors. We performed cross-sample integration analysis of CNs from all patients and generated a unified set

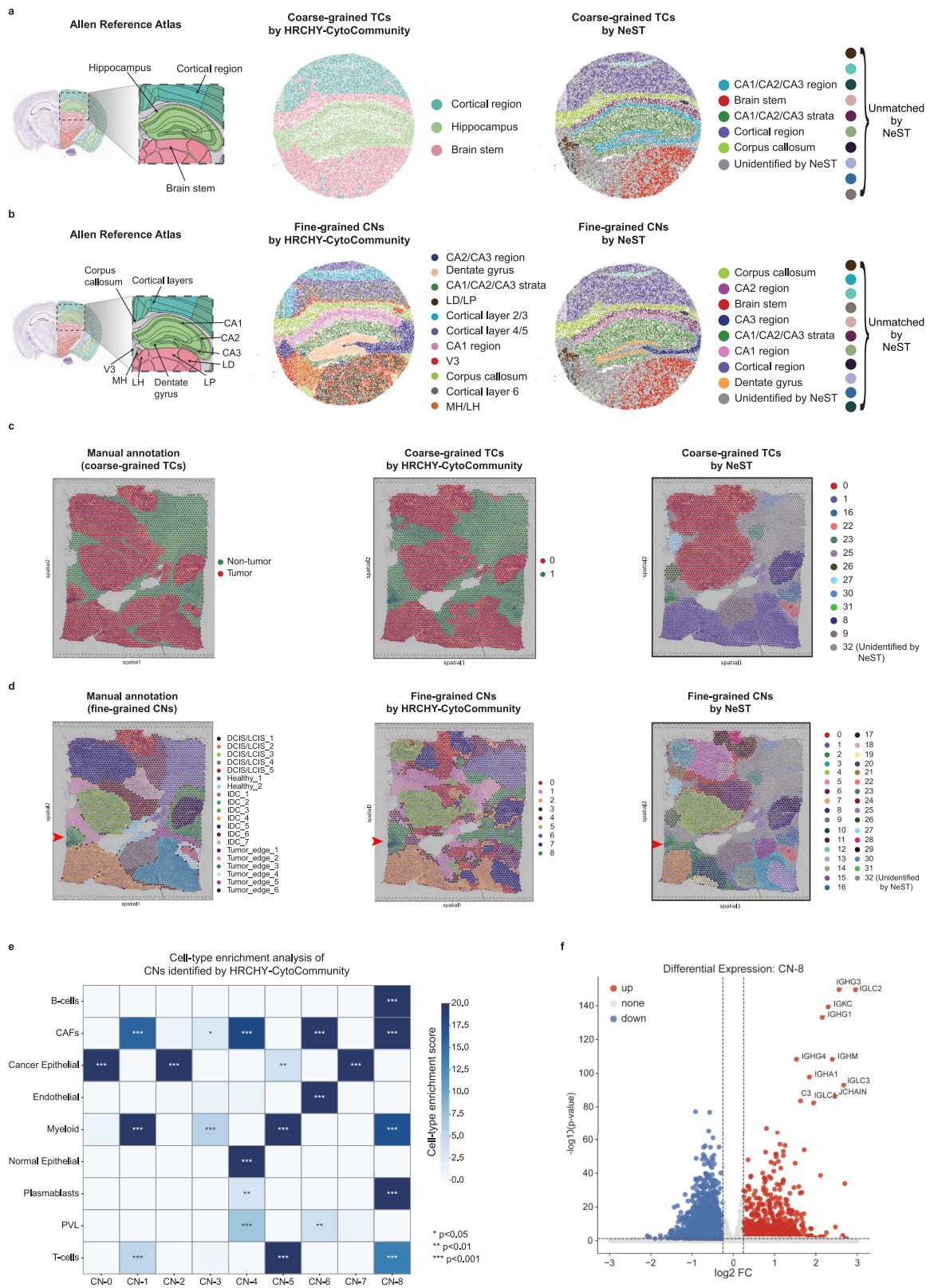

of 26 CNs (Methods; Fig.5c; Supplementary Fig. 10). We found that these unified CNs captured both conserved and patient-specific biological features. For example, both patients 4 and 9 contained unified CN-6 (neoplastic cell-enriched) in their neoplastic TCs and unified CN-22 (CD4+ T cell- and B cell-enriched) in their immune TCs. However, unified CN-8 (macrophage-, CD8+ T cell- and endothelial cell-enriched) was unique to patient 4's immune TC, and unified CN-20 (neoplastic-

neutrophil mixed) appeared only in patient 9's neoplastic TC (Fig. 5c, d). These findings indicate that even among compartmentalized tumors, individuals maintain distinct multicellular organizational patterns.

We further stratified the patients by performing hierarchical clustering based on the unified CN composition profiles of each sample. While some patients (e.g., patients 3 and 32) exhibited highly

**Fig. 4 | Evaluation of HRCHY-CytoCommunity's cross-platform generalization using the mouse hippocampus Slide-seq V2 and human breast cancer Visium V1 datasets. a** Coarse-grained TC reference from the Allen Mouse Brain Atlas[42] (left), and TCs identified by HRCHY-CytoCommunity (middle) and NeST (right) in the mouse hippocampus. **b** Fine-grained CN reference from the Allen Mouse Brain Atlas (left), and CNs identified by HRCHY-CytoCommunity (middle) and NeST (right) in the mouse hippocampus. V3, third ventricle; MH, medial habenula; LH, lateral habenula; LP, lateral posterior nucleus of the thalamus; LD, lateral dorsal nucleus of the thalamus. **c** Manual annotation of coarse-grained TCs from Xu et al.[43]. (left), and TCs identified by HRCHY-CytoCommunity (middle) and NeST (right) in the human breast cancer. **d** Manual annotation of fine-grained CNs from Xu et al. (left), and CNs identified by HRCHY-CytoCommunity (middle) and NeST (right) in the human breast cancer. **e** Heatmap showing cell-type enrichment scores for CNs identified by HRCHY-CytoCommunity. Enrichment score was defined as -log$_{10}$(adjusted $P$-value). $P$-values were computed using one-sided Wilcoxon rank-sum tests and adjusted with the Benjamini-Hochberg method. *, $P$-value < 0.05; **, $P$-value < 0.01; ***, $P$-value < 0.001. Exact adjusted $P$-values are provided in the Source Data file. **f** Differential gene expression analysis of CN-8 (identified by HRCHY-CytoCommunity) versus other CNs. Each point represents a gene, with the y-axis showing -log$_{10}$(adjusted $P$-value) and the x-axis showing log$_2$(FoldChange). FC, Fold Change. $P$-values were computed using two-sided Wilcoxon rank-sum tests, with significance thresholds set at |log$_2$FC| > 0.25 and $P$-value < 0.05. Unidentified indicates cells not assigned to any TC or CN. Unmatched denotes TCs or CNs that could not be aligned with manual annotations. Red arrowheads indicate the IDC_3 region manually annotated by Xu et al., which was reclassified by both HRCHY-CytoCommunity and NeST as a non-tumor CN. Source data are provided as a Source Data file.

similar unified CN compositions, others (e.g., patients 9 and 10) showed clear divergence (Fig. 5e, left; Supplementary Fig. 10). To determine whether these clustering patterns reflect inherent spatial organizational heterogeneity rather than mere differences in cell-type abundance, we also performed the clustering using cell-type composition profiles. The results indicated that although some patients with similar cell-type compositions (e.g., patients 28 and 35) also shared comparable CN compositions, it was more common to observe marked differences in unified CN configurations among patients with highly similar cell-type distributions, as exemplified by patients 4 and 41 (Fig. 5e; Supplementary Fig. 10). These findings suggest that fine-grained CNs capture inter-patient heterogeneity in spatial tissue organization that is not fully explained by cell-type proportion alone. Therefore, while coarse-grained TC analysis can distinguish compartmentalized from mixed tumors[10], the analysis of fine-grained CNs offers potential for further subclassification of patients.

## Hierarchical tissue structure-based stratification of breast cancer patients

To validate the biological significance and clinical relevance of the hierarchical tissue structures identified by HRCHY-CytoCommunity, we applied the method to a breast cancer spatial proteomics dataset generated using the imaging mass cytometry (IMC) technology[9]. The dataset consists of 263 breast cancer patients, including 190 survivors and 73 deceased individuals. HRCHY-CytoCommunity identified 30 unified TCs and 50 unified CNs across the cohort.

We first stratified patients into five distinct groups by performing hierarchical clustering based on the presence or absence of unified TCs as features (Fig. 6a, top) and conducted survival analysis using associated clinical data (Fig. 6b). Patients in Group 3, characterized by the presence of TC-13, exhibited significantly better prognosis compared to the remaining cohorts (hazard ratio = 0.2, $P$-value = 0.031; Fig. 6c). TC-13 represents a coarse-grained structure significantly enriched with CK7$^+$ neoplastic cells (Fig. 6a, bottom), consistent with previous findings that CK7$^+$ neoplastic cell phenotype is associated with favorable clinical outcomes[9].

We next focused on Group 1, which contained the largest number of patients but lacked obvious shared coarse-grained TC characteristics. To investigate whether finer-grained CN features could further stratify this group, we used the presence or absence of a specific unified CN as the clustering feature. This analysis revealed CNs with significant prognostic value (Fig. 6d, e; Supplementary Fig. 11), and we explored their potential functions through cell-type enrichment analysis (Methods; Fig. 6f). For example, within Group 1, a subgroup of 29 patients containing unified CN-21 showed significantly worse prognosis (hazard ratio = 2.2, $P$-value = 0.026; Fig. 6d). CN-21 was significantly enriched with small elongated fibroblasts in all 29 patients, and in some cases also exhibited significant enrichment of small circular fibroblasts or diverse cancer-associated fibroblast (CAF) subtypes, including Fibronectin$^{hi}$, Vimentin$^{hi}$, and SMA$^{hi}$ Vimentin$^{hi}$

fibroblasts (Fig. 6f). Accumulating evidence indicates that CAFs promote tumor growth, metastatic and therapeutic resistance[47–49], and may form a dense cellular barrier that impedes immune-neoplastic cell interactions[48,50]. In this study, we observed that CN-21, enriched with normal fibroblasts and CAFs, functioned as a fence-like structure separating immune cell-enriched CNs (e.g., CN-6 and CN-40) from neoplastic cell-enriched CNs (e.g., CN-8, CN-36, and CN-28), as illustrated in patients P14, P23, and P112 (Fig. 6g; Supplementary Fig. 12).

Another notable example was a 13-patient subgroup within Group 1 containing unified CN-34, which was associated with a markedly elevated risk of death (hazard ratio = 5.4, $P$-value < 0.001; Fig. 6e). CN-34 was significantly enriched with T cell subpopulation-1 in all 13 patients, and in some patients also showed significant enrichment of macrophages (Fig. 6f). This CN exhibited a spatially dispersed distribution surrounded by diverse neoplastic cell- or CAF-enriched CNs (e.g., CN-1, CN-3, and CN-47), as observed in patients P54, P80, and P191 (Fig. 6h; Supplementary Fig. 12). This spatial pattern suggests that immune cell infiltration into the tumor core may be obstructed, potentially leading to regional immune evasion[51].

In summary, these findings demonstrate that HRCHY-CytoCommunity enables hierarchical patient stratification by leveraging multi-scale tissue structures from coarse-grained TCs to fine-grained CNs. This approach reveals patient subgroups with distinct survival implications across different spatial scales of tissue organization.

## Ablation, robustness, and scalability analyses of HRCHY-CytoCommunity

To systematically evaluate the stability and scalability of HRCHY-CytoCommunity, we performed a series of complementary analyses, including ablation studies, sensitivity analyses, robustness assessments, and scalability analyses (Supplementary Notes 3–6).

Comprehensive ablation experiments demonstrated that each key component of the HRCHY-CytoCommunity model, including consistency and balance regularization, adaptive edge pruning, and adaptive $\alpha$ scheduling (Methods), contributed positively to overall performance (Supplementary Note 3; Supplementary Figs. 13–15). Sensitivity analyses indicated that the model maintained stable performance across a wide range of hyperparameter settings, including DropNode rate, $\lambda\_consis$, $\lambda\_balance$, the number of neighbors $K$ in the KNN graph (cell-cell proximity graph) construction, and the number of perturbations used for generating perturbed cell-cell proximity graphs (Supplementary Note 4; Supplementary Figs. 16–18). Robustness evaluation against cell-type label inaccuracies and variations in annotation resolution further confirmed that the model consistently identified reliable hierarchical tissue structures under noisy or heterogeneous input annotations (Supplementary Note 5; Supplementary Figs. 19 and 20). Finally, scalability analyses on large-scale datasets (containing up to 473k spots) demonstrated high computational efficiency in both runtime and memory usage. For example, on a dataset

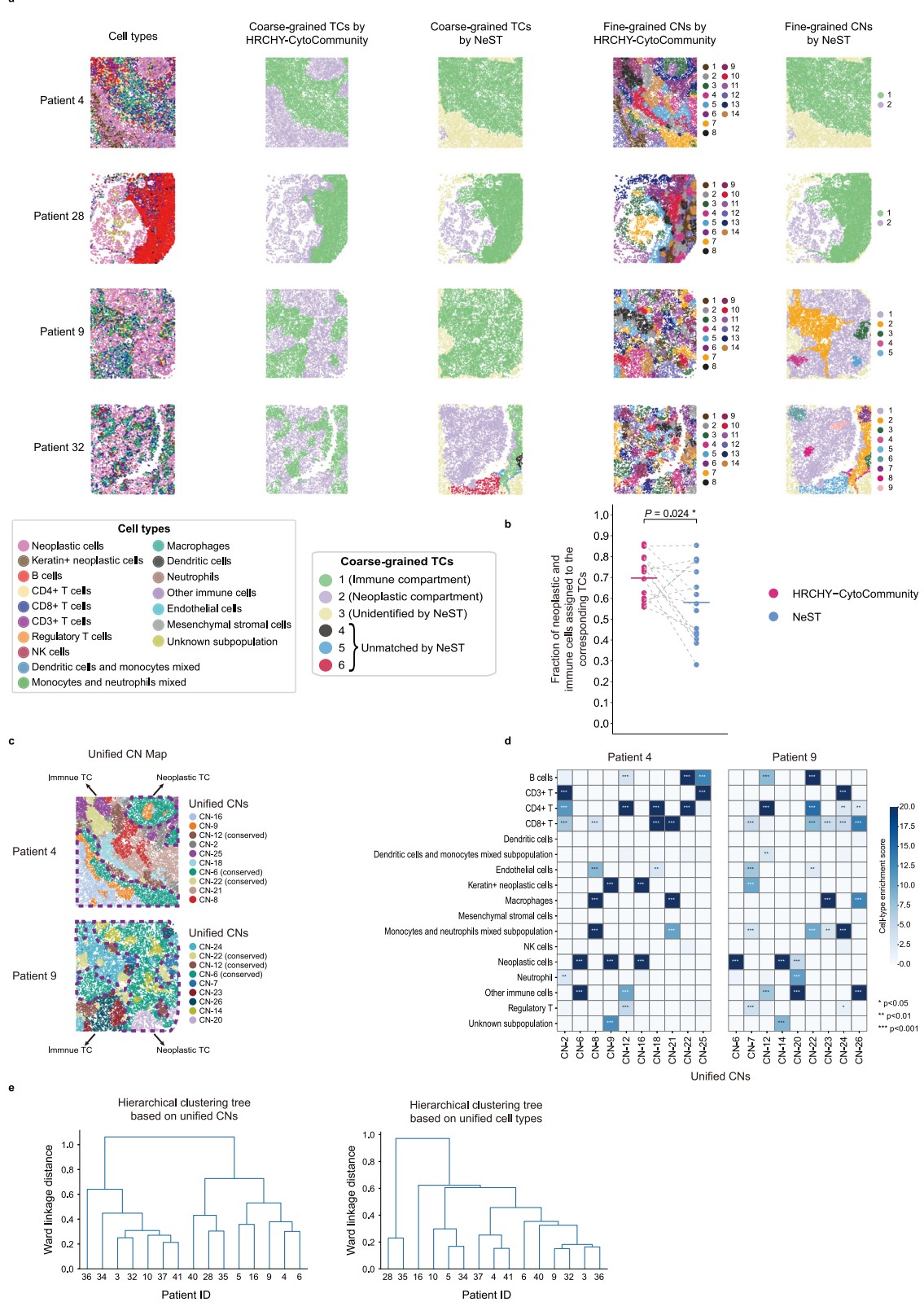

of 473k spots, HRCHY-CytoCommunity completed hierarchical tissue structure identification within 8.02 minutes, using only 8.69 GB of memory, confirming its capability to handle large-scale spatial omics datasets (Supplementary Note 6; Supplementary Figs. 21 and 22).

Taken together, these results demonstrate that HRCHY-CytoCommunity is robust, scalable, and generalizable across diverse data modalities.

## Discussion

We present HRCHY-CytoCommunity, a GNN-based framework designed to identify hierarchical tissue structures based on cell types and their spatial locations. By leveraging differentiable graph pooling, adaptive edge pruning, and consistency and balance regularization during training, HRCHY-CytoCommunity provides a robust, end-to-end approach that directly links single-cell features to multi-scale

**Fig. 5 | Cross-sample analysis of hierarchical tissue structures in the TNBC MIBI-TOF dataset. a** Representative single-cell spatial maps from four TNBC patients, showing compartmentalized tumors with clear spatial segregation between immune and neoplastic cells. Cells are colored by cell types (first column), coarse-grained TCs identified by HRCHY-CytoCommunity (second column) and NeST (third column), and fine-grained CNs identified by HRCHY-CytoCommunity (fourth column) and NeST (fifth column). Unidentified indicates cells not assigned to any TC or CN. **b** Proportion of neoplastic and immune cells correctly assigned to corresponding TCs. Each point corresponds to the performance on an individual sample, with horizontal bars indicating the mean performance across n = 15 samples. Points from the same sample are connected by grey dashed lines. *P*-value was calculated using a one-sided paired *t*-test. **c** Unified CN maps for patient 4 and patient 9. Cells are colored by unified CNs. Purple dashed lines outline neoplastic cell-dominated TCs, while remaining regions denote immune cell-dominated TCs. **d** Heatmaps showing cell-type enrichment scores for each unified CN in patient 4 and patient 9. Enrichment score was defined as $-\log_{10}$(adjusted *P*-value). *P*-values were computed using hypergeometric tests for enrichment (over-representation) and adjusted with the Benjamini-Hochberg method. Exact adjusted *P*-values are provided in the Source Data file. **e** Hierarchical clustering trees based on unified CN composition profiles (left) and cell-type composition profiles (right). Source data are provided as a Source Data file.

tissue architecture. Unlike existing hierarchical methods such as NeST[34], which relies on gene expression hotspots, HRCHY-CytoCommunity uses cell-type annotations as core features. This design makes it widely applicable to diverse spatial omics technologies, especially those with limited gene or protein features, that can characterize cellular phenotypes from complementary perspectives. Furthermore, in contrast to NeST, HRCHY-CytoCommunity produces fully nested hierarchical assignments with complete cellular coverage, offering clearer biological interpretability and enabling a more comprehensive understanding of how individual cells collectively build tissues and organs.

Different from non-hierarchical methods that rely on *post hoc* adjustment of clustering resolution to infer pseudo-hierarchies, HRCHY-CytoCommunity integrates the learning of multiple organizational levels within a unified end-to-end framework. This joint optimization approach enables information to flow bidirectionally across scales during training, promoting the identification of hierarchically consistent and functionally relevant tissue architectures. Extensive benchmarking on spatial omics datasets from various tissues, technologies, and modalities demonstrates that HRCHY-CytoCommunity consistently outperforms NeST and other non-hierarchical methods in terms of both accuracy and biological relevance. The incorporation of consistency and balance regularization further enhances its robustness and reproducibility. Moreover, through efficient sparse graph operations, HRCHY-CytoCommunity achieves high computational scalability, making it well-suited for large-scale and high-resolution spatial datasets.

To support cross-sample integration analysis, HRCHY-CytoCommunity includes an additional module that performs cell-type enrichment-based clustering, generating a unified set of nested multicellular structures across all samples. This addresses the challenge of inconsistent structural labels between samples. Applied to a TNBC dataset, the method successfully identified both conserved and patient-specific tissue architectures. Furthermore, using hierarchical tissue structures as features in survival analysis, we hierarchically stratified breast cancer patients from an IMC dataset into prognostically distinct groups, underscoring the clinical relevance of multi-scale spatial organization and suggesting new avenues for exploring hierarchical tumor microenvironments and related immunotherapeutic strategies.

As HRCHY-CytoCommunity explicitly models tissue organization as a hierarchy, it is particularly well-suited for organs with clearly stratified anatomical or functional layers. Its performance may be limited in tissues lacking strong hierarchical organization, such as liver, where metabolic zonation is continuously distributed along the porto-central axis[52]. Moreover, while the use of discrete cell-type annotations enhances the method's generalizability across platforms, it may not fully capture structures defined by continuous molecular gradients, transient states, or cell-cell communication signals. Future versions could incorporate multi-view learning to integrate such continuous features. In the meantime, methods like NicheCompass[20] (focused on cellular communication-driven structures) and ONTraC[53] (designed for spatially continuous niche trajectories) can complement HRCHY-CytoCommunity to provide a more comprehensive perspective on

tissue organization. Further extensions could also incorporate graph sampling or alignment techniques to improve cross-sample integration within an end-to-end analytical workflow. Lastly, although this study focused on a two-level hierarchy, the underlying framework of HRCHY-CytoCommunity can be readily extended to deeper hierarchies (Supplementary Note 8). Future work may explore adaptive mechanisms to automatically infer the optimal hierarchical depth or to model tissues with non-uniform organizational principles.

As spatial omics technologies continue to evolve, there is a growing need for versatile and scalable computational methods that can decipher tissue organization across platforms with varying resolution and feature throughput. By focusing on cell-type spatial maps, a common output across spatial omics technologies, HRCHY-CytoCommunity represents a general and scalable framework for simultaneously identifying multi-level hierarchical tissue structures. It provides a foundational tool for decoding principles of tissue organization by systematically bridging spatial hierarchies from single cells to multicellular modules, and ultimately to intact tissues and organs.

## Methods

### Soft hierarchical tissue structure assignment

We began by constructing an undirected KNN graph as the cell-cell proximity graph to represent the spatial omics data, where each node corresponds to a cell or spot. For single-cell-resolution datasets, the node attribute vector was constructed using one-hot encoding to capture cell type information (Fig. 1a). For low-resolution datasets, the node attribute vector was constructed using the estimated cell-type composition of each spot. The graph was built by computing Euclidean distances between nodes (cells) based on their spatial coordinates, connecting each node to its K nearest neighbors (excluding itself).

For the undirected KNN graph with $n$ nodes, we applied a single graph convolution layer[54] with the ReLU activation function to produce a node embedding matrix $Z^{(1)} \in \mathbb{R}^{n \times d}$, formulated as

$$Z^{(1)} = \mathrm{ReLU}\left(\mathrm{GNN}_1\left(X^{(1)}, A^{(1)}; \theta_{\mathrm{GNN}_1}\right)\right) \qquad (1)$$

where each row of $Z^{(1)}$ is a learned $d$-dimensional embedding vector of a cell. $X^{(1)} \in \mathbb{R}^{n \times t}$ denotes the initial cell-type attribute matrix, and $t$ is the total number of cell types. $A^{(1)} \in \{0, 1\}^{n \times n}$ is the adjacency matrix of the undirected KNN graph. The graph convolution operator was defined as

$$\mathbf{z_i^{(1)}} = \mathrm{ReLU}\left(\theta_1 \mathbf{x_i^{(1)}} + \theta_2 \sum_{j \in N(i)} \mathbf{x_j^{(1)}}\right) \qquad (2)$$

where $\mathbf{x_i}$ is the embedding vector of node $i$, and $N(i)$ denotes the first-order neighborhood derived from the matrix $A^{(1)}$. $\theta_1$ and $\theta_2$ are trainable parameters in the graph neural network $\mathrm{GNN}_1$. The embedding dimension $d$ was empirically set to 128 in this study.

Next, we used a fully-connected neural network with no hidden layers (also referred to as a linear layer) and a Softmax activation function to convert the node embedding matrix $Z^{(1)} \in \mathbb{R}^{n \times d}$ into a soft

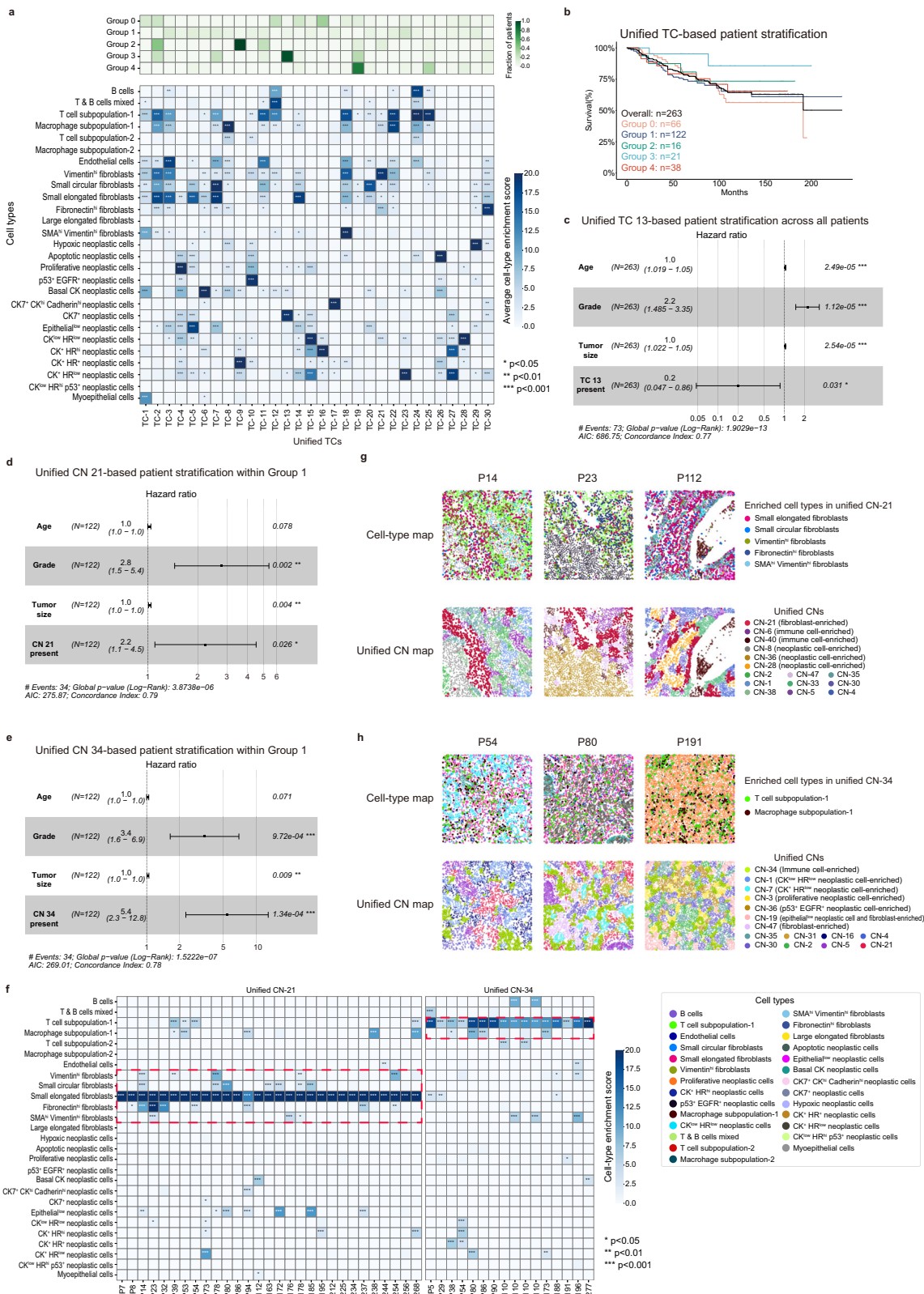

fine-grained tissue structure assignment matrix $S^{(1)} \in \mathbb{R}^{n \times c_1}$, which was formulated as below.

$$S^{(1)} = \text{Softmax}\left(\text{FC}_1\left(Z^{(1)}; \theta_{\text{FC}_1}\right)\right) \quad (3)$$

where each element of $S^{(1)}$ represents the probability of a cell (row) belonging to one of the $c_1$ fine-grained tissue structures

(column). $\theta_{\text{FC}_1}$ represents trainable parameters in the fully-connected neural network $\text{FC}_1$. The hyperparameter $c_1$ specifies the maximum number of fine-grained tissue structures to be detected and the model automatically determines the final number (less than or equal to $c_1$).

To further identify coarse-grained tissue structures, we applied a differentiable graph pooling layer[35,55] to generate a coarsened graph

**Fig. 6 | Hierarchical tissue structure-based stratification of breast cancer patients. a** Heatmaps showing the fraction of patients in each group containing each unified TC (top), and average cell-type enrichment scores for each unified TC (bottom). **b** Kaplan-Meier survival curves of 263 breast cancer patients, categorized into five groups based on unified TCs. **c** Forest plot of hazard ratios (HRs) and 95% confidence intervals (CIs) from multivariable Cox proportional hazards regression (adjusting for age, tumor grade, and tumor size) evaluating the association between unified TC-13 and overall survival ($N = 263$ patients, 73 events). Presence of unified TC-13 was significantly associated with improved survival (HR = 0.20, $P$-value = 0.031). **d** Forest plot of HRs and 95% CIs from multivariable Cox regression for unified CN-21 within Group 1 ($N = 122$ patients, 34 events). Presence of unified CN-21 was significantly associated with worse survival (HR = 2.2, $P$-value = 0.026). **e** Forest plot of HRs and 95% CIs from multivariable Cox regression for unified CN-34 within Group 1 ($N = 122$ patients, 34 events). Presence of unified CN-34 was significantly

associated with worse survival (HR = 5.4, $P$-value < 0.001). In (**c**–**e**), square markers indicate HR point estimates, and horizontal error bars represent the corresponding 95% CIs. **f** Heatmaps showing cell-type enrichment scores for unified CN-21 and CN-34 across patients. Significantly enriched cell types in these two CNs are highlighted by red dashed boxes. In both **a** and **f** enrichment score was defined as -$\log_{10}$(adjusted $P$-value). $P$-values were computed using hypergeometric tests for enrichment (over-representation) and adjusted with the Benjamini-Hochberg method. *, $P$-value < 0.05; **, $P$-value < 0.01; ***, $P$-value < 0.001. Exact adjusted $P$-values are provided in the Source Data file. **g** Representative single-cell spatial maps from patients containing unified CN-21. Cells are colored by cell types (top) and unified CNs (bottom). **h** Representative single-cell spatial maps from patients containing unified CN-34. Cells are colored by cell types (top) and unified CNs (bottom). Source data are provided as a Source Data file.

with $c_1$ pooled nodes, where each pooled node represents a fine-grained tissue structure. The adjacency matrix $A^{(2)} \in \mathbb{R}^{c_1 \times c_1}$ and node embedding matrix $X^{(2)} \in \mathbb{R}^{c_1 \times d}$ of the coarsened graph were formulated as follows.

$$A^{(2)} = S^{(1)T} A^{(1)} S^{(1)} \tag{4}$$

$$X^{(2)} = S^{(1)T} Z^{(1)} \tag{5}$$

The resulting coarsened graph is fully-connected with self-loops. Self-loops were removed and the adjacency matrix was normalized as

$$\hat{A} = A^{(2)} - I_{c_1} diag(A^{(2)}) \tag{6}$$

$$A^{pool} = \hat{D}^{-\frac{1}{2}} \hat{A} \hat{D}^{-\frac{1}{2}} \tag{7}$$

where $\hat{D}$ is the degree matrix of $\hat{A}$. Each entry in $A^{pool}$ represents the normalized strength of the connection between two fine-grained tissue structures.

To relieve the smoothing effect of message-passing operation, we introduced an adaptive edge-pruning threshold $T_{edge\_pruning}$. Only edges with weights in the adjacency matrix $A^{pool}$ greater than this threshold are retained and their weights are reset to 1.

$$\hat{A}_{ij}^{(2)} = \begin{cases} 1 & if\ A_{ij}^{pool} > T_{edge\_pruning} \\ 0 & otherwise \end{cases} \tag{8}$$

Since excessive or insufficient edge pruning may affect model performance (Supplementary Note 3; Supplementary Figs. 13b–15b), the adaptive threshold was defined as

$$T_{edge\_pruning} = \frac{1}{c_1 - 1} \tag{9}$$

This value corresponds to the case where all fine-grained tissue structures are connected with equal strength.

Finally, another graph convolution and fully-connected layers were applied to generate the soft coarse-grained tissue structure assignment matrix:

$$Z^{(2)} = ReLU\left(GNN_2\left(X^{(2)}, \hat{A}^{(2)}; \theta_{GNN_2}\right)\right) \tag{10}$$

$$S^{(2)} = Softmax\left(FC_2\left(Z^{(2)}; \theta_{FC_2}\right)\right) \tag{11}$$

Each element of $S^{(2)} \in \mathbb{R}^{c_1 \times c_2}$ represents the probability of a fine-grained structure (row) belonging to one of the $c_2$ coarse-grained structures (column). $Z^{(2)} \in \mathbb{R}^{c_1 \times d}$ is the updated embedding matrix of

pooled nodes in the coarsened graph. $\theta_{GNN_2}$ and $\theta_{FC_2}$ represent trainable parameters in the graph neural network $GNN_2$ and fully-connected neural network $FC_2$, respectively. The hyperparameter $c_2$ specifies the maximum number of coarse-grained tissue structures to be detected, and the model automatically determines the final number (less than or equal to $c_2$).

The loss function with regard to the soft hierarchical tissue structure assignment (base module) was defined as follows.

$$L_{Base} = \alpha \times L_{Fine} + (1 - \alpha) \times L_{Coarse} \tag{12}$$

where $\alpha$ is a weight parameter used for balancing the fine-grained loss $L_{Fine}$ and the coarse-grained loss $L_{Coarse}$. We further introduced an adaptive scheduling strategy for determining $\alpha$ to dynamically balance these two losses throughout training. Specifically, $\alpha$ was scheduled to decay from 0.9 to 0.1 during training, shifting focus from fine-grained to coarse-grained structure learning. Both $L_{Fine}$ and $L_{Coarse}$ use the graph MinCut-based formulation, optimizing the matrix $S^{(1)}$ and $S^{(2)}$, respectively.

$$L_{Fine} = -\frac{\sum_{j=1}^{c_1}\left(S^{(1)T} A^{(1)} S^{(1)}\right)_{jj}}{\sum_{j=1}^{c_1}\left(S^{(1)T} D^{(1)} S^{(1)}\right)_{jj}} + \left\|\frac{S^{(1)T} S^{(1)}}{\|S^{(1)T} S^{(1)}\|_F} - \frac{I_{c_1}}{\sqrt{c_1}}\right\|_F \tag{13}$$

$$L_{Coarse} = -\frac{\sum_{k=1}^{c_2}\left(S^{(2)T} A^{(2)} S^{(2)}\right)_{kk}}{\sum_{k=1}^{c_2}\left(S^{(2)T} D^{(2)} S^{(2)}\right)_{kk}} + \left\|\frac{S^{(2)T} S^{(2)}}{\|S^{(2)T} S^{(2)}\|_F} - \frac{I_{c_2}}{\sqrt{c_2}}\right\|_F \tag{14}$$

where $D^{(1)} \in \mathbb{R}^{n \times n}$ and $D^{(2)} \in \mathbb{R}^{c_1 \times c_1}$ are degree matrices derived from the undirected KNN graph and the coarsened graph, respectively. Both $L_{Fine}$ and $L_{Coarse}$ consist of two terms. The first term encourages clustering of strongly connected nodes, and the second term encourages orthogonal and balanced tissue structure assignments[35].

## Robust hierarchical tissue structure identification

To enhance the stability of HRCHY-CytoCommunity, we introduced a consistency and balance regularization module for identifying robust hierarchical tissue structures (Fig. 1b). Specifically, we adapted the GRAND framework[56] to the hierarchical tissue structure assignment task. The core idea involves stochastically generating multiple perturbed versions of the cell-cell proximity graph (i.e., KNN graph) by randomly dropping node features, and then enforcing cluster assignment consistency across these perturbed graphs. An entropy-based balance regularization was also used to prevent cluster collapse. This module improves both the robustness and accuracy of hierarchical tissue structure identification (Supplementary Note 3; Supplementary Figs. 13a–15a).

First, given the original KNN graph with adjacency matrix $A^{(1)}$ and node attribute matrix $X^{(1)}$, we randomly removed the entire attribute

vectors of a subset of nodes. This dropout strategy was referred to as DropNode. In contrast to standard dropout, which only masks individual elements of $X^{(1)}$, DropNode removes all features of selected nodes, thereby accounting for graph structural effects and explicitly reducing reliance on specific neighboring nodes. This generates more stochastic perturbations and has been shown to achieve better performance than standard dropout[56].

Formally, for each node $i$, a binary mask $\epsilon_i$ was sampled from a Bernoulli distribution $\epsilon_i \sim \text{Bernoulli}(1-\delta)$, where $\delta$ is a predefined DropNode rate. The perturbed feature vector was then computed as $\widetilde{\mathbf{x}_i} = \epsilon_i \cdot \mathbf{x_i}$, where $\mathbf{x_i}$ represents the original feature vector of node $i$. The perturbed feature matrix $\widetilde{X}$ was used only during training. During inference, features were rescaled by multiplying $(1-\delta)$ to match the expected activation magnitude of $\widetilde{X}$.

Next, the soft hierarchical tissue structure assignment module was performed on each perturbed cell-cell proximity graph. By repeating the DropNode procedure $M$ times, $M$ sets of predicted labels for both fine-grained CNs and coarse-grained TCs were obtained. To evaluate prediction consistency, we calculated the normalized assignment distribution center for each node:

$$\bar{\mathbf{s}_i} = \frac{1}{m}\sum_{m=1}^{M} \mathbf{s_i^{(m)}} \tag{15}$$

where $\mathbf{s_i^{(m)}}$ denotes the predicted tissue structure assignment distribution of node $i$ in the $m$-th perturbed graph.

Finally, the consistency regularization loss was defined as the average $L_2$ distance between the assignment distributions from each perturbed graph and the distribution center:

$$L_{Consis} = \frac{1}{M}\sum_{m=1}^{M}\sum_{i=1}^{n} \left\| \mathbf{s_i^{(m)}} - \bar{\mathbf{s}_i} \right\|_2^2 \tag{16}$$

This loss encourages prediction consistency under multiple stochastic perturbations, thereby improving the robustness of hierarchical tissue structure assignments produced by HRCHY-CytoCommunity.

However, consistency regularization alone may increase the risk of the collapse of the coarse-grained assignment layer (i.e., all nodes being assigned to a single cluster), trivially minimizing the consistency loss. To prevent such a collapse, we introduced an entropy-based balance regularization term:

$$L_{Balance} = 1 - \frac{1}{M}\sum_{m=1}^{M} \frac{\mathbf{H_m^{(2)}}}{\log c_2} \tag{17}$$

$$\mathbf{H_m^{(2)}} = -\sum_{k=1}^{c_2} s_{m,k}^{(2)} \log s_{m,k}^{(2)} \tag{18}$$

where $\mathbf{H_m^{(2)}}$ is the entropy of the cluster assignment distribution for the $m$-th perturbation, and $c_2$ is the number of coarse-grained TCs. This term encourages balanced cluster assignments by maximizing the entropy of the predicted distributions across clusters.

The overall loss function of HRCHY-CytoCommunity was then defined as

$$L_{Total} = L_{Base} + \lambda_1 \times L_{Consis} + \lambda_2 \times L_{Balance} \tag{19}$$

where $L_{Base}$ is the base loss from the soft hierarchical tissue structure assignment, $L_{Consis}$ is the consistency regularization loss, and $L_{Balance}$ is the entropy-based balance regularization loss. The hyperparameters $\lambda_1$ and $\lambda_2$ control the relative contributions of these terms. A comprehensive list of the hyperparameter settings used for all datasets was provided in Supplementary Table 1.

## Determination of optimal numbers of hierarchical tissue structures based on cluster stability

To identify hierarchical tissue structures using HRCHY-CytoCommunity, users are required to pre-specify the desired numbers of fine-grained CNs, denoted as $c_1$, and coarse-grained TCs, denoted as $c_2$. Inspired by the approach implemented in CellCharter, HRCHY-CytoCommunity employs a cluster stability-based procedure to determine the optimal values of the pair $(c_1, c_2)$. Specifically, the procedure begins by defining a search range for both $c_1$ and $c_2$, as well as the number of independent runs $R$. For each candidate $(c_1, c_2)$ pair, HRCHY-CytoCommunity performed $R$ runs. To evaluate the consistency of the resulting clusters (i.e., CNs and TCs) across runs, we computed the Fowlkes-Mallows Index (FMI)[57], which quantifies the similarity between two cluster assignments as follows:

$$FMI = \frac{TP}{\sqrt{(TP+FP)(TP+FN)}} \tag{20}$$

where TP, FP, and FN represent true positives, false positives, and false negatives, respectively, evaluated over pairs of cells assigned to the same or different clusters. The FMI ranges from 0 to 1, with higher values indicating greater similarity between cluster assignments.

In HRCHY-CytoCommunity, the FMI was computed separately for the identified CNs and TCs across all pairs of runs for a given $(c_1, c_2)$ configuration. This yielded a set of CN-level FMIs and a set of TC-level FMIs. For each level, the median FMI across all run pairs was computed to summarize the cluster stability at that level. The final stability score for a $(c_1, c_2)$ configuration was then defined as the average of the CN-level and TC-level median FMIs. The optimal $(c_1, c_2)$ pair was selected as the one that yields the highest stability score. Among the $R$ models associated with the optimal $(c_1, c_2)$ configuration, the model achieving the lowest training loss was chosen to output the final hierarchical tissue structures.

In this study, for datasets where the ground-truth number of CNs or TCs is known, we fixed the known value and applied the cluster stability-based procedure to determine the optimal number of structures at the other hierarchical level. For multi-sample datasets, the optimal $(c_1, c_2)$ pair was selected based on the highest average cluster stability score across all samples.

## Unified hierarchical tissue structure generation across samples

To facilitate integrated analysis of multiple tissue samples, we developed a clustering-based module to align hierarchical tissue structures across different samples (Supplementary Fig. 1). First, cell-type enrichment scores were computed for each tissue structure (including both coarse-grained and fine-grained structures) within every sample using the hypergeometric test. Then, we horizontally concatenated the cell-type enrichment score matrices from all samples to construct a composite feature matrix. This composite matrix was then input into a clustering algorithm to group tissue structures from different samples into a unified set of hierarchical tissue structures.

For the human TNBC MIBI-TOF dataset and the human breast cancer IMC dataset, we applied hierarchical clustering with Ward's method. Euclidean distance was used to measure the similarity between patients' tissue structure profiles.

## Running of published methods

We compared the performance of HRCHY-CytoCommunity with seven established tissue structure identification methods, including six widely-used non-hierarchical methods, GraphST[15], NicheCompass[20], CellCharter[21], SpaGCN[22], Giotto Suite[24], and SpaSEG[23], as well as NeST[34], a state-of-the-art method specifically designed for identifying nested hierarchical structures in spatial transcriptomics data. To ensure a fair and consistent comparison, the number of TCs and CNs to be detected

by each method was set to match the ground-truth annotations provided in the original studies.

The Python code base for NeST was downloaded from https://github.com/bwalker1/NeST. This code base was applied to seven datasets (Supplementary Data 1). In accordance with NeST requirements, we utilized protein or mRNA expression data and cell spatial coordinates as inputs. For benchmarking purposes, we considered the first layer of the NeST output as coarse-grained TCs and the last layer as the fine-grained CNs. In cases where a cell population was assigned to multiple tissue structures within the same layer simultaneously, cells were allocated to the tissue structure with the purpose of obtaining the highest accuracy. For imaging-based datasets with limited numbers of features, the model hyperparameters were adjusted to ensure the identification of multiple single-gene hotspots and hierarchical structures. Notably, NeST does not require pre-specification of the number of tissue structures but demands the configuration of eight other hyperparameters. Detailed settings were listed in Supplementary Table 2.

The Python package GraphST (v.1.1.1) was applied to three datasets (Supplementary Data 1). For all three datasets, default hyperparameters were used for model training, following the official tutorial at https://deepst-tutorials.readthedocs.io/en/latest/. Notably, for the mouse spleen CODEX dataset, all 30 protein markers were used as highly variable genes (HVGs) due to the limited features. Spatial clustering was subsequently performed using the mclust package in R.

The Python package NicheCompass (v.0.3.0) was applied to four datasets (Supplementary Data 1). For the mouse spleen CODEX dataset, we followed the single-sample tutorial at https://nichecompass.readthedocs.io/en/latest/tutorials/notebooks/mouse_cns_single_sample.html for better performance. For the other three datasets, we followed the sample-integration tutorial at https://nichecompass.readthedocs.io/en/latest/tutorials/notebooks/mouse_cns_sample_integration.html. Due to the large numbers of cells in the mouse ICH Stereo-seq dataset and the human CRC Visium HD dataset, edge_batch_size was set to 1024. For the mouse spleen CODEX dataset, the scanpy.pp.neighbors and scanpy.tl.louvain functions were used with resolution = 0.005, 0.002, and 0.005 to identify coarse-grained TCs and 0.055, 0.035, and 0.045 to identify fine-grained CNs for samples BALBc-1, BALBc-2, and BALBc-3, respectively. For the mouse hypothalamic preoptic region MERFISH dataset, the parameter resolution was set to 0.02 and 0.6 for all five samples to identify coarse-grained TCs and fine-grained CNs, respectively. For the mouse ICH Stereo-seq dataset, the parameter resolution was set to 0.05 and 0.8 for all eight samples to identify coarse-grained TCs and fine-grained CNs, respectively. For the human CRC Visium HD dataset, the parameter resolution was set to 0.009 and 0.1 for all three samples to identify coarse-grained TCs and fine-grained CNs, respectively.

The Python package CellCharter (v.0.3.5) was applied to four datasets (Supplementary Data 1). Following the official tutorial at https://cellcharter.readthedocs.io/en/latest/index.html, scArches[58] was used for dimension reduction and integration for the mouse spleen CODEX dataset. For other spatial transcriptomics datasets, scVI[59] was used for dimension reduction and integration. Default hyperparameters were used to retrieve aggregated cell embeddings. The gmm.predict function was used to identify TCs and CNs.

The Python package SpaGCN (v.1.2.7) was applied to three datasets (Supplementary Data 1). All parameters were set according to the official tutorial at https://github.com/jianhuupenn/SpaGCN/blob/master/tutorial/tutorial.ipynb, except that max_epochs was increased to 200 for training convergence. Notably, for the mouse spleen CODEX dataset, the number of principal components was set to 30 due to limited numbers of features.

The Python implementation of SpaSEG was obtained from https://github.com/y-bai/SpaSEG. SpaSEG was applied to two datasets (Supplementary Data 1). All parameters were set following the official tutorial at https://github.com/y-bai/SpaSEG/blob/main/DLPFC_151673_demo.ipynb. During data preprocessing, the appropriate platforms were selected according to the datasets' platforms.

The R package Giotto Suite (v4.2.1) was applied to the MERFISH dataset following the two-stage workflow recommended in the official tutorial. First, genes with significant spatial patterns were identified using the spatial co-expression module detection method at https://giottosuite.com/articles/spatial_coexpression_modules.html. Then the identified genes were used as features for spatial clustering with the HMRF (Hidden Markov Random Field) model[12](https://giottosuite.com/articles/hmrf.html).

## Cell-type deconvolution using RCTD

Cell-type composition of each spot in the mouse hippocampus Slide-seq V2 dataset and the human breast cancer Visium V1 dataset was inferred using the RCTD[41] algorithm implemented in the R package spacexr (v2.2.0). Following the official tutorial (https://github.com/dmcable/spacexr), we set CELL_MIN_INSTANCE = 1 and UMI_min = 1 in the create.RCTD function for both datasets. In the run.RCTD function, we set doublet_mode = multi for the Visium V1 dataset, and set doublet_mode = doublet for the Slide-seq V2 dataset, based on the different spatial resolutions of these two technologies. For each dataset, single-cell RNA-seq data from matched tissue types were used as a reference to train the RCTD model (Data Availability). The resulting cell-type proportion matrices were used as the input node attributes in HRCHY-CytoCommunity.

## Quantitative performance evaluation using CODEX, MERFISH, Stereo-seq, and Visium HD datasets

For the mouse spleen CODEX dataset, the ground-truth assignments of cells to fine-grained CNs, including red pulp, marginal zone, B-cell zone, and periarteriolar lymphoid sheath (PALS), were derived from the original study[36]. For the mouse hypothalamic preoptic region MERFISH dataset, the outlines of hypothalamic nuclei regions were obtained from the original study[7], and the ground-truth assignments of cells to CNs were manually annotated based on these outlines. For the mouse ICH Stereo-seq dataset, the CN annotations were obtained from the original study[60]. For the human CRC Visium HD dataset, the TC annotations were obtained from the original study[3]. We quantitatively evaluated the performance of HRCHY-CytoCommunity and benchmarked methods using two evaluation metrics: Macro-F1 score and Adjusted Mutual Information (AMI) with calculations performed using the Python package scikit-learn (v1.1.2). The formulas for these evaluation metrics are as follows.

$$F1\,score = \frac{2 \times (Precision \times Recall)}{Precision + Recall} \tag{21}$$

$$Precision = \frac{TP}{TP + FP} \tag{22}$$

$$Recall = \frac{TP}{TP + FN} \tag{23}$$

$$AMI = \frac{I(GT; TS) - E\{I(GT; TS)\}}{\frac{1}{2}[H(GT) + H(TS)] - E\{I(GT; TS)\}} \tag{24}$$

where true positives (TP), true negatives (TN), false positives (FP) and false negatives (FN) represent the comparisons between the predicted tissue structures and the ground-truth cell assignments. $E\{I(GT; TS)\}$ denotes the expected mutual information between the ground-truth assignments of cells (GT) and the tissue structure predictions (TS). $H(GT)$ and $H(TS)$ represent the entropies of the ground-truth assignments of cells and the tissue structure predictions, respectively. The Macro-F1 score was calculated as the average of the F1 scores across all

types of ground-truth tissue structures within the dataset. The AMI is a metric based on Shannon information theory[61] that assesses the agreement between the predicted tissue structures and the ground-truth assignments, adjusting for the possibility of random agreement.

## Cell-type enrichment scores of tissue structures

To perform a quantitative assessment of cell-type composition within tissue structures, we defined an enrichment score for each cell type in each tissue structure as $-\log_{10}(P$-value). For a single-cell spatial omics sample, the $P$-value was determined through a hypergeometric test, taking into account four key parameters: (1) the number of cells of the given type within the tissue structure, (2) the total number of cells within the tissue structure, (3) the number of cells of that type in the sample, and (4) the total number of cells in the sample.

For a low-resolution (spot-based) spatial transcriptomics sample (e.g., Visium V1), where each spot contains a mixture of cell types, enrichment analysis was performed using cell-type deconvolution results. Specifically, for a cell type in an identified tissue structure, we compared the cell-type fractions across all spots within the structure against those from all other spots in the sample using a one-sided Wilcoxon rank-sum test to determine whether this cell type was significantly enriched within the identified structure. To account for multiple comparisons, the resulting $P$-values were adjusted with the Benjamini-Hochberg method[62].

## Survival analysis

To validate the clinical relevance of the hierarchical tissue structures identified by HRCHY-CytoCommunity, we performed survival analysis using unified TCs or unified CNs as features in combination with clinical survival data from the breast cancer IMC dataset[9]. All Kaplan-Meier survival curves were generated using the R package survival (v3.6-4). To evaluate the independent prognostic value of TC- or CN-derived features, we employed multivariable Cox proportional hazard regression models, adjusting for clinical covariates including patient age, tumor grade, and tumor size.

## Reporting summary

Further information on research design is available in the Nature Portfolio Reporting Summary linked to this article.

## Data availability

All data used in this study are publicly available. This study used the following eight publicly available spatial omics datasets, including a mouse spleen CODEX dataset (https://data.mendeley.com/datasets/zjnpwh8m5b/1), a mouse hypothalamic preoptic region MERFISH dataset (https://datadryad.org/stash/dataset/doi:10.5061/dryad.8t8s248), a human triple-negative breast cancer MIBI-TOF dataset (https://mibi-share.ionpath.com), a human breast cancer IMC dataset (https://zenodo.org/record/3518284#.Y2UQ0-xBybg), a human CRC Visium HD dataset (https://www.10xgenomics.com/platforms/visium/product-family/dataset-human-crc), a human breast cancer Visium V1 dataset (https://www.10xgenomics.com/datasets/human-breast-cancer-block-a-section-1-1-standard-1-1-0), a mouse hippocampus Slide-seq V2 dataset (https://singlecell.broadinstitute.org/single_cell/study/SCP815/highly-sensitive-spatial-transcriptomics-at-near-cellular-resolution-with-slide-seqv2), and a mouse ICH Stereo-seq dataset (https://db.cngb.org/stomics/stmich/). Two single-cell transcriptomic datasets were used as references for cell-type deconvolution in the spot-based spatial transcriptomics data, including a human breast cancer scRNA-seq dataset (GEO: GSE176078; https://www.ncbi.nlm.nih.gov/geo/query/acc.cgi?acc=GSE176078), and a mouse hippocampus scRNA-seq dataset (https://singlecell.broadinstitute.org/single_cell/study/SCP948/robust-decomposition-of-cell-type-mixtures-in-spatial-transcriptomics#study-download). No new data were generated in this study. Source data are provided with this paper.

## Code availability

HRCHY-CytoCommunity is an open-access Python package available in the GitHub repository at https://github.com/huBioinfo/HRCHY-CytoCommunity, under the MIT license. The specific version of the code associated with this publication is archived in Zenodo and is accessible via https://zenodo.org/records/18137898[63].

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

## Acknowledgements

This work was supported by a National Natural Science Foundation of China (NSFC) grant No. 62422211 and a Scientific Research Innovation Capability Support Project for Young Faculty No. SRICSPYF-ZY2025003 to Y. H., NSFC grants no. 62550005, 62132015, and No. U22A2037 to L. G., and an NSFC grant no. 62302386 and a National Key Research and Development Program of China No. 2024YFC2707102 to J. Lin. We thank the Key Laboratory of Computational Bioinformatics of Xi'an at Xidian University for providing computing support.

## Author contributions

Y. H. and L. G. conceived and designed the study. R. X., Z. W., and Y. H. designed the HRCHY-CytoCommunity algorithm. R. X. and Z. W. implemented the HRCHY-CytoCommunity algorithm. J. Liu, H. X., and Y. X. provided additional input during the method development. R. X., Z. W., J. Liu, and Y. H. performed the data analysis. R. X. and Z. W. provided support for the software package development. Y. H., L. G., and J. Lin supervised the study. R. X., Y. H., Z. W., J. Lin, and L. G. wrote the manuscript.

## Competing interests

The authors declare no competing interests.
