## [Transparent Peer Review file · Nature Communications]

HRCHY-CytoCommunity identifies hierarchical tissue organization in cell-type spatial maps

Corresponding Author: Professor Lin Gao

Version 0:

Reviewer comments:

Reviewer #1

(Remarks to the Author)

HRCHY-CytoCommunity presents a novel graph neural network (GNN)-based framework for identifying hierarchical multicellular structures in spatial omics data. Leveraging differentiable graph pooling and graph pruning, the method robustly uncovers nested tissue compartments and cellular neighborhoods from cell-type spatial maps.

Major advances of this approach:

- 1) HRCHY-CytoCommunity outperforms the state-of-the-art NeST method across multiple datasets (mouse spleen CODEX, hypothalamic MERFISH, TNBC MIBI-TOF). It accurately identifies spatially discontinuous structures and achieves higher accuracy, adjusted mutual information, and Macro-F1 scores.
- 2) The hierarchical majority voting strategy enhances reproducibility while preserving nested relationships between TCs and CNs. Unlike NeST, it assigns all cells to structures, avoiding coverage gaps.
- 3) A cell-type enrichment-based clustering module aligns structures across samples, revealing conserved and patient-specific biological features (e.g., 26 unified CNs in TNBC).
- 4) The method reveals functional hierarchies, such as neuron/non-neuron interactions in the hypothalamus and immune-tumor compartmentalization in TNBC, aligning with known biology. Applied to breast cancer, HRCHY-CytoCommunity enables multi-scale prognostic stratification.

Limitations:

- 1) Researchers need to pre-define/select structure counts/levels. Are there any quantitative indexes to guide the hyperparameter settings?
- 2) Can this model be adapted to RNA-seq based spot-structure data, such as Visium HD & StereoSeq?

Overall, this work is a significant advance in spatial omics analysis with biological and clinical insights.

(Remarks on code availability)

Reviewer #2

(Remarks to the Author)

In this study, Wang et al. introduced a spatial clustering framework HRCHY-CytoCommunity. Unlike conventional spatial clustering methods such as SpaGCN or graphST that deliver a single level spatial domain clustering, the proposed method can present hierarchical spatial clustering, detecting nested spatial domains and structures that can not be detected with the conventional methods. Considering the complexity of the tissue microenvironment in various complex diseases such as cancer, the capability to detect hierarchical spatial clustering can help us better understand the spatial patterns underlying various biological conditions and diseases, therefore it surely will be of interest of an ever-growing community of single-cell spatial genomics. Although I am fully convinced the importance of the problem and appreciate the authors effort in

addressing this issue, I found that the authors may need to enhance the benchmarking (current only one single method was compared and benchmarked) and add a ablation study (to showcase the necessity of the all modules that they utilized). Beyond showing the effectiveness of the method and its superior performance in comparison with other method, the authors may also want to show more evidence to justify why the proposed touchup major voting strategy (NOT a integral part of the model and thus can't be optimized all together) is solid and won't damage the performance dramatically compared to other integrative alternative strategy (see major comment #2)

Major concerns

(1) A potential limitation of the HRCHY-CytoCommunity method lies in its reliance on user-specified hyperparameters c_1 and c_2 , which define the number of fine-grained cellular neighborhoods (CNs) and coarse-grained tissue compartments (TCs), respectively. While the authors acknowledge that the model can learn fewer clusters than specified if overestimated, there is no principled strategy provided for selecting appropriate values of c_1 and c_2 in practice. This imposes a significant constraint on the method's generalizability, particularly in real-world settings where prior knowledge about hierarchical tissue architecture is limited or absent. Without an automated or data-driven approach to determine optimal resolution, users must rely on ad hoc or domain-specific heuristics, which may lead to suboptimal or biologically misleading interpretations. A discussion on how to systematically tune or infer these hyperparameter, ideally integrated to be part of the model, would substantially improve the applicability and robustness of the framework.

(2) The hierarchical majority voting strategy used in HRCHY-CytoCommunity, while helpful for improving stability and enforcing a nested structure, is applied post hoc and is not integrated into the model's training objective. This disconnection introduces two key issues. First, the learned soft assignments are not explicitly optimized to yield stable or hierarchically consistent outputs, which may lead to mismatches between training-time objectives and final outputs. Second, because the voting procedure is non-differentiable, it prevents end-to-end training and may obscure the interpretation of how hierarchical structures are learned. More critically, this touch-up procedure could obscure systematic instability in the underlying GNN assignments, which would not be evident without the ensemble step. As a result, the biological or clinical significance of the identified structures may hinge on a potentially brittle post-processing routine.

As a practical alternative, the authors could consider incorporating a consistency regularization strategy, which encourages alignment of cluster assignments across stochastic runs during training, or a differentiable nesting regularization term, which softly penalizes misalignment between fine- and coarse-grained structures. These approaches would enable the model to learn hierarchical relationships more robustly and transparently during optimization. One quick strategy is to introduce consistency regularization during training. This involves running the model multiple times with stochastic perturbations (e.g., dropout or graph augmentation) and penalizing discrepancies in the predicted cluster assignments across runs. By minimizing divergence (e.g., KL divergence) between outputs, the model is encouraged to produce stable and robust hierarchical assignments without relying on post hoc voting.

At a minimum, the authors should implement and compare one of these integrative alternatives against the current ensemble-based (touchup major voting based) strategy to demonstrate that the model's performance and robustness are not overly dependent on post hoc correction. Such a comparison would strengthen the claims of generalizability and biological interpretability. If the performances are indeed similar (even slightly worse), then the authors may not need to update all the results with the new model structure, which is quite intensive.

(3) The developed method integrates multiple components, including a hierarchical GNN with differentiable pooling, a post hoc majority-voting ensemble for robustness, and a clustering-based module for cross-sample alignment. However, the manuscript does not provide an ablation study to evaluate the individual contribution of these modules to overall performance. In particular, it remains unclear how much of the reported improvement over baseline methods arises from the GNN-based hierarchical learning itself, versus the ensemble voting or cross-sample unification steps. An ablation study removing or modifying each module, such as training without voting, disabling edge pruning, or skipping the alignment step, would be valuable for confirming that each component is essential and for understanding where the gains in performance and robustness originate.

(4) The current benchmarking is restricted to a single method, NeST, which, while hierarchical, is designed for high-dimensional spatial transcriptomics and does not generalize well to low-dimensional or cell-type-based spatial data such as spatial proteomics. This narrow comparison limits the ability to assess the broader utility and competitiveness of HRCHY-CytoCommunity. In particular, several more recent and widely used frameworks—such as SpaSEG, Giotto, and GraphST—are highly relevant comparators. Although these methods do not explicitly model hierarchical structures, they can be run at multiple resolutions or applied recursively to simulate hierarchical segmentation. Benchmarking against such methods would provide a more comprehensive evaluation of whether HRCHY-CytoCommunity's explicit hierarchical modeling offers practical advantages in terms of spatial coherence, robustness, and biological relevance. The authors are strongly encouraged to include comparisons to these additional methods or, at a minimum, to discuss their relative performance.

(5) The reliance on cell-type annotations as input features introduces a dependence on upstream preprocessing quality, which is not evaluated. The authors should assess how robust their method is to label inaccuracies or varying annotation resolutions, or clarify the assumptions about cell-type consistency across samples.

(6) Given the multi-step pipeline and ensemble-based framework, the scalability of HRCHY-CytoCommunity in terms of runtime and memory is not discussed. The authors should provide runtime/memory profiling or clarify the computational requirements to facilitate practical adoption.

Minor comments

(1) HRCHY-CytoCommunity models a fixed two-level hierarchy. This assumption may limit applicability to tissues with deeper or non-uniform hierarchical organization. The authors should discuss whether and how the framework can be

extended to more flexible or adaptive hierarchical depths.

(2) The interpretability of the discovered tissue structures is not systematically evaluated beyond cell-type enrichment. It would be valuable to assess how consistently the model discovers known anatomical or functional regions across samples and tissues, and how uninterpretable clusters are handled.

(3) a 2-levels hierarchical structure", ⇒ "a 2-level hierarchical structure"

(4) Use of inconsistent capitalization for "Cellular Neighborhoods" and "Tissue Compartments" (sometimes capitalized, sometimes not). Maybe the authors can pick one consistent style (e.g., lowercase unless at the start of a sentence).

(5) Line 179, "the coarse-grained red pulp compartment could be further divided into two fine-grained substructures, red pulp and marginal zone". This phrasing is somewhat confusing, as it suggests that "red pulp" is both a coarse-grained and a fine-grained region, and it categorizes the marginal zone, which is typically considered a transitional or fine-grained region between red and white pulp, as a substructure of red pulp. This may need clarification or correction, especially for readers with histological expertise.

(Remarks on code availability)

The github repository generally looks good to me.

The provided GitHub repository includes a basic example dataset (mouse brain MERFISH) and a structured walkthrough consisting of four steps: graph construction, soft clustering, ensemble voting, and visualization. While this demonstrates the core workflow, there are several issues. The example dataset is limited in scope, benchmark datasets from the paper (e.g., spleen or TNBC) are not provided. Also, the documentation lacks clarity on hyperparameters, and the scripts have limited error handling.

Reviewer #3

(Remarks to the Author)

The manuscript reviewed introduces HRCHY-CytoCommunity, a method for identifying hierarchical tissue structures using only spatial graphs and cell-type annotations. The study presents a method to perform hierarchical structure identification using graph neural networks with pruning and pooling. The study exemplified this strategy in multiple datasets, highlighting the identification of discontinuous spatial patterns and presenting an improved performance in datasets with few expression features (e.g., protein-based data).

While these are promising goals, the manuscript relies heavily on the assumption that hierarchical organization in tissues can be fully captured using cell-type annotations alone. This may be an oversimplification of complex tissues where cellular behavior is driven by signaling gradients, cellular interactions, or dynamic gene programs. We believe this together with others exposed below, limit the current version of the manuscript from being suitable for publication. In our opinion, only a deeply revised version of the manuscript, with clarification of the following concerns should be considered for publication.

Major Comments

1. Assumption of Hierarchy: The method assumes that tissue structures are organized hierarchically, which is not universally true. While this may apply to some biological systems (e.g., lymphoid tissues, brain), in others, spatial patterns are more continuous or transitional. The manuscript should explicitly discuss the biological validity of this assumption and its limitations.

2. Evaluation on More Diverse Datasets: The manuscript focuses primarily on datasets with well-separated cell types and pre-annotated structures. To evaluate the robustness of HRCHY-CytoCommunity, we recommend including sequencing-based spatial transcriptomics datasets—especially those from the NeST study or similar benchmarks—to compare with methods like NeST, which rely on transcriptome-wide data.

3. Limitations of Using Cell-Type Annotation Only: The exclusive reliance on cell-type labels poses a major limitation. High-quality annotations may not always be available, and annotation resolution varies by dataset. Furthermore, critical spatial structures may be defined not by cell types but by cell-cell interactions, gradients, or activity states. These limitations should be acknowledged, and potential solutions or future directions discussed. For instance, the author could compare the results of two different levels of annotation (finer vs coarse)..

4. Comparison with Additional Methods: The current comparisons omit several state-of-the-art spatial domain identification tools, including SpaGCN, NicheCompass, CellCharter, Gaston-Mix among others. While these do not explicitly identify hierarchical domains, many can be run iteratively or parameter-tuned to extract structures at different scales. Including such comparisons would provide a fairer performance benchmark.

5. Loss Threshold Parameter (β): The parameter β is introduced as a threshold for selecting structures based on graph loss but is not reflected in the mathematical formulations (Equations 7–9). The manuscript should be updated to incorporate β directly into the equations, with a formal explanation of its role and its effect on structure selection.

6. Details of Model Parameters: Please add a table summarizing the hyperparameters used for each dataset: number of GNN layers, MLP layers, values for α , β , and pruning thresholds. Without this information, reproducibility and understanding of model performance across datasets are limited.

7. Sensitivity Analysis: The study would benefit from conducting and reporting a sensitivity analysis for α , β , and edge pruning on at least one image-based and one sequencing-based dataset. This is critical to demonstrate model stability and the impact of key hyperparameters on performance.

8. Choice and Estimation of Number of Structures (c_1 , c_2): The model requires the number of fine- and coarse-grained structures to be pre-defined. The manuscript should clarify whether these are learned or fixed, and how users should choose them. If large values are set, how does the model handle oversegmentation? A discussion or heuristic strategy would be helpful.

9. Interpretability of Nested Structures: The manuscript suggests that assigning each fine-grained structure to a single coarse-grained one improves interpretability. However, biological systems often contain overlapping or ambiguous structures. For example, cells in a boundary region may legitimately belong to multiple domains. Instead of enforcing strict assignments, the model might benefit from allowing flexible memberships or highlighting ambiguity.

10. Survival Analysis Details: The survival analysis lacks details on the inclusion of clinical covariates. Without covariate adjustment, survival differences may be confounded. If covariates (e.g., age, sex, tumor grade) were not included, either re-run the analysis with appropriate adjustments or justify why this analysis is still valid.

11. Validation with Simulated Data: Consider including a synthetic dataset with known hierarchical ground truth to benchmark model accuracy and interpretability under controlled conditions.

12. Multiple Sample Integration: The method currently runs HRCHY-CytoCommunity independently per sample, followed by clustering enrichment profiles. This strategy might not fully capture shared spatial structure across samples. Consider discussing or exploring alternatives, such as joint embedding approaches or graph alignment techniques.

Minor Comments

1. Please add a legend to Figure 3a to clarify the visualized elements and improve readability.

2. The rationale for selecting the "3TC" annotation in the bregma mouse brain sample is not provided. Please justify its use or clarify its biological relevance.

3. For the brain example, it would be informative to overlay the results with reference annotations from the Allen Brain Atlas to validate spatial domains at both coarse and fine levels.

4. In some examples, fine-grained domains appear more as transitional gradients than discrete clusters. This challenges the assumption of discrete, nested hierarchies and should be discussed.

5. While this method builds on Cytocommunity, the distinction lies primarily in the added CNN for hierarchical domain inference. Please clarify what conceptual or performance advancements are made beyond the original Cytocommunity approach.

6. The claim that NeST is not applicable to low-feature datasets needs further explanation. Since NeST uses gene expression, it may still be applicable depending on the resolution. If excluded from comparisons, the manuscript should clearly justify why.

7. The use of manual annotations in unorganized organs (e.g., lung, kidney) is potentially problematic, as these organs lack strong spatial hierarchy. Please explain the rationale behind these choices and whether such annotations are used as training or just evaluation.

(Remarks on code availability)

The repository provided is well detailed and includes all information needed to reproduce the analysis presented and use the tool.

Reviewer #4

(Remarks to the Author)

(Remarks on code availability)

Version 1:

Reviewer comments:

Reviewer #1

(Remarks to the Author)

Identifying multi-scale tissue structures from ST data is crucial for downstream biological discovery. This work provides a novel method to identify hierarchical structures.

This submission has made substantial revisions and addressed all my proposed questions. I recommend acceptance of this work.

(Remarks on code availability)

Reviewer #2

(Remarks to the Author)

I am satisfied with the authors' response to my review comments. The revisions are detailed and directly address all major concerns raised in the initial review. Specifically, the authors have substantially strengthened the work by expanding the

benchmarking, adding comprehensive ablation and robustness analyses, and most importantly replacing the post hoc majority-voting strategy with a principled, end-to-end regularization framework. These changes not only improve the technical rigor of the method but also enhance its transparency and practical relevance. I have no further concerns

(Remarks on code availability)
did it last round

Reviewer #3

(Remarks to the Author)

The authors have substantially improved the manuscript in comparison to the first version. We are satisfied with the answers given to our concerns and with the modifications done in text and figures. We would like to highlight the level of detail provided in the answer's to reviewer's document, which better contextualize the benefits and limitations of this method in comparison to previous ones. We recommend the acceptance of the study.

(Remarks on code availability)

Code provided is sufficient to use the tool and reproduce the analysis.

Reviewer #4

(Remarks to the Author)

(Remarks on code availability)

Response to reviewers

We would like to sincerely thank the editor and the reviewers for their constructive comments and insightful suggestions, which have greatly helped us improve both the clarity and quality of our manuscript. In summary, we have addressed the following five aspects, which are described in detail in the point-by-point responses to the reviewer comments below.

1. Model modification and performance improvement

- We replaced the *post hoc* majority-voting mechanism in the original version of HRCHY-CytoCommunity with the consistency and balance regularization terms that are directly incorporated into the model's optimization objective. These modifications improved both the accuracy and stability of hierarchical tissue structure identification and allowed fully end-to-end differentiable training (see **Methods** subsection "Robust hierarchical tissue structure identification" and **Fig. 1b**).
- We introduced a cluster stability-based heuristic strategy to guide the determination of optimal numbers of hierarchical tissue structures, thereby enhancing the method's practicality (see **Methods** subsection "Determination of optimal numbers of hierarchical tissue structures based on cluster stability").

2. Expanded evaluation and additional benchmarking

- As suggested by Reviewers 2 and 3, we expanded the benchmarking to include six additional state-of-the-art non-hierarchical tissue structure identification methods (SpaGCN, SpaSEG, Giotto Suite, GraphST, NicheCompass, and CellCharter), besides NeST. The results demonstrated that HRCHY-CytoCommunity achieved consistently superior or competitive performance in identifying hierarchical tissue structures (see **Figs. 2-4, Supplementary Figs. 2-6, and Supplementary Notes 1 and 2**).
- We further applied HRCHY-CytoCommunity to four additional sequencing-based spatial transcriptomics datasets spanning diverse resolutions and technologies, confirming its broad applicability across imaging- and sequencing-based platforms (see **Figs. 2e-h, Fig. 4, Supplementary Figs. 3, 5-8 and Supplementary Notes 1 and 2**).

3. Comprehensive ablation, sensitivity, robustness, and scalability analyses

- We added comprehensive ablation and sensitivity analyses to evaluate the contribution of each model component (i.e., consistency and balance regularization, adaptive edge pruning, and adaptive α scheduling) and sensitivity of each key hyperparameter (i.e., the DropNode rate, λ_{consis} , λ_{balance} , the number of neighbors K in the KNN graph (cell-cell proximity graph) construction, and the number of perturbations used for generating perturbed cell-cell proximity graphs) (see **Supplementary Figs. 13-18 and Supplementary Notes 3 and 4**).

- We conducted robustness assessments against cell-type label inaccuracies and annotation resolution variations, demonstrating that HRCHY-CytoCommunity remains stable and robust under noisy or heterogeneous input annotations (see **Supplementary Figs. 19 and 20**, and **Supplementary Note 5**).
- We further performed scalability analysis, showing that HRCHY-CytoCommunity is highly efficient in both runtime and memory usage. On a large-scale 473k-spot dataset, the full model of HRCHY-CytoCommunity completed hierarchical tissue structures identification in 8.02 minutes, requiring only 8.69 GB of memory, confirming its strong scalability to large-scale spatial omics datasets (see **Supplementary Figs. 21 and 22**, and **Supplementary Note 6**).

4. Enhancement in survival analysis and biological interpretation

- All survival analyses were revised to use multivariable Cox proportional hazards models instead of log-rank tests, allowing adjustment for clinical covariates and providing more reliable effect estimates. (see **Methods** subsection “Survival analysis” and **Figs. 6c-e**).
- We expanded applications of HRCHY-CytoCommunity across multiple tissues (e.g., brain, colorectal cancer, and breast cancer) and provided clear biological interpretations of the identified coarse- and fine-grained structures based on their histological contexts.

5. Clarifications, textual revisions, and presentation improvements

- We clarified the methodological scope of the manuscript, refined figure legends and descriptions for improved readability, and ensured that all key parameters and equations (e.g., the reformulated loss functions) are explicitly defined.

Overall, besides modifications and expansions of existing analyses and original figures, we have added a new main figure (Figure 4), 20 new supplementary figures, nine new supplementary notes, and four new supplementary tables. With these additional analyses motivated by the editor’s and reviewers’ insightful comments, we believe that the manuscript has been substantially strengthened.

- **Figure 4.** Evaluation of HRCHY-CytoCommunity’s cross-platform generalization using the mouse hippocampus Slide-seq V2 and human breast cancer Visium V1 datasets.
- **Supplementary Figure 3.** Spatial maps of fine-grained CNs in the human CRC Visium HD dataset.
- **Supplementary Figure 4.** Spatial maps of coarse-grained TCs in the mouse hypothalamic preoptic region MERFISH dataset.
- **Supplementary Figure 5.** Performance evaluation of HRCHY-CytoCommunity using the mouse ICH Stereo-seq dataset.
- **Supplementary Figure 6.** Spatial maps of coarse-grained TCs in the mouse ICH Stereo-seq dataset.

- **Supplementary Figure 7.** Cell-type map and cluster stability across different parameter settings on the mouse hippocampus Slide-seq V2 dataset.
- **Supplementary Figure 8.** Cell-type map and cluster stability across different parameter settings on the human breast cancer Visium V1 dataset.
- **Supplementary Figure 11.** Complementary results of survival analysis for the breast cancer IMC dataset.
- **Supplementary Figure 12.** Average cell-type enrichment scores of unified CNs in the breast cancer IMC dataset.
- **Supplementary Figure 13.** Ablation study of key components in HRCHY-CytoCommunity using the simulated dataset.
- **Supplementary Figure 14.** Ablation study of key components in HRCHY-CytoCommunity on coarse-grained TC identification using real spatial omics datasets.
- **Supplementary Figure 15.** Ablation study of key components in HRCHY-CytoCommunity on fine-grained CN identification using real spatial omics datasets.
- **Supplementary Figure 16.** Sensitivity analysis of hyperparameters in HRCHY-CytoCommunity using the simulated dataset.
- **Supplementary Figure 17.** Sensitivity analysis of hyperparameters in HRCHY-CytoCommunity on coarse-grained TC identification using real spatial omics datasets.
- **Supplementary Figure 18.** Sensitivity analysis of hyperparameters in HRCHY-CytoCommunity on fine-grained CN identification using real spatial omics datasets.
- **Supplementary Figure 19.** Robustness assessment of HRCHY-CytoCommunity to label inaccuracies and annotation resolution variations using the mouse hypothalamic preoptic region MERFISH dataset.
- **Supplementary Figure 20.** Robustness assessment of HRCHY-CytoCommunity to label inaccuracies and annotation resolution variations using the human TNBC MIBI-TOF dataset.
- **Supplementary Figure 21.** Runtime benchmarking using Visium HD datasets.
- **Supplementary Figure 22.** Memory usage benchmarking using Visium HD datasets.
- **Supplementary Figure 23.** Performance evaluation of HRCHY-CytoCommunity using the simulated dataset.
- **Supplementary Figure 24.** Ambiguous structures in the mouse Slide-seq V2 dataset.

- **Supplementary Note 1.** Fine-grained CNs identified in the mouse spleen CODEX and human CRC Visium HD datasets.
- **Supplementary Note 2.** Coarse-grained TCs identified in the mouse hypothalamic preoptic region MERFISH and intracerebral hemorrhage Stereo-seq datasets.
- **Supplementary Note 3.** Ablation studies of HRCHY-CytoCommunity.
- **Supplementary Note 4.** Sensitivity analysis of hyperparameters in HRCHY-CytoCommunity.
- **Supplementary Note 5.** Robustness assessment of HRCHY-CytoCommunity.
- **Supplementary Note 6.** Scalability and speed analysis.
- **Supplementary Note 7.** Experimental details for reproducibility.
- **Supplementary Note 8.** Extension of HRCHY-CytoCommunity to three or more hierarchical levels.

- **Supplementary Note 9.** Identification of ambiguous structures in hierarchical tissue organization.
- **Supplementary Table 1.** Details of spatial omics datasets used for performance evaluation.
- **Supplementary Table 2.** Hyperparameter settings of HRCHY-CytoCommunity used in all datasets.
- **Supplementary Table 3.** Hyperparameter settings of NeST used in all datasets.
- **Supplementary Table 4.** Cell-type composition profiles for CNs in simulation data.

In the following, reviewers' comments remain in **black**, our point-by-point answers appear in **blue**. Text and figures we have added or altered in the main manuscript are also highlighted in **blue**, while the supplementary information appears in **black**, as the entire document has been comprehensively updated.

Response to reviewer comments

Reviewer #1 (Remarks to the Author):

HRCHY-CytoCommunity presents a novel graph neural network (GNN)-based framework for identifying hierarchical multicellular structures in spatial omics data. Leveraging differentiable graph pooling and graph pruning, the method robustly uncovers nested tissue compartments and cellular neighborhoods from cell-type spatial maps. Overall, this work is a significant advance in spatial omics analysis with biological and clinical insights.

Major advances of this approach:

- 1) HRCHY-CytoCommunity outperforms the state-of-the-art NeST method across multiple datasets (mouse spleen CODEX, hypothalamic MERFISH, TNBC MIBI-TOF). It accurately identifies spatially discontinuous structures and achieves higher accuracy, adjusted mutual information, and Macro-F1 scores.
- 2) The hierarchical majority voting strategy enhances reproducibility while preserving nested relationships between TCs and CNs. Unlike NeST, it assigns all cells to structures, avoiding coverage gaps.
- 3) A cell-type enrichment-based clustering module aligns structures across samples, revealing conserved and patient-specific biological features (e.g., 26 unified CNs in TNBC).
- 4) The method reveals functional hierarchies, such as neuron/non-neuron interactions in the hypothalamus and immune-tumor compartmentalization in TNBC, aligning with known biology. Applied to breast cancer, HRCHY-CytoCommunity enables multi-scale prognostic stratification.

Limitations:

R1C1

1. Researchers need to pre-define/select structure counts/levels. Are there any quantitative indexes to guide the hyper-parameter settings?

We appreciate the reviewer’s insightful comment. In the revised manuscript, inspired by the state-of-the-art spatial domain detection method CellCharter¹, we introduced a **cluster stability metric based on the Fowlkes-Mallows Index (FMI)** to guide the selection of the numbers of fine-grained cellular neighborhoods (CNs, denoted as c_1) and coarse-grained tissue compartments (TCs, denoted as c_2).

In this approach, users were first required to define a candidate search range for (c_1, c_2) and the number of repeated runs R . For each candidate pair (c_1, c_2) , the model was trained R times independently. The stability of the resulting hierarchical structures (i.e., CNs and TCs) was then quantified using the FMI computed across runs. Specifically, the FMI was computed separately for the identified CNs and TCs across all pairs of runs for a given (c_1, c_2) configuration. This yielded a set of CN-level FMIs and a set of TC-level FMIs. For each level, the median FMI across all run pairs was computed to summarize the cluster stability at that level. The final stability score for a (c_1, c_2) configuration was defined as the average of the CN-level and TC-level median FMIs. The optimal (c_1, c_2) pair was selected as the one that yields the highest stability score. Finally, among the R models trained under the optimal (c_1, c_2) configuration, the model achieving the lowest training loss was chosen to output the final hierarchical tissue structures.

This procedure provides a quantitative criterion to guide the selection of (c_1, c_2) rather than relying solely on *ad hoc* or domain-specific heuristics. More importantly, it also mitigates the risk of over-segmentation when large values of (c_1, c_2) are used. Specifically, when the number of clusters is overestimated, the resulting clusters tend to be unstable across repeated runs, which is reflected by low FMI stability. Conversely, when the chosen (c_1, c_2) properly matches the underlying tissue organization, the model yields reproducible partitions with high FMI stability.

Taken together, instead of requiring users to specify (c_1, c_2) solely based on prior knowledge, HRCHY-CytoCommunity employs an FMI-based cluster stability criterion to determine the optimal values of (c_1, c_2) , while guarding against over-segmentation. This ensures that HRCHY-CytoCommunity can identify robust and stable hierarchical structures.

Full details are provided in the **Methods** section under the title “**Determination of optimal numbers of hierarchical tissue structures based on cluster stability**”.

R1C2

2. Can this model be adapted to RNA-seq based spot-structure data, such as Visium HD & StereoSeq?

We appreciate the reviewer for raising this valuable point. **Yes, HRCHY-CytoCommunity can be directly applied to RNA-seq based spot-structure data such as Visium HD and Stereo-seq.** To demonstrate this, we systematically evaluated its performance on four representative sequencing-based spatial transcriptomics datasets spanning diverse resolutions and technologies. Specifically:

1. **Human colorectal cancer (CRC) Visium HD dataset (8 μm -bin resolution):** HRCHY-CytoCommunity was the only method that accurately distinguished tumor and normal compartments in the P2CRC sample. Similarly, in the remaining samples, these two compartments were also clearly delineated, though with moderate inclusion of intestinal epithelial-enriched regions within the tumor compartment (Response Fig. 1b, left column). Quantitatively, HRCHY-CytoCommunity achieved the highest average AMI and Macro-F1 scores among all representative state-of-the-art methods (Response Fig. 1d).
2. **Mouse intracerebral hemorrhage (ICH) Stereo-seq dataset (Single-cell resolution):** HRCHY-CytoCommunity achieved high accuracy in identifying fine-grained CNs, particularly excelling in detecting small-scale CNs such as olfactory areas (OLF), meninges, and pallidum (PAL) (Response Fig. 2b, left column). Quantitatively, HRCHY-CytoCommunity achieved significantly higher AMI and Macro-F1 scores than NeST (one-sided paired *t*-test, *P*-values $< 1\text{E-}7$) and other non-hierarchical methods (one-sided paired *t*-test, *P*-values < 0.001), with performance comparable to NicheCompass (Response Fig. 2d).
3. **Mouse hippocampus Slide-seq V2 dataset (10 μm -spot resolution):** HRCHY-CytoCommunity recovered biologically meaningful hierarchical structures. For coarse-grained TCs, our model accurately delineated the cortical region, hippocampus, and brain stem (Response Fig. 3b, middle panel). For fine-grained CNs, our model accurately recovered substructures including the CA1-CA3 regions, dentate gyrus, corpus callosum, and distinct cortical layers (Response Fig. 3c, middle panel), closely matching the reference Allen Mouse Brain Atlas².
4. **Human breast cancer Visium V1 dataset (55 μm -spot resolution):** HRCHY-CytoCommunity accurately separated tumor and non-tumor compartments (Response Fig. 4b, middle panel) and identified fine-grained CNs consistent with manual annotation, such as DCIS/LCIS_1, DCIS/LCIS_3, and IDC_4 (Response Fig. 4c, middle panel), in line with manual annotation from Xu et al.³.

In summary, these results demonstrate the **strong generalizability** of HRCHY-CytoCommunity to RNA-seq based spot-structure data spanning diverse technologies and resolutions.

Full details of the extended performance evaluation on sequencing-based data are provided in the **Results** section under the titles: “**Performance evaluation on coarse-grained TC**

identification”, “Performance evaluation on fine-grained CN identification”, and “Cross-platform generalization of HRCHY-CytoCommunity to low-resolution spatial transcriptomics”, as well as in Supplementary Notes 1 and 2.

Response Fig. 1 (extracted from Fig. 2). Performance evaluation of HRCHY-CytoCommunity on the human CRC Visium HD dataset. (a) Spatial maps of human CRC samples (8 μ m bins) generated by Visium HD technology. Bins are colored by cell types (left) and manually annotated coarse-grained structures (right). **(b)** Coarse-grained TCs identified by HRCHY-CytoCommunity (left) and NeST (right). **(c)** Coarse-grained TCs identified by non-hierarchical methods, including CellCharter, NicheCompass, and SpaSEG. **(d)** AMI and Macro-F1 scores calculated using manual compartment annotations. Each point corresponds to the performance on an individual sample, with horizontal bars indicating the mean performance across all samples. Points from the same sample are connected by grey dashed lines. *P*-values were calculated using one-sided paired *t*-tests. *, *P*-value < 0.05; **, *P*-value < 0.01; ***, *P*-value < 0.001. “Unidentified” indicates cells not assigned to any TC. “Unmatched” denotes TCs that could not be aligned with manual annotations.

Response Fig. 2 (Supplementary Fig. 5). Performance evaluation of HRCHY-CytoCommunity on the mouse ICH Stereo-seq dataset. (a) Spatial maps showing the distribution of cell types (left) and molecularly

defined regions (right). **(b)** Fine-grained CNs identified by HRCHY-CytoCommunity (left) and NeST (right). **(c)** Fine-grained CNs identified by non-hierarchical tissue structure identification methods, including CellCharter, NicheCompass, GraphST, and SpaGCN. “Unidentified” indicates cells not assigned to any CN. **(d)** AMI and Macro-F1 scores calculated using molecularly defined regions. Each point corresponds to the performance on an individual sample, with horizontal bars indicating the mean performance across all samples. Points from the same samples are connected by grey dashed lines. *P*-values were calculated using a one-sided paired *t*-test. *, *P*-value < 0.05; **, *P*-value < 0.01; ***, *P*-value < 0.001. “Unidentified” indicates cells not assigned to any CN. L2/3, cortical layer 2 and cortical layer 3; L4, cortical layer 4; L5, cortical layer 5; L6, cortical layer 6; Lesion, Lesion region; injured_STR, injured striatum region; injured_cortex, injured cortex region; injured_PAL, injured pallidum region; DG, dentate gyrus; FT, fiber tracts; HY, hypothalamus; OLF, olfactory areas; MH, medial habenula; CA, Ammon’s horn; STRd, striatum dorsal region; STRv, striatum ventral region; TH, thalamus; VS, ventricular systems; isl, islm, islands of Calleja and major island of Calleja; PAL, pallidum; RT, reticular nucleus of the thalamus; CASp, field Ammon’s horn, pyramidal layer; LSX, lateral septal complex; sAMY, striatum-like amygdala nuclei; CTXsp, cortical subplate.

Response Fig. 3 (extracted from Fig. 4 and Supplementary Fig. 6). Performance evaluation of HRCHY-CytoCommunity on the mouse hippocampus Slide-seq V2 dataset. (a) Cluster stability measure used by HRCHY-CytoCommunity to determine the optimal numbers of coarse-grained TCs and fine-grained CNs. The Fowlkes-Mallows Index-based cluster stability values under each parameter setting are connected by solid lines. The red circle marks the selected stable resolution of tissue structures. **(b)** Coarse-grained TC reference from the Allen Mouse Brain Atlas ² (left), and TCs identified by HRCHY-CytoCommunity (middle) and NeST (right) in the mouse hippocampus. **(c)** Fine-grained CN reference from the Allen Mouse Brain Atlas (left), and CNs identified by HRCHY-CytoCommunity (middle) and NeST (right) in the mouse hippocampus. V3, third ventricle; MH, medial habenula; LH, lateral habenula; LP, lateral posterior nucleus of the thalamus; LD, lateral dorsal nucleus of the thalamus. “Unidentified” indicates cells not assigned to any TC or CN. “Unmatched” denotes TCs or CNs that could not be aligned with the Allen Mouse Brain Atlas.

Response Fig. 4 (extracted from Fig. 4 and Supplementary Fig. 7). Performance evaluation of HRCHY-CytoCommunity on the human breast cancer Visium V1 dataset. (a) Cluster stability measure used by HRCHY-CytoCommunity to determine the optimal numbers of coarse-grained TCs and fine-grained CNs. The Fowlkes-Mallows Index-based cluster stability values under each parameter setting are connected by solid lines. The red circle marks the selected stable resolution of tissue structures. **(b)** Manual annotation of

coarse-grained TCs from Xu et al.³ (left), and TCs identified by HRCHY-CytoCommunity (middle) and NeST (right) in the human breast cancer. (c) Manual annotation of fine-grained CNs from Xu et al. (left), and CNs identified by HRCHY-CytoCommunity (middle) and NeST (right) in the human breast cancer. “Unidentified” indicates cells not assigned to any TC or CN. Red arrowheads indicate the IDC_3 region manually annotated by Xu et al., which was reclassified by both HRCHY-CytoCommunity and NeST as a non-tumor CN.

Reviewer #2 (Remarks to the Author):

In this study, Wang et al. introduced a spatial clustering framework HRCHY-CytoCommunity. Unlike conventional spatial clustering methods such as SpaGCN or graphST that deliver a single level spatial domain clustering, the proposed method can present hierarchical spatial clustering, detecting nested spatial domains and structures that can not be detected with the conventional methods. Considering the complexity of the tissue microenvironment in various complex diseases such as cancer, the capability to detect hierarchical spatial clustering can help us better understand the spatial patterns underlying various biological conditions and diseases, therefore it surely will be of interest of an ever-growing community of single-cell spatial genomics. Although I am fully convinced the importance of the problem and appreciate the authors effort in addressing this issue, I found that the authors may need to enhance the benchmarking (current only one single method was compared and benchmarked) and add a ablation study (to showcase the necessity of the all modules that they utilized). Beyond showing the effectiveness of the method and its superior performance in comparison with other method, the authors may also want to show more evidence to justify why the proposed touchup major voting strategy (NOT a integral part of the model and thus can't be optimized all together) is solid and won't damage the performance dramatically compared to other integrative alternative strategy (see major comment #2)

Major concerns

R2C1

(1) A potential limitation of the HRCHY-CytoCommunity method lies in its reliance on user-specified hyperparameters c_1 and c_2 , which define the number of fine-grained cellular neighborhoods (CNs) and coarse-grained tissue compartments (TCs), respectively. While the authors acknowledge that the model can learn fewer clusters than specified if overestimated, there is no principled strategy provided for selecting appropriate values of c_1 and c_2 in practice. This imposes a significant constraint on the method's generalizability, particularly in real-world settings where prior knowledge about hierarchical tissue architecture is limited or absent. Without an automated or data-driven approach to determine optimal resolution, users must rely on ad hoc or domain-specific heuristics, which may lead to suboptimal or biologically misleading interpretations. A discussion on how to systematically tune or infer these

hyperparameter, ideally integrated to be part of the model, would substantially improve the applicability and robustness of the framework.

We appreciate the reviewer’s insightful comment and acknowledge that requiring users to predefine the numbers of fine-grained cellular neighborhoods (CNs, denoted as c_1) and coarse-grained tissue compartments (TCs, denoted as c_2) could constrain the generalizability of HRCHY-CytoCommunity. To address this issue, inspired by the state-of-the-art spatial domain detection method CellCharter¹, we introduced a **data-driven heuristic strategy based on cluster stability** to determine the optimal values of (c_1, c_2) in the revised manuscript.

In this approach, users were first required to define a candidate search range for (c_1, c_2) and the number of repeated runs R . For each candidate pair (c_1, c_2) , the model was trained R times independently. The stability of the resulting hierarchical structures (i.e., CNs and TCs) was then quantified using the Fowlkes-Mallows Index (FMI) computed across runs. Specifically, the FMI was computed separately for the identified CNs and TCs across all pairs of runs for a given (c_1, c_2) configuration. This yielded a set of CN-level FMIs and a set of TC-level FMIs. For each level, the median FMI across all run pairs was computed to summarize the cluster stability at that level. The final stability score for a (c_1, c_2) configuration was defined as the average of the CN-level and TC-level median FMIs. The optimal (c_1, c_2) pair was selected as the one that yields the highest stability score. Finally, among the R models trained under the optimal (c_1, c_2) configuration, the model achieving the lowest training loss was chosen to output the final hierarchical tissue structures.

This procedure provides a data-driven criterion to guide the selection of (c_1, c_2) rather than relying solely on *ad hoc* or domain-specific heuristics. More importantly, it also mitigates the risk of over-segmentation when large values of (c_1, c_2) are used. Specifically, when the number of clusters is overestimated, the resulting clusters tend to be unstable across repeated runs, which is reflected by lower FMI stability. Conversely, when the chosen (c_1, c_2) properly matches the underlying tissue organization, the model yields reproducible partitions with high FMI stability.

Taken together, instead of requiring users to specify (c_1, c_2) solely based on prior knowledge, HRCHY-CytoCommunity employs a **data-driven heuristic strategy to determine the optimal values of (c_1, c_2)** , while guarding against over-segmentation. This ensures that HRCHY-CytoCommunity can identify robust and stable hierarchical tissue structures.

Full details are provided in the **Methods** section under the title “**Determination of optimal numbers of hierarchical tissue structures based on cluster stability**”.

R2C2

(2) The hierarchical majority voting strategy used in HRCHY-CytoCommunity, while helpful for improving stability and enforcing a nested structure, is applied *post hoc* and is not integrated into the model’s training objective. This disconnection introduces two key issues. First, the

learned soft assignments are not explicitly optimized to yield stable or hierarchically consistent outputs, which may lead to mismatches between training-time objectives and final outputs. Second, because the voting procedure is non-differentiable, it prevents end-to-end training and may obscure the interpretation of how hierarchical structures are learned. More critically, this touch-up procedure could obscure systematic instability in the underlying GNN assignments, which would not be evident without the ensemble step. As a result, the biological or clinical significance of the identified structures may hinge on a potentially brittle post-processing routine.

As a practical alternative, the authors could consider incorporating a consistency regularization strategy, which encourages alignment of cluster assignments across stochastic runs during training, or a differentiable nesting regularization term, which softly penalizes misalignment between fine- and coarse-grained structures. These approaches would enable the model to learn hierarchical relationships more robustly and transparently during optimization. One quick strategy is to introduce consistency regularization during training. This involves running the model multiple times with stochastic perturbations (e.g., dropout or graph augmentation) and penalizing discrepancies in the predicted cluster assignments across runs. By minimizing divergence (e.g., KL divergence) between outputs, the model is encouraged to produce stable and robust hierarchical assignments without relying on post hoc voting.

At a minimum, the authors should implement and compare one of these integrative alternatives against the current ensemble-based (touchup major voting based) strategy to demonstrate that the model's performance and robustness are not overly dependent on post hoc correction. Such a comparison would strengthen the claims of generalizability and biological interpretability. If the performances are indeed similar (even slightly worse), then the authors may not need to update all the results with the new model structure, which is quite intensive.

We appreciate the reviewer for this constructive suggestion. We fully agree that the *post hoc* majority-voting ensemble has limitations, as it is non-differentiable and not directly integrated into the model training. In the revised manuscript, we have addressed this concern by **replacing the *post hoc* ensemble with a dropout-based consistency regularization strategy during training**. Specifically,

- We introduced a **consistency regularization term** (adapted from the GRAND⁴ framework) that enforces agreement of hierarchical structure assignments across multiple stochastic perturbations, where these perturbations are generated using DropNode (random masking of node attribute vectors) (Response Fig. 5b).
- We further introduced an **entropy-based balance regularization term** that encourages balanced cluster assignments by maximizing the entropy of the predicted distributions across clusters to prevent collapse of the coarse-grained TC assignments.
- These regularization terms were jointly optimized with the unsupervised hierarchical tissue structure assignment objective, **enabling fully end-to-end training without relying on *post hoc* corrections** (Response Fig. 5b).

Response Fig. 5 (Fig. 1). Overview of the HRCHY-CytoCommunity framework. The framework formulates hierarchical tissue structure identification as a hierarchical community detection problem on a cell-cell proximity graph. HRCHY-CytoCommunity includes two modules: **(a)** a soft hierarchical structure assignment base module and **(b)** a robust hierarchical structure identification module. **(a)** The base module begins with a cell-cell proximity graph, where nodes represent cells and their attributes are one-hot encoded cell-type vectors. A graph convolution layer followed by a fully-connected layer transforms node attribute vectors into soft fine-grained cellular neighborhood (CN) assignment vectors. A differentiable graph pooling layer is then used to generate a coarsened completed graph, where each pooled node represents a CN. Adaptive edge pruning is applied to mitigate over-smoothing. Another graph convolution and fully-connected layers process the pooled node embeddings to produce soft coarse-grained tissue compartment (TC) assignment vectors. The entire process is optimized by a graph minimum cut (MinCut)-based loss function, denoted as L_{Base} . **(b)** To enhance the accuracy and robustness of hierarchical tissue structure

identification, M perturbed cell-cell proximity graphs are generated via DropNode (random masking of node attribute vectors). Each graph is processed by the base module to produce soft fine-grained CN and coarse-grained TC assignment matrices. Consistency and balance regularization during training enforces stability across these M assignments. The total loss, denoted as L_{Total} , combines the base MinCut loss (L_{Base}) with a consistency regularization term (L_{Consis} , which minimizes divergence across M assignments) and a balance regularization term (L_{Balance} , which prevents cluster collapse). Finally, this module yields stable assignment matrices, from which final robust hierarchical tissue structures are derived.

To assess the effectiveness of the consistency and balance regularization terms, we conducted ablation studies on both simulated and real datasets. The results demonstrated that **HRCHY-CytoCommunity with these regularization terms substantially improves the accuracy and stability of coarse-grained TC identification compared to HRCHY-CytoCommunity without regularization** (denoted as *base model*) (Response Figs. 6a and b). For fine-grained CN identification, HRCHY-CytoCommunity consistently achieved stable and strong performance (Response Figs. 6a and c)

To further evaluate whether this regularization strategy provides additional advantages in accuracy over the original *post hoc* ensemble procedure, we compared the performance of the *base model* trained with consistency and balance regularization (i.e., *Base model + Consistency & Balance reg*) against the majority voting ensemble of 10 independent runs of the *base model* (i.e., *Base model + Majority voting ensemble*). The results showed that **the regularized model consistently achieved higher accuracy across all real datasets** (Response Fig. 7a).

We also examined whether the regularization strategy improves model stability and prevents the cluster collapse by comparing the similarity of predictions across repeated independent runs in the *base model* and the regularized model. Stability was quantified using the FMI between pairs of resulting clusters. The results showed that **introducing the consistency and balance regularization terms enhances the model stability and effectively prevents cluster collapse** (Response Figs. 7b and c).

In summary, these results demonstrate that incorporating the consistency and balance regularization terms substantially **improves both accuracy and stability** of HRCHY-CytoCommunity. Accordingly, in the revised manuscript, **all ensemble-based results reported in the original version have been replaced by those obtained from the model with consistency and balance regularization**.

Full details of the consistency and balance regularization strategy are provided in the **Methods** section under the title “**Robust hierarchical tissue structure identification**”. Full details of the ablation studies are provided in **Supplementary Note 3** and **Supplementary Figs. 13-15**.

Response Fig. 6 (extracted from Supplementary Figs. 13-15). Ablation study of consistency and balance regularization on simulated and real spatial omics datasets. Hierarchical tissue structure identification under different model configurations. Each configuration was run 10 times with different random seeds. Boxplot elements are defined as: center line, median; box limits, upper and lower quartiles; whiskers, $1.5\times$ interquartile range. Performances are quantified by Adjusted Mutual Information (AMI) and Macro-F1 scores. Each point corresponds to the performance of one run. Ablations were performed on **(a)** the simulated dataset, **(b)** real datasets with ground-truth coarse-grained TCs including the mouse spleen CODEX dataset (imaging-based) and the human colorectal cancer (CRC) Visium HD dataset (sequencing-based), as well as **(c)** real datasets with ground-truth fine-grained CNs including the mouse hypothalamic preoptic region MERFISH dataset (imaging-based) and the mouse intracerebral hemorrhage (ICH) Stereo-seq dataset (sequencing-based).

Response Fig. 7. Accuracy and stability evaluation of different versions of HRCHY-CytoCommunity. (a) Accuracy evaluation comparing the base model of HRCHY-CytoCommunity trained with consistency and balance regularization terms (i.e., Base model + Consistency & Balance reg) against the base model with *post hoc* ensemble (i.e., Base model + Majority voting ensemble) across all real datasets with ground-truth annotations generated from diverse technologies (i.e., the mouse spleen CODEX dataset, the human colorectal cancer Visium HD dataset, the mouse hypothalamic preoptic region MERFISH dataset, and the mouse intracerebral hemorrhage Stereo-seq dataset). Accuracy is quantified by Adjusted Mutual Information (AMI) and Macro-F1 scores. (b-c) Stability evaluation of (b) identified coarse-grained TCs and (c) identified fine-grained CNs from the base model and the regularized model (i.e., Base model + Consistency & Balance reg). The stability score is quantified by Fowlkes-Mallows Index (FMI). Specifically, when all cells were assigned to a single cluster in a run (cluster collapse), the stability score involving that run was set to zero. Boxplot elements are defined as: center line, median; box limits, upper and lower quartiles; whiskers, $1.5 \times$

interquartile range. Each dot represents the stability score between results from two independent runs with varying random seeds.

R2C3

(3) The developed method integrates multiple components, including a hierarchical GNN with differentiable pooling, a post hoc majority-voting ensemble for robustness, and a clustering-based module for cross-sample alignment. However, the manuscript does not provide an ablation study to evaluate the individual contribution of these modules to overall performance. In particular, it remains unclear how much of the reported improvement over baseline methods arises from the GNN-based hierarchical learning itself, versus the ensemble voting or cross-sample unification steps. An ablation study removing or modifying each module, such as training without voting, disabling edge pruning, or skipping the alignment step, would be valuable for confirming that each component is essential and for understanding where the gains in performance and robustness originate.

We appreciate the reviewer's valuable suggestion. In the revised manuscript, we conducted comprehensive ablation studies on both simulated and real datasets to evaluate the individual contributions of key components in HRCHY-CytoCommunity. Notably, in this revised version, the *post hoc* majority-voting ensemble used in the original manuscript has been replaced by the consistency and balance regularization strategy, and the clustering-based cross-sample alignment module is an optional component that is not involved in performance benchmarking. Therefore, our ablation studies focus on the following three key components:

- 1) **Consistency and balance regularization terms**, designed to improve the stability of hierarchical tissue structure assignments and prevent cluster collapse of coarse-grained TC identification.
- 2) **Adaptive edge pruning**, designed to automatically remove edges between fine-grained CNs to reduce over-smoothing during message passing in coarse-grained TC identification.
- 3) **Adaptive α scheduling**, which dynamically balances fine- and coarse-level tissue structure identification during training.

Our results showed that:

1. For the task of coarse-grained TC identification:
 - a) The best performance was consistently achieved when both the consistency and balance regularization terms were applied together, confirming their complementary roles in improving accuracy and stability while preventing cluster collapse (Response Figs. 8a and 9a).

- b) The adaptive edge-pruning strategy outperformed both extreme configurations (i.e., preserving all edges and removing all edges) in both simulated and real datasets, confirming its effectiveness in filtering trivial connections between fine-grained CNs (Response Figs. 8b and 9b).
- c) The adaptive α scheduling strategy had little effect on the simulated dataset but improved performance in real datasets, suggesting that it facilitates progressive optimization of hierarchical clustering under complex biological structures (Response Figs. 8c and 9c).

2. For the task of fine-grained CN identification:

- a) The three components did not substantially alter performance, which remained stable across different settings in both simulated and real datasets (Response Figs. 8 and 10). This is consistent with expectations, as these components were primarily designed for preventing suboptimal coarse-grained TC identification.

Taken together, these ablation studies demonstrate that the three evaluated components of HRCHY-CytoCommunity play critical and complementary roles in coarse-grained TC identification. In contrast, for fine-grained CN identification, the model maintains robust performance across varying component settings.

Full details of the ablation studies are provided in **Supplementary Note 3** and **Supplementary Figs. 13-15**.

Response Fig. 8 (Supplementary Fig. 13). Ablation study of key components in HRCHY-CytoCommunity using the simulated dataset. Hierarchical tissue structure identification under different model configurations. Each configuration was run 10 times with different random seeds. Boxplot elements are defined as: center line, median; box limits, upper and lower quartiles; whiskers, $1.5 \times$ interquartile range. Performances are quantified by Adjusted Mutual Information (AMI) and Macro-F1 scores. Each point corresponds to the performance of one run. Ablation studies were performed on **(a)** consistency and balance regularization terms, **(b)** the adaptive edge-pruning strategy, and **(c)** the adaptive α scheduling strategy.

Response Fig. 9 (Supplementary Fig. 14). Ablation study of key components in HRCHY-CytoCommunity on coarse-grained TC identification using real spatial omics datasets. Hierarchical tissue structure identification under different model configurations using the mouse spleen CODEX dataset (imaging-based) and the human colorectal cancer (CRC) Visium HD dataset (sequencing-based). Each run was run 10 times with different random seeds. Boxplot elements are defined as: center line, median; box limits, upper and lower quartiles; whiskers, $1.5\times$ interquartile range. Performances are quantified by Adjusted Mutual Information (AMI) and Macro-F1 scores. Each point corresponds to the performance of one run. Ablation studies were performed on **(a)** consistency and balance regularization terms, **(b)** the adaptive edge-pruning strategy, and **(c)** the adaptive α scheduling strategy.

Response Fig. 10 (Supplementary Fig. 15). Ablation study of key components in HRCHY-CytoCommunity on fine-grained CN identification using real spatial omics datasets. Hierarchical tissue structure identification under different model configurations using the mouse hypothalamic preoptic region MERFISH dataset (imaging-based) and the mouse intracerebral hemorrhage (ICH) Stereo-seq dataset (sequencing-based). Each configuration was run 10 times with different random seeds. Boxplot elements are defined as: center line, median; box limits, upper and lower quartiles; whiskers, $1.5\times$ interquartile range. Performances are quantified by Adjusted Mutual Information (AMI) and Macro-F1 scores. Each point corresponds to the performance of one run. Ablation studies were performed on **(a)** consistency and balance regularization terms, **(b)** the adaptive edge-pruning strategy, and **(c)** the adaptive α scheduling strategy.

R2C4

(4) The current benchmarking is restricted to a single method, NeST, which, while hierarchical, is designed for high-dimensional spatial transcriptomics and does not generalize well to low-dimensional or cell-type-based spatial data such as spatial proteomics. This narrow comparison limits the ability to assess the broader utility and competitiveness of HRCHY-CytoCommunity. In particular, several more recent and widely used frameworks—such as SpaSEG, Giotto, and

GraphST—are highly relevant comparators. Although these methods do not explicitly model hierarchical structures, they can be run at multiple resolutions or applied recursively to simulate hierarchical segmentation. Benchmarking against such methods would provide a more comprehensive evaluation of whether HRCHY-CytoCommunity's explicit hierarchical modeling offers practical advantages in terms of spatial coherence, robustness, and biological relevance. The authors are strongly encouraged to include comparisons to these additional methods or, at a minimum, to discuss their relative performance.

We thank the reviewer for the important suggestion. Following the suggestion from Reviewers 2 and 3, in the revised manuscript, we have benchmarked HRCHY-CytoCommunity against six state-of-the-art non-hierarchical tissue structure identification methods, including SpaSEG⁵, Giotto Suite⁶, GraphST⁷, SpaGCN⁸, NicheCompass⁹, and CellCharter¹. Since these methods do not explicitly identify hierarchical tissue structures, we applied them separately at coarse-grained TC and fine-grained CN resolutions. Performances were evaluated using Adjusted Mutual Information (AMI) and Macro-F1 scores across two datasets with ground-truth TCs and two datasets with ground-truth CNs.

Our results demonstrate that **HRCHY-CytoCommunity consistently achieved superior or competitive performance than non-hierarchical methods**. Specifically:

1. For coarse-grained TC identification:

- a) On the **mouse spleen CODEX dataset (imaging-based)**, HRCHY-CytoCommunity was the only method that consistently resolved the biologically meaningful lymphoid compartment containing B-zone, periarteriolar lymphoid sheath, and marginal zone across all samples (Response Fig. 11b, left column), whereas non-hierarchical methods largely failed to capture this hierarchical structure (GraphST succeeded only in one sample). Quantitatively, HRCHY-CytoCommunity achieved significantly higher AMI and Macro-F1 scores than most non-hierarchical methods (one-sided paired *t*-test *P*-values < 0.05), while exhibiting performance comparable to GraphST (Response Fig. 11d).
- b) On the **human colorectal cancer (CRC) Visium HD dataset (sequencing-based)**, HRCHY-CytoCommunity was the only method that accurately distinguished tumor and normal compartments in the P2CRC sample, while all non-hierarchical methods failed to do so. Similarly, in the remaining samples, these two compartments were also clearly delineated, though with moderate inclusion of intestinal epithelial-enriched regions within the tumor compartment (Response Fig. 12b, left column). Quantitatively, HRCHY-CytoCommunity achieved the highest average AMI and Macro-F1 scores among all representative non-hierarchical methods (Response Fig. 12d).

2. For fine-grained CN identification:

- a) On the **mouse hypothalamic preoptic region MERFISH dataset (imaging-based)**, HRCHY-CytoCommunity successfully recovered CNs consistent with manually

delineated nuclei with clear boundaries (Response Fig. 13b, left column), whereas other non-hierarchical methods identified intermixed CNs and consistently failed to capture smaller regions like PS and PaAP in most samples (Response Fig. 13c). Quantitatively, HRCHY-CytoCommunity significantly outperformed all non-hierarchical methods (one-sided paired *t*-test, $p < 0.05$) in both AMI and Macro-F1 metrics, with the exception of CellCharter on the AMI and GraphST on the Macro-F1 score, where the differences were not statistically significant (Response Fig. 13d).

- b) On the **mouse intracerebral hemorrhage (ICH) Stereo-seq dataset (sequencing-based)** HRCHY-CytoCommunity achieved high accuracy in identifying fine-grained CNs, particularly excelling in detecting small-scale CNs such as olfactory areas (OLF), meninges, and pallidum (PAL) (Response Fig. 14b, left column). Quantitatively, HRCHY-CytoCommunity achieved significantly higher AMI and Macro-F1 scores than most non-hierarchical methods (one-sided paired *t*-test, P -values < 0.001), with performance comparable to NicheCompass (Response Fig. 14d).

In summary, these results indicate that HRCHY-CytoCommunity is not only quantitatively competitive but also qualitatively superior, as it **identifies biologically meaningful structures missed by non-hierarchical models**. This highlights that the **explicit hierarchical framework yields more accurate, stable, and interpretable multi-scale tissue organization** across diverse technologies.

Full implementation details of these non-hierarchical spatial domain detection methods are provided in the **Methods** section under the title “**Running of published methods**”. Full details of the performance evaluation are provided in the **Results** section under the titles: “**Performance evaluation on coarse-grained TC identification**”, and “**Performance evaluation on fine-grained CN identification**”, as well as in **Supplementary Notes 1 and 2**.

Response Fig. 11 (extracted from Fig. 2) Performance evaluation of HRCHY-CytoCommunity on the mouse spleen CODEX dataset. (a) Spatial maps of healthy mouse spleen samples generated by CODEX technology. Cells are colored by cell types (left) and manually annotated compartments (right), which serve as the ground truth for coarse-grained TCs. **(b)** Coarse-grained TCs identified by HRCHY-CytoCommunity (left) and NeST (right). **(c)** Coarse-grained TCs identified by non-hierarchical methods, including CellCharter, NicheCompass, GraphST, and SpaGCN. **(d)** Adjusted Mutual Information (AMI) and Macro-F1 scores calculated using manual compartment annotations. Each point corresponds to the performance on an individual sample, with horizontal bars indicating the mean performance across all samples. Points from the same sample are connected by grey dashed lines. *P*-values were calculated using one-sided paired *t*-tests. *, *P*-value < 0.05; **, *P*-value < 0.01; ***, *P*-value < 0.001. “Unidentified” indicates cells not assigned to any TC. “Unmatched” denotes TCs that could not be aligned with manual annotations.

Response Fig. 12 (extracted from Fig. 2). Performance evaluation of HRCHY-CytoCommunity on the human CRC Visium HD dataset. (a) Spatial maps of human CRC samples (8 μm bins) generated by Visium HD technology. Bins are colored by cell types (left) and manually annotated coarse-grained structures (right). **(b)** Coarse-grained TCs identified by HRCHY-CytoCommunity (left) and NeST (right). **(c)** Coarse-grained TCs identified by non-hierarchical methods, including CellCharter, NicheCompass, and SpaSEG. **(d)** AMI and Macro-F1 scores calculated using manual compartment annotations. Each point corresponds to the performance on an individual sample, with horizontal bars indicating the mean performance across all samples. Points from the same sample are connected by grey dashed lines. P -values were calculated using one-sided paired t -tests. *, P -value < 0.05; **, P -value < 0.01; ***, P -value < 0.001. “Unidentified” indicates cells not assigned to any TC. “Unmatched” denotes TCs that could not be aligned with manual annotations.

Response Fig. 13 (Fig. 3). Performance evaluation of HRCHY-CytoCommunity on the mouse hypothalamic preoptic region MERFISH dataset. (a) Spatial maps of mouse hypothalamic preoptic region samples generated by MERFISH technology. Cells are colored by cell types (left) and manually annotated hypothalamic nuclei (right), which were generated based on the outlines of hypothalamic nuclei (middle) and serve as the ground truth for fine-grained CNs. **(b)** Fine-grained CNs identified by HRCHY-CytoCommunity (left) and NeST (right). **(c)** Fine-grained CNs identified by non-hierarchical methods, including CellCharter, NicheCompass, GraphST, SpaGCN, SpaSEG, and Giotto Suite. **(d)** Adjusted Mutual Information (AMI) and Macro-F1 scores calculated using manually annotated hypothalamic nuclei. Each point corresponds to the performance on an individual sample, with horizontal bars indicating the mean performance across all samples. Points from the same sample are connected by grey dashed lines. *P*-values were calculated using one-sided paired *t*-tests. *, *P*-value < 0.05; **, *P*-value < 0.01; ***, *P*-value < 0.001. “Unidentified” indicates cells not assigned to any CN.

Response Fig. 14 (Supplementary Fig. 5). Performance evaluation of HRCHY-CytoCommunity on the mouse ICH Stereo-seq dataset. (a) Spatial maps showing the distribution of cell types (left) and molecularly defined regions (right). (b) Fine-grained CNs identified by HRCHY-CytoCommunity (left) and NeST (right).

(c) Fine-grained CNs identified by non-hierarchical tissue structure identification methods, including CellCharter, NicheCompass, GraphST, and SpaGCN. “Unidentified” indicates cells not assigned to any CN.

(d) AMI and Macro-F1 scores calculated using molecularly defined regions. Each point corresponds to the performance on an individual sample, with horizontal bars indicating the mean performance across all samples. Points from the same samples are connected by grey dashed lines. *P*-values were calculated using a one-sided paired *t*-test. *, *P*-value < 0.05; **, *P*-value < 0.01; ***, *P*-value < 0.001. L2/3, cortical layer 2 and cortical layer 3; L4, cortical layer 4; L5, cortical layer 5; L6, cortical layer 6; Lesion, Lesion region; injured_STR, injured striatum region; injured_cortex, injured cortex region; injured_PAL, injured pallidum region; DG, dentate gyrus; FT, fiber tracts; HY, hypothalamus; OLF, olfactory areas; MH, medial habenula; CA, Ammon’s horn; STRd, striatum dorsal region; STRv, striatum ventral region; TH, thalamus; VS, ventricular systems; isl, islm, islands of Calleja and major island of Calleja; PAL, pallidum; RT, reticular nucleus of the thalamus; CA_{sp}, field Ammon’s horn, pyramidal layer; LSX, lateral septal complex; sAMY, striatum-like amygdala nuclei; CTX_{sp}, cortical subplate.

R2C5

(5) The reliance on cell-type annotations as input features introduces a dependence on upstream preprocessing quality, which is not evaluated. The authors should assess how robust their method is to label inaccuracies or varying annotation resolutions, or clarify the assumptions about cell-type consistency across samples.

We thank the reviewer for raising this critical point. In the revised manuscript, we systematically evaluated the robustness of HRCHY-CytoCommunity to both label inaccuracies and varying annotation resolutions using two datasets with complex cell-type compositions (i.e., the mouse hypothalamic preoptic region MERFISH dataset and the human TNBC MIBI-TOF dataset). Specifically:

1. **Robustness to label inaccuracies:** We used the hierarchical tissue structures identified from unperturbed cell-type annotations as the reference. Cell-type labels were then randomly perturbed by 10%–50%, and the accuracy and consistency of the resulting hierarchical structures were compared with the reference. The FMI metric was used to quantify the consistency between the perturbed and reference structures across different noise levels.

- a) In the MERFISH dataset, the identified hierarchical structures remained highly consistent until the label error rate exceeded 40% (Response Fig. 15a; **average FMI > 0.5 when label error rate = 40%**). The accuracy of the fine-grained CNs (measured by AMI and Macro-F1 scores) showed only a moderate decline, with performance largely preserved under reasonable noise levels (Response Figs. 15b and c)
- b) In the MIBI-TOF dataset, hierarchical structures remained highly consistent across perturbation rates (Response Fig. 16a; **average FMI > 0.65 when label error rate = 50%**). The accuracy of the identified coarse-grained TCs (measured by the proportion

of neoplastic and immune cells correctly assigned to their corresponding TCs) was largely preserved under reasonable levels of annotation noise, with only a moderate decrease as the label error rate increased (Response Fig. 16b).

2. **Robustness to varying annotation resolutions:** We further assessed robustness to annotation resolution by comparing resulting structures obtained under different levels of cell-type granularity (i.e., 15 vs. 9 cell types in MERFISH, 17 vs. 11 cell types in TNBC MIBI-TOF). The FMI was used to quantify the similarity between identified structures. Specifically,
 - a) HRCHY-CytoCommunity consistently recovered stable hierarchical tissue structures across annotation resolutions, achieving **average FMI scores of 0.673** in the MERFISH dataset (Response Figs. 15d-f) and **0.590** in the MIBI-TOF dataset across samples (Response Figs. 16c-e).

Taken together, these analyses demonstrate that **HRCHY-CytoCommunity is strongly robust to both label inaccuracies and variations in annotation resolution**, alleviating concerns regarding the strong dependence on upstream cell type annotation quality.

Full details of the robustness assessment are provided in **Supplementary Note 5** and **Supplementary Figs. 19 and 20**.

Response Fig. 15 (Supplementary Fig. 19). Robustness assessment of HRCHY-CytoCommunity to label inaccuracies and annotation resolution variations using the mouse hypothalamic preoptic region MERFISH dataset. (a) Hierarchical tissue structure similarity under increasing cell-type label perturbation rates (10%-50%), quantified by Fowlkes-Mallows Index (FMI)-based robustness score. Higher values indicate greater structural consistency with the unperturbed reference. (b-c) Fine-grained CN identification accuracy under label perturbations, measured by Adjusted Mutual Information (AMI) (b) and Macro-F1 scores (c). Each point represents performance on one sample, with horizontal bars represent the mean performance across $n = 5$ samples. Performances (points) on the same sample are connected by grey dashed lines. Mean values under each label error rate are connected by blue solid lines. Shaded areas denote standard deviation. (d) Single-cell spatial maps showing the distribution of 15 cell types (left), coarse-grained TCs (middle) and fine-grained CNs (right) identified by HRCHY-CytoCommunity based on this high-resolution annotation. (e) Single-cell spatial maps showing the distribution of 9 cell types (left), coarse-grained TCs (middle) and fine-grained CNs (right) identified by HRCHY-CytoCommunity based on this low-resolution annotation. (f) Hierarchical tissue structure similarity between different cell-type annotation resolutions, quantified by FMI-based robustness score. The red dashed line indicates the mean similarity (0.673) relative to the 15-cell-type annotation-based reference.

Response Fig. 16 (Supplementary Fig. 20). Robustness assessment of HRCHY-CytoCommunity to label inaccuracies and annotation resolution variations using the human TNBC MIBI-TOF dataset. (a) Hierarchical tissue structure similarity under increasing cell-type label perturbation rates (10%-50%),

quantified by Fowlkes-Mallows Index (FMI)-based robustness score. Higher values indicate greater structural consistency with the unperturbed reference. **(b)** Coarse-grained TC identification accuracy under label perturbations, measured by proportion of neoplastic and immune cells correctly assigned to their corresponding TCs. Each point represents performance on one compartmentalized tumor sample, with horizontal bars represent the mean performance across $n = 15$ samples. Performances (points) on the same sample are connected by grey dashed lines. Mean values under each label error rate are connected by blue solid lines. Shaded areas denote standard deviation. **(c)** Single-cell spatial maps showing the distribution of 17 cell types (left), coarse-grained TCs (middle) and fine-grained CNs (right) identified by HRCHY-CytoCommunity based on this high-resolution annotation. **(d)** Single-cell spatial maps showing the distribution of 11 cell types (left), coarse-grained TCs (middle) and fine-grained CNs (right) identified by HRCHY-CytoCommunity based on this low-resolution annotation. **(e)** Hierarchical tissue structure similarity between different cell-type annotation resolutions, quantified by FMI-based robustness score. The red dashed line indicates the mean similarity (0.590) relative to the 17-cell-type annotation-based reference.

R2C6

(6) Given the multi-step pipeline and ensemble-based framework, the scalability of HRCHY-CytoCommunity in terms of runtime and memory is not discussed. The authors should provide runtime/memory profiling or clarify the computational requirements to facilitate practical adoption.

We thank the reviewer for this valuable suggestion. In the revised manuscript, we have added a dedicated scalability analysis to **assess the runtime and memory efficiency of HRCHY-CytoCommunity**. Specifically, we downsampled the largest Visium HD sample (P2CRC; 473,318 spots, 18,085 genes) to generate subsets of 1k, 10k, 100k, and 473k spots. HRCHY-CytoCommunity was evaluated in two configurations: (i) the base model (without consistency and balance regularization), and (ii) the full model (with both regularization terms). We also compared its performance against six other tissue structure identification methods, including NeST, CellCharter, NicheCompass, GraphST, SpaGCN, and SpaSEG. Our results showed that:

1. **Runtime scalability:** Both models of HRCHY-CytoCommunity exhibited excellent computational efficiency. The base model achieved the fastest training speed, completing training and hierarchical structure identification on the full 473k-spot dataset in **1.39 minutes**. The full model, which involves repeated stochastic perturbations for stability, completed the same dataset in **8.02 minutes**, ranking third after the base model and SpaSEG (Response Fig. 17a).
2. **Memory scalability:** Both models of HRCHY-CytoCommunity showed nearly **linear growth in memory usage** with data size increasing, reflecting great scalability. On the full 473k-spot dataset, the base model required only **4.09 GB** of total memory, and the full model required **8.69 GB** (Response Fig. 17b). A detailed breakdown further showed that GPU memory usage remained moderate for both models (**2.70 GB** for the base model and **7.26 GB** for the full model at 473k spots), indicating that HRCHY-CytoCommunity can be

efficiently trained on commonly available GPUs, even for large-scale datasets (Response Fig. 17d).

In summary, these results demonstrate that HRCHY-CytoCommunity achieves **highly competitive scalability in both runtime and memory efficiency**, supporting its practical application to increasingly large and high-resolution spatial omics datasets.

Full details of the scalability analysis are provided in **Supplementary Note 6** and **Supplementary Figs. 21 and 22**.

Response Fig. 17 (extracted from **Supplementary Figs 21 and 22**) **Runtime and memory scalability analysis using the Visium HD dataset (a-b)** Total runtime (a) and total memory usage (b) of eight methods including HRCHY-CytoCommunity without consistency and balance regularization (denoted as base model), HRCHY-CytoCommunity with consistency and balance regularization (denoted as full model), NeST, CellCharter, NicheCompass, GraphST, SpaGCN, and SpaSEG across Visium HD datasets of increasing sizes (from 1,000 to 473,318 spots) (c-d) Detailed decomposition of (c) runtime (d) memory usage for the all evaluated methods on the 473k-spot dataset. For each method, the runtime is separated into the training stage

(red) and clustering stage (yellow); the total memory usage is separated into the GPU memory usage (green) and RAM usage (brown).

Minor comments

R2C1

(1) HRCHY-CytoCommunity models a fixed two-level hierarchy. This assumption may limit applicability to tissues with deeper or non-uniform hierarchical organization. The authors should discuss whether and how the framework can be extended to more flexible or adaptive hierarchical depths.

We thank the reviewer for this insightful comment. Indeed, the current implementation of HRCHY-CytoCommunity models a fixed two-level hierarchy (fine-grained CNs and coarse-grained TCs), which may limit its applicability to tissues with deeper or non-uniform hierarchical organization. Conceptually, the framework is not restricted to two levels. By iteratively applying the differentiable graph pooling module and extending the MinCut-based loss function, HRCHY-CytoCommunity can be naturally generalized to three or more hierarchical levels (e.g., smaller-scale niche–CN–TC). In this study, we focused on a two-level hierarchy due to the lack of reliable multi-level ground-truth references for quantitative benchmarking, but the framework itself readily supports deeper hierarchies. For the non-uniform hierarchical organization, we agree that adaptive identification of tissue structures with variable hierarchical depths represents an important direction for future development. In the revised manuscript, we provided a detailed unified framework of extending HRCHY-CytoCommunity to arbitrary hierarchical depths in **Supplementary Note 8 (Lines 285-331, Pages 8-9 in Supplementary Information)**. We have also explicitly acknowledged this limitation and outlined future directions in the revised **Discussion** section.

“Lastly, although this study focused on a two-level hierarchy, the underlying framework of HRCHY-CytoCommunity can be readily extended to deeper hierarchies (Supplementary Note 8). Future work may explore adaptive mechanisms to automatically infer the optimal hierarchical depth or to model tissues with non-uniform organizational principles”

R2C2

(2) The interpretability of the discovered tissue structures is not systematically evaluated beyond cell-type enrichment. It would be valuable to assess how consistently the model discovers known anatomical or functional regions across samples and tissues, and how uninterpretable clusters are handled.

We thank the reviewer for this constructive suggestion. In the revised manuscript, we have systematically evaluated the interpretability and consistency of hierarchical tissue structures

identified by HRCHY-CytoCommunity across multiple datasets with known anatomical or functional organization.

For tissues with well-characterized hierarchical architectures, including the mouse spleen (CODEX), mouse hypothalamus (MERFISH), mouse hippocampus (Slide-seq V2), and mouse brain (Stereo-seq) datasets, HRCHY-CytoCommunity consistently identified anatomically and functionally meaningful regions across different samples. For example, in the spleen, two major compartments corresponding to the red pulp and lymphoid compartments were robustly recovered in all three samples. In the hypothalamus, the model consistently detected symmetric nuclei corresponding to distinct functional domains across five samples. Similarly, in the hippocampus dataset, HRCHY-CytoCommunity accurately delineated the cortical region, hippocampus, brain stem, and their corresponding substructures, in agreement with the known anatomical organization described in the Allen Mouse Brain Atlas ². Moreover, for compartmentalized tumor tissues, including colorectal cancer (Visium HD) and breast cancer (Visium V1 and MIBI-TOF), HRCHY-CytoCommunity consistently identified the tumor-dominated and normal/immune cell-dominated compartments.

For tissues lacking well-established histological ground truth, we annotated each identified TC or CN using cell-type enrichment analysis to determine the dominant cell populations and their associated biological functions. This approach enables functional interpretation of previously uncharacterized clusters.

Collectively, these analyses demonstrate that **HRCHY-CytoCommunity consistently identifies biologically interpretable and reproducible hierarchical tissue structures** across diverse tissue types and experimental platforms.

R2C3

(3) a 2-levels hierarchical structure”, ⇒ “a 2-level hierarchical structure”

We thank the reviewer for this correction. In the revised manuscript, we have revised the text from "a 2-levels hierarchical structure" to "a 2-level hierarchical structure" throughout the manuscript.

R2C4

(4) Use of inconsistent capitalization for “Cellular Neighborhoods” and “Tissue Compartments” (sometimes capitalized, sometimes not). Maybe the authors can pick one consistent style (e.g., lowercase unless at the start of a sentence).

We thank the reviewer for pointing out this inconsistency. In the revised manuscript, we have standardized the terminology such that the full terms "cellular neighborhoods" and "tissue compartments" are written in lowercase throughout the text, except when appearing at the beginning of a sentence or in section titles. Their abbreviations (CNs and TCs) remain

capitalized, consistent with common conventions. This revision has been applied consistently across the entire manuscript.

R2C5

(5) Line 179, “the coarse-grained red pulp compartment could be further divided into two fine-grained substructures, red pulp and marginal zone”. This phrasing is somewhat confusing, as it suggests that “red pulp” is both a coarse-grained and a fine-grained region, and it categorizes the marginal zone, which is typically considered a transitional or fine-grained region between red and white pulp, as a substructure of red pulp. This may need clarification or correction, especially for readers with histological expertise.

We thank the reviewer for bringing this up and apologize for the confusion. In the revised manuscript, we have clarified the description and corrected the compartment terminology to *red pulp compartment* and *lymphoid compartment* (comprising the periarteriolar lymphoid sheath, B-zone, and marginal zone), in alignment with previous histological studies^{10–12}. Some studies collectively classify these three substructures as part of the white pulp, whereas others describe the marginal zone as a transitional structure between red and white pulp. To avoid ambiguity and ensure biological accuracy, we therefore refer to this compartment as the *lymphoid compartment* throughout the revised manuscript.

Reviewer #2 (Remarks on code availability):

The github repository generally looks good to me.

The provided GitHub repository includes a basic example dataset (mouse brain MERFISH) and a structured walkthrough consisting of four steps: graph construction, soft clustering, ensemble voting, and visualization. While this demonstrates the core workflow, there are several issues. The example dataset is limited in scope, benchmark datasets from the paper (e.g., spleen or TNBC) are not provided. Also, the documentation lacks clarity on hyperparameters, and the scripts have limited error handling.

We thank the reviewer for this valuable feedback on our code. In the revised version, we have substantially improved the GitHub repository to enhance the usability and reproducibility of HRCHY-CytoCommunity.

Specifically, we have made the following updates:

1. **Expanded the example data and tutorials.** In addition to the basic mouse brain MERFISH dataset, we now provide two additional representative subsets of the **mouse spleen (CODEX)** and the **human TNBC (MIBI-TOF)** datasets used in the paper, along with corresponding preprocessed inputs for testing the hierarchical structure identification workflow. The full datasets can be accessed through the data portals referenced in the *Methods* section, following their respective data access policies.

2. **Improved documentation.** We have added detailed explanations of key hyperparameters, including DropNode rate, λ_{consis} , λ_{balance} , and number of perturbations, together with recommended values and tuning strategies.
3. **Enhanced robustness of scripts.** Error handling and input validation have been incorporated to ensure that the code runs smoothly across different computational environments.

These updates ensure that users can reproduce the full HRCHY-CytoCommunity workflow—from graph construction to hierarchical structure identification—using publicly available data and well-documented parameter settings. To further enhance usability, we have packaged HRCHY-CytoCommunity as a Python library and released it on PyPI (<https://pypi.org/project/hrchy-cytocommunity/>). The updated source code for HRCHY-CytoCommunity is available at <https://github.com/huBioinfo/HRCHY-CytoCommunity>.

Reviewer #3 (Remarks to the Author):

The manuscript reviewed introduces HRCHY-CytoCommunity, a method for identifying hierarchical tissue structures using only spatial graphs and cell-type annotations. The study presents a method to perform hierarchical structure identification using graph neural networks with pruning and pooling. The study exemplified this strategy in multiple datasets, highlighting the identification of discontinuous spatial patterns and presenting an improved performance in datasets with few expression features (e.g., protein-based data).

While these are promising goals, the manuscript relies heavily on the assumption that hierarchical organization in tissues can be fully captured using cell-type annotations alone. This may be an oversimplification of complex tissues where cellular behavior is driven by signaling gradients, cellular interactions, or dynamic gene programs. We believe this together with others exposed below, limit the current version of the manuscript from being suitable for publication. In our opinion, only a deeply revised version of the manuscript, with clarification of the following concerns should be considered for publication.

Major Comments

R3C1

1. **Assumption of Hierarchy:** The method assumes that tissue structures are organized hierarchically, which is not universally true. While this may apply to some biological systems (e.g., lymphoid tissues, brain), in others, spatial patterns are more continuous or transitional.

The manuscript should explicitly discuss the biological validity of this assumption and its limitations.

We appreciate the reviewer for this constructive suggestion. We fully agree that the assumption of hierarchical organization does not universally apply to all tissues. Indeed, our framework is motivated by the observation that many biological systems are naturally organized in a hierarchical manner^{10,13–15}. For example, the spleen comprises coarse-grained compartments (i.e., red pulp and lymphoid pulp) that further contain fine-grained substructures (i.e., marginal zone, B-cell zone, and periarteriolar lymphoid sheath). Similarly, tumor ecosystems often exhibit multi-scale organization, from macro-tissue compartments^{15–18} (e.g., tumor core and stroma) to micro-cellular neighborhoods^{19–22} (e.g., tertiary lymphoid structures and micropapillary patterns). Capturing such hierarchies facilitated the association between spatial context and tumor development, prognosis, and therapy response^{15,23,24}.

Following the reviewer’s suggestion, we have **clarified the scope of our method in the revised manuscript**. Specifically, in the **Introduction** section, we highlight several organs known to exhibit hierarchical organization, such as the brain, tumor ecosystems, and lymphoid tissues like the spleen, to better contextualize the motivation for HRCHY-CytoCommunity. In the **Discussion** section, we explicitly acknowledged the limitation that our method may not be suitable for organs without hierarchical organization, such as the liver, whose metabolic zonation²⁵ is continuously distributed along the porto-central axis..

The following revisions have been made in the manuscript:

- **Introduction:** *“Many tissues exhibit inherent hierarchical organization, wherein large structures contain smaller, functionally specialized substructures. This pattern is evident in the layered architecture of the brain, the zoned morphology of lymphoid organs such as the spleen, and the compartmentalized ecosystem of tumors^{25–27}.”*
- **Discussion:** *“As HRCHY-CytoCommunity explicitly models tissue organization as a hierarchy, it is particularly well-suited for organs with clearly stratified anatomical or functional layers. Its performance may be limited in tissues lacking strong hierarchical organization, such as liver, where metabolic zonation is continuously distributed along the porto-central axis⁵⁴.”*

R3C2

2. Evaluation on More Diverse Datasets: The manuscript focuses primarily on datasets with well-separated cell types and pre-annotated structures. To evaluate the robustness of HRCHY-CytoCommunity, we recommend including sequencing-based spatial transcriptomics datasets—especially those from the NeST study or similar benchmarks—to compare with methods like NeST, which rely on transcriptome-wide data.

We thank the reviewer for this important suggestion. In the revised manuscript, we have expanded the evaluation of HRCHY-CytoCommunity to include four representative

sequencing-based spatial transcriptomics datasets covering different resolutions and technologies, including two datasets (**the mouse hippocampus Slide-seq V2 dataset and the human breast cancer Visium V1 dataset**) originally used in the NeST study.

- 1. Mouse hippocampus Slide-seq V2 dataset (10 μm -spot resolution)** HRCHY-CytoCommunity recovered biologically meaningful hierarchical structures spanning from coarse-grained tissue compartments (TCs) including the cortical region, hippocampus, and brain stem (Response Fig. 18b, middle panel) to fine-grained cellular neighborhoods (CNs) such as CA1-CA3 regions, dentate gyrus, corpus callosum, and distinct cortical layers (Response Fig. 18c, middle panel), aligned well with the Allen Mouse Brain Atlas. In contrast, NeST produced small and fragmented substructures rather than large-scale coherent TCs at the coarse-grained level. Moreover, it left a considerable fraction of spots unassigned (labeled as “unidentified”) for both TC and CN levels, limiting the interpretability of the tissue structure (Response Figs. 18b and c, right panels).
- 2. Human breast cancer Visium V1 dataset (55 μm -spot resolution)** HRCHY-CytoCommunity accurately distinguished tumor and non-tumor compartments (Response Fig. 19b, middle panel) and recovered CNs closely matching manual annotations (Response Fig. 19c, middle panel). While NeST correctly identified several CNs consistent with manual annotations (e.g., DCIS/LCIS_1, DCIS/LCIS_3, and IDC_4), it incorrectly partitioned discontinuous tumor regions into multiple fragmented TCs at the coarse-grained level (Response Fig. 19b; right panel). Similar to the Slide-seq V2 case, NeST left the most spots located in Healthy_1 CN unassigned, reducing interpretability and completeness of the tissue structure (Response Fig. 19c; right panel).
- 3. Human colorectal cancer (CRC) Visium HD dataset (8 μm -bin resolution)** HRCHY-CytoCommunity accurately distinguished tumor and normal compartments across all samples, with moderate inclusion of intestinal epithelial-enriched regions within the tumor compartment in some samples (Response Fig. 20b, left column). Conversely, NeST incorrectly assigned most bins to the tumor compartment (Response Fig. 20b, right column). At the fine-grained level, HRCHY-CytoCommunity identified spatially coherent CNs across samples (Response Fig. 20c, left column), while NeST produced fragmented and less interpretable CNs (Response Fig. 20c, right column, cyan circles). Quantitatively, HRCHY-CytoCommunity achieved significantly higher Macro-F1 scores than NeST for coarse-grained TC identification across all three samples (one-sided paired t -test P -values < 0.01 ; Response Fig. 20d, right panel) and substantially higher average AMI values than NeST (Response Fig. 20d, left panel).
- 4. Mouse intracerebral hemorrhage (ICH) Stereo-seq dataset (single-cell resolution)** HRCHY-CytoCommunity effectively detected biologically meaningful fine-grained CNs, such as olfactory areas (OLF), meninges, and pallidum (PAL) (Response Fig. 21b, left column), whereas NeST failed to capture these CNs in most samples (Response Fig. 21b, right column). At the coarse-grained level, HRCHY-CytoCommunity consistently identified three biologically meaningful compartments: TC-0 (brain stem), TC-1 (cortex), and TC-2 (striatum), while NeST failed to distinguish these TCs. (Response Fig. 21c) Quantitatively, HRCHY-CytoCommunity achieved significantly higher AMI and Macro-F1 scores than NeST across all nine samples in fine-grained CN identification (one-sided paired t -test P -values $< 1\text{E-}7$; Response Fig. 21d).

Taken together, these results demonstrate that HRCHY-CytoCommunity generalizes well to sequencing-based datasets and **consistently recovers biologically interpretable hierarchical tissue structures across technologies and resolutions**. Compared with NeST, which often produces fragmented substructures and leaves many spots unassigned, HRCHY-CytoCommunity effectively captures both large- and small-scale spatially coherent tissue structures and achieves complete cellular coverage.

Full details of the extended performance evaluation on sequencing-based datasets are provided in the **Results** section under the titles: “**Performance evaluation on coarse-grained TC identification**”, “**Performance evaluation on fine-grained CN identification**”, and “**Cross-platform generalization of HRCHY-CytoCommunity to low-resolution spatial transcriptomics**”, as well as in **Supplementary Notes 1 and 2**.

Response Fig. 18 (extracted from Fig. 4 and Supplementary Fig. 6). Performance evaluation of HRCHY-CytoCommunity on the mouse hippocampus Slide-seq V2 dataset. (a) Cluster stability measure used by HRCHY-CytoCommunity to determine the optimal numbers of coarse-grained TCs and fine-grained CNs. The Fowlkes-Mallows Index-based cluster stability values under each parameter setting are connected

by solid lines. The red circle marks the selected stable resolution of tissue structures. **(b)** Coarse-grained TC reference from the Allen Mouse Brain Atlas ² (left), and TCs identified by HRCHY-CytoCommunity (middle) and NeST (right) in the mouse hippocampus. **(c)** Fine-grained CN reference from the Allen Mouse Brain Atlas (left), and CNs identified by HRCHY-CytoCommunity (middle) and NeST (right) in the mouse hippocampus. V3, third ventricle; MH, medial habenula; LH, lateral habenula; LP, lateral posterior nucleus of the thalamus; LD, lateral dorsal nucleus of the thalamus. “Unidentified” indicates cells not assigned to any TC or CN. “Unmatched” denotes TCs or CNs that could not be aligned with the Allen Mouse Brain Atlas .

Response Fig. 19 (extracted from Fig. 4 and Supplementary Fig. 7). Performance evaluation of HRCHY-CytoCommunity on the human breast cancer Visium V1 dataset. (a) Cluster stability measure used by HRCHY-CytoCommunity to determine the optimal numbers of coarse-grained TCs and fine-grained CNs. The Fowlkes-Mallows Index-based cluster stability values under each parameter setting are connected by solid lines. The red circle marks the selected stable resolution of tissue structures. **(b)** Manual annotation of coarse-grained TCs from Xu et al.³ (left), and TCs identified by HRCHY-CytoCommunity (middle) and NeST (right) in the human breast cancer. **(c)** Manual annotation of fine-grained CNs from Xu et al. (left), and CNs identified by HRCHY-CytoCommunity (middle) and NeST (right) in the human breast cancer. “Unidentified” indicates cells not assigned to any TC or CN. Red arrowheads indicate the IDC₃ region

manually annotated by Xu et al., which was reclassified by both HRCHY-CytoCommunity and NeST as a non-tumor CN.

Response Fig. 20 (extracted from Fig. 2 and Supplementary Fig. 3). Performance comparison of HRCHY-CytoCommunity and NeST on the human CRC Visium HD dataset. (a) Spatial maps of human CRC samples (8 μ m bins) generated by Visium HD technology. Bins are colored by cell types (left) and manually annotated coarse-grained structures (right). **(b)** Coarse-grained TCs identified by HRCHY-

CytoCommunity (left) and NeST (right). **(c)** Fine-grained CNs identified by HRCHY-CytoCommunity (left) and NeST (right). **(d)** AMI and Macro-F1 scores calculated using manual compartment annotations. Each point corresponds to the performance on an individual sample, with horizontal bars indicating the mean performance across all samples. Points from the same sample are connected by grey dashed lines. *P*-values were calculated using one-sided paired *t*-tests. *, *P*-value < 0.05; **, *P*-value < 0.01; ***, *P*-value < 0.001. “Unidentified” indicates cells not assigned to any TC or CN. “Unmatched” denotes TCs that could not be aligned with manual annotations.

Response Fig. 21 (extracted from Supplementary Figs. 5 and 6). Performance comparison of HRCHY-CytoCommunity and NeST on the mouse ICH Stereo-seq dataset. (a) Spatial maps showing the

distribution of cell types (left) and molecularly defined regions (right). **(b)** Fine-grained CNs identified by HRCHY-CytoCommunity (left) and NeST (right). **(c)** Coarse-grained TCs identified by HRCHY-CytoCommunity (left) and NeST (right). **(d)** AMI and Macro-F1 scores calculated using molecularly defined regions. Each point corresponds to the performance on an individual sample, with horizontal bars indicating the mean performance across all samples. Points from the same samples are connected by grey dashed lines. *P*-values were calculated using a one-sided paired *t*-test. *, *P*-value < 0.05; **, *P*-value < 0.01; ***, *P*-value < 0.001. “Unidentified” indicates cells not assigned to any TC or CN. “Unmatched” denotes TCs that could not be aligned with manual annotations. L2/3, cortical layer 2 and cortical layer 3; L4, cortical layer 4; L5, cortical layer 5; L6, cortical layer 6; Lesion, Lesion region; injured_STR, injured striatum region; injured_cortex, injured cortex region; injured_PAL, injured pallidum region; DG, dentate gyrus; FT, fiber tracts; HY, hypothalamus; OLF, olfactory areas; MH, medial habenula; CA, Ammon’s horn; STRd, striatum dorsal region; STRv, striatum ventral region; TH, thalamus; VS, ventricular systems; isl, islm, islands of Calleja and major island of Calleja; PAL, pallidum; RT, reticular nucleus of the thalamus; CASp, field Ammon’s horn, pyramidal layer; LSX, lateral septal complex; sAMY, striatum-like amygdala nuclei; CTXsp, cortical subplate.

R3C3

3. Limitations of Using Cell-Type Annotation Only: The exclusive reliance on cell-type labels poses a major limitation. High-quality annotations may not always be available, and annotation resolution varies by dataset. Furthermore, critical spatial structures may be defined not by cell types but by cell-cell interactions, gradients, or activity states. These limitations should be acknowledged, and potential solutions or future directions discussed. For instance, the author could compare the results of two different levels of annotation (finer vs coarse).

We thank the reviewer for raising this valuable point. Indeed, the current implementation of HRCHY-CytoCommunity relies exclusively on cell-type annotations as input features, which raises concerns about its dependence on annotation quality and resolution that may vary across datasets. This design choice was motivated by the broad availability of cell-type annotations across diverse spatial omics datasets, which facilitates the generalizability of HRCHY-CytoCommunity across datasets. Moreover, it enhances the interpretability of the identified hierarchical tissue structures.

To address this concern, we systematically evaluated the robustness of HRCHY-CytoCommunity to both cell-type **label inaccuracies** and **annotation resolution variations** using two datasets with complex cell-type compositions (i.e., the mouse hypothalamic preoptic region MERFISH dataset and the human TNBC MIBI-TOF dataset), as detailed below:

1. **Robustness to cell-type label inaccuracies:** We used the hierarchical tissue structures identified from unperturbed cell-type annotations as the reference. Cell-type labels were then randomly perturbed by 10%–50%, and the accuracy and consistency of the resulting hierarchical structures were compared with the reference. The Fowlkes-Mallows Index

(FMI) was used to quantify the consistency between perturbed and reference structures across different noise levels.

- a) In the MERFISH dataset, the identified hierarchical structures remained highly consistent until the label error rate exceeded 40% (Response Fig. 22a; **average FMI > 0.5 when label error rate = 40%**). The accuracy of the fine-grained CNs (measured by AMI and Macro-F1 scores) showed only a moderate decline, with performance largely preserved under reasonable noise levels (Response Figs. 22b and c)
 - b) In the MIBI-TOF dataset, hierarchical structures remained highly consistent across perturbation rates (Response Fig. 23a; **average FMI > 0.65 when label error rate = 50%**). The accuracy of the identified coarse-grained TCs (measured by the proportion of neoplastic and immune cells correctly assigned to their corresponding TCs) was largely preserved under reasonable levels of annotation noise, with only a moderate decrease as the label error rate increased (Response Fig. 23b).
2. **Robustness to cell-type annotation resolutions:** We further assessed robustness to annotation resolution by comparing resulting structures obtained under different levels of cell-type granularity (i.e., 15 vs. 9 cell types in MERFISH, 17 vs. 11 cell types in TNBC MIBI-TOF). The FMI was used to quantify the similarity between identified structures. Specifically,
- a) HRCHY-CytoCommunity consistently recovered stable hierarchical tissue structures across annotation resolutions, achieving **average FMI scores of 0.673** in the MERFISH dataset (Response Figs. 22d-f) and **0.590** in the MIBI-TOF dataset across samples (Response Figs. 23c-e).

In summary, these analyses demonstrate that **HRCHY-CytoCommunity is strongly robust to both annotation noise and annotation resolution variations**, alleviating concerns regarding strong dependence on high-quality annotations and specific annotation resolution.

Furthermore, we explicitly **acknowledge the limitation** that this design may not capture tissue structures defined by continuous gradients, transient states, or functional interactions that are not well represented by discrete cell-type labels. We have added this limitation to the **Discussion** section and outlined future directions.

The corresponding discussion of the limitation of HRCHY-CytoCommunity is shown below.

“Moreover, while the use of discrete cell-type annotations enhances the method’s generalizability across platforms, it may not fully capture structures defined by continuous molecular gradients, transient states, or cell-cell communication signals. Future versions could incorporate multi-view learning to integrate such continuous features. In the meantime, methods like NicheCompass²⁰ (focused on cellular communication-driven structures) and ONTraC⁵⁵ (designed for spatially continuous niche trajectories) can complement HRCHY-CytoCommunity to provide a more comprehensive perspective on tissue organization.”

Full details of the robustness assessment are provided in Supplementary Note 5 and Supplementary Figs. 19 and 20.

Response Fig. 22 (Supplementary Fig. 19). Robustness assessment of HRCHY-CytoCommunity to label inaccuracies and annotation resolution variations using the mouse hypothalamic preoptic region MERFISH dataset. (a) Hierarchical tissue structure similarity under increasing cell-type label perturbation rates (10%-50%), quantified by Fowlkes-Mallows Index (FMI)-based robustness score. Higher values indicate greater structural consistency with the unperturbed reference. (b-c) Fine-grained CN identification accuracy under label perturbations, measured by Adjusted Mutual Information (AMI) (b) and Macro-F1 scores (c). Each point represents performance on one sample, with horizontal bars represent the mean performance across $n = 5$ samples. Performances (points) on the same sample are connected by grey dashed lines. Mean values under each label error rate are connected by blue solid lines. Shaded areas denote standard deviation. (d) Single-cell spatial maps showing the distribution of 15 cell types (left), coarse-grained TCs (middle) and fine-grained CNs (right) identified by HRCHY-CytoCommunity based on this high-resolution annotation. (e) Single-cell spatial maps showing the distribution of 9 cell types (left), coarse-grained TCs (middle) and fine-grained CNs (right) identified by HRCHY-CytoCommunity based on this low-resolution annotation. (f) Hierarchical tissue structure similarity between different cell-type annotation resolutions, quantified by FMI-based robustness score. The red dashed line indicates the mean similarity (0.673) relative to the 15-cell-type annotation-based reference.

Response Fig. 23 (Supplementary Fig. 20). Robustness assessment of HRCHY-CytoCommunity to label inaccuracies and annotation resolution variations using the human TNBC MIBI-TOF dataset. (a) Hierarchical tissue structure similarity under increasing cell-type label perturbation rates (10%-50%),

quantified by Fowlkes-Mallows Index (FMI)-based robustness score. Higher values indicate greater structural consistency with the unperturbed reference. **(b)** Coarse-grained TC identification accuracy under label perturbations, measured by proportion of neoplastic and immune cells correctly assigned to their corresponding TCs. Each point represents performance on one compartmentalized tumor sample, with horizontal bars represent the mean performance across $n = 15$ samples. Performances (points) on the same sample are connected by grey dashed lines. Mean values under each label error rate are connected by blue solid lines. Shaded areas denote standard deviation. **(c)** Single-cell spatial maps showing the distribution of 17 cell types (left), coarse-grained TCs (middle) and fine-grained CNs (right) identified by HRCHY-CytoCommunity based on this high-resolution annotation. **(d)** Single-cell spatial maps showing the distribution of 11 cell types (left), coarse-grained TCs (middle) and fine-grained CNs (right) identified by HRCHY-CytoCommunity based on this low-resolution annotation. **(e)** Hierarchical tissue structure similarity between different cell-type annotation resolutions, quantified by FMI-based robustness score. The red dashed line indicates the mean similarity (0.590) relative to the 17-cell-type annotation-based reference.

R3C4

4. Comparison with Additional Methods: The current comparisons omit several state-of-the-art spatial domain identification tools, including SpaGCN, NicheCompass, CellCharter, Gaston-Mix among others. While these do not explicitly identify hierarchical domains, many can be run iteratively or parameter-tuned to extract structures at different scales. Including such comparisons would provide a fairer performance benchmark.

We appreciate the reviewer for the constructive suggestion. Following the suggestion from Reviewers 2 and 3, we have benchmarked HRCHY-CytoCommunity against six state-of-the-art non-hierarchical spatial domain detection methods, including **SpaGCN**, **NicheCompass**, **CellCharter**, **SpaSEG**, **Giotto Suite**, and **GraphST**, in the revised manuscript. Since these methods do not explicitly identify hierarchical tissue structures, we applied them separately at coarse-grained TC and fine-grained CN resolutions. Performances were evaluated using Adjusted Mutual Information (AMI) and Macro-F1 scores across two datasets with ground-truth TCs and two datasets with ground-truth CNs.

Our results demonstrated that **HRCHY-CytoCommunity consistently achieved superior or competitive performance than non-hierarchical methods**. Specifically:

1. For coarse-grained TC identification:

- a) On the **mouse spleen CODEX dataset (imaging-based)**, HRCHY-CytoCommunity was the only method that consistently resolved the biologically meaningful lymphoid compartment containing B-zone, periarteriolar lymphoid sheath, and marginal zone across all samples (Response Fig. 24b, left column), whereas non-hierarchical methods largely failed to capture this hierarchical structure (GraphST succeeded only in one sample). Quantitatively, HRCHY-CytoCommunity achieved significantly higher AMI

and Macro-F1 scores than most non-hierarchical methods (one-sided paired *t*-test *P*-values < 0.05), while exhibiting performance comparable to GraphST (Response Fig. 24d).

- b) On the **human colorectal cancer (CRC) Visium HD dataset (sequencing-based)**, HRCHY-CytoCommunity was the only method that accurately distinguished tumor and normal compartments in the P2CRC sample, while all non-hierarchical methods failed to do so. Similarly, in the remaining samples, these two compartments were also clearly delineated, though with moderate inclusion of intestinal epithelial-enriched regions within the tumor compartment (Response Fig. 25b, left column). Quantitatively, HRCHY-CytoCommunity achieved the highest average AMI and Macro-F1 scores among all representative non-hierarchical methods (Response Fig. 25d).

2. For fine-grained CN identification:

- a) On the **mouse hypothalamic preoptic region MERFISH dataset (imaging-based)**, HRCHY-CytoCommunity successfully recovered CNs consistent with manually delineated nuclei with clear boundaries (Response Fig. 26b, left column), whereas other methods identified intermixed CNs and failed to capture smaller regions like PS and PaAP in most samples. Quantitatively, HRCHY-CytoCommunity significantly outperformed all non-hierarchical methods in both AMI and Macro-F1 metrics (one-sided paired *t*-test, *p* < 0.05), with the exception of CellCharter on the AMI score and GraphST on the Macro-F1 score, where the differences were not statistically significant (Response Fig. 26d)
- b) On the **mouse intracerebral hemorrhage (ICH) Stereo-seq dataset (sequencing-based)**, HRCHY-CytoCommunity achieved high accuracy in identifying fine-grained CNs, particularly excelling in detecting small-scale CNs such as olfactory areas (OLF), meninges, and pallidum (PAL) (Response Fig. 27b, left column). Quantitatively, HRCHY-CytoCommunity achieved significantly higher AMI and Macro-F1 scores than most non-hierarchical methods (one-sided paired *t*-test, *P*-values < 0.001), with performance comparable to NicheCompass (Response Fig. 27d)

Taken together, these results indicate that HRCHY-CytoCommunity is not only quantitatively competitive but also qualitatively superior, as it **identifies biologically meaningful structures missed by non-hierarchical methods**. This highlights that the **explicit hierarchical framework yields more accurate, stable, and interpretable multi-scale tissue organization** across diverse technologies.

Full implementation details of these non-hierarchical spatial domain detection methods are provided in the **Methods** section under the title “**Running of published methods**”. Full details of the performance evaluation are provided in the **Results** section under the title: “**Performance evaluation on coarse-grained TC identification**” and “**Performance evaluation on fine-grained CN identification**”, as well as in **Supplementary Notes 1 and 2**.

Response Fig. 24 (extracted from Fig. 2) Performance evaluation of HRCHY-CytoCommunity on the mouse spleen CODEX dataset. (a) Spatial maps of healthy mouse spleen samples generated by CODEX technology. Cells are colored by cell types (left) and manually annotated compartments (right), which serve as the ground truth for coarse-grained TCs. **(b)** Coarse-grained TCs identified by HRCHY-CytoCommunity (left) and NeST (right). **(c)** Coarse-grained TCs identified by non-hierarchical methods, including CellCharter, NicheCompass, GraphST, and SpaGCN. **(d)** Adjusted Mutual Information (AMI) and Macro-F1 scores calculated using manual compartment annotations. Each point corresponds to the performance on an individual sample, with horizontal bars indicating the mean performance across all samples. Points from the same sample are connected by grey dashed lines. *P*-values were calculated using one-sided paired *t*-tests. *, *P*-value < 0.05; **, *P*-value < 0.01; ***, *P*-value < 0.001. “Unidentified” indicates cells not assigned to any TC. “Unmatched” denotes TCs that could not be aligned with manual annotations.

Response Fig. 25 (extracted from Fig. 2). Performance evaluation of HRCHY-CytoCommunity on the human CRC Visium HD dataset. (a) Spatial maps of human CRC samples (8 μm bins) generated by Visium HD technology. Bins are colored by cell types (left) and manually annotated coarse-grained structures (right). **(b)** Coarse-grained TCs identified by HRCHY-CytoCommunity (left) and NeST (right). **(c)** Coarse-grained TCs identified by non-hierarchical methods, including CellCharter, NicheCompass, and SpaSEG. **(d)** AMI and Macro-F1 scores calculated using manual compartment annotations. Each point corresponds to the performance on an individual sample, with horizontal bars indicating the mean performance across all samples. Points from the same sample are connected by grey dashed lines. P -values were calculated using one-sided paired t -tests. *, P -value < 0.05; **, P -value < 0.01; ***, P -value < 0.001. “Unidentified” indicates cells not assigned to any TC. “Unmatched” denotes TCs that could not be aligned with manual annotations.

Response Fig. 26 (Fig. 3). Performance evaluation of HRCHY-CytoCommunity on the mouse hypothalamic preoptic region MERFISH dataset. (a) Spatial maps of mouse hypothalamic preoptic region samples generated by MERFISH technology. Cells are colored by cell types (left) and manually annotated hypothalamic nuclei (right), which were generated based on the outlines of hypothalamic nuclei (middle) and serve as the ground truth for fine-grained CNs. **(b)** Fine-grained CNs identified by HRCHY-CytoCommunity (left) and NeST (right). **(c)** Fine-grained CNs identified by non-hierarchical methods, including CellCharter, NicheCompass, GraphST, SpaGCN, SpaSEG, and Giotto Suite. **(d)** Adjusted Mutual Information (AMI) and Macro-F1 scores calculated using manually annotated hypothalamic nuclei. Each point corresponds to the performance on an individual sample, with horizontal bars indicating the mean performance across all samples. Points from the same sample are connected by grey dashed lines. *P*-values were calculated using one-sided paired *t*-tests. *, *P*-value < 0.05; **, *P*-value < 0.01; ***, *P*-value < 0.001. “Unidentified” indicates cells not assigned to any CN.

Response Fig. 27 (Supplementary Fig. 5). Performance evaluation of HRCHY-CytoCommunity on the mouse ICH Stereo-seq dataset. (a) Spatial maps showing the distribution of cell types (left) and molecularly defined regions (right). (b) Fine-grained CNs identified by HRCHY-CytoCommunity (left) and NeST (right).

(c) Fine-grained CNs identified by non-hierarchical tissue structure identification methods, including CellCharter, NicheCompass, GraphST, and SpaGCN. “Unidentified” indicates cells not assigned to any CN. (d) AMI and Macro-F1 scores calculated using molecularly defined regions. Each point corresponds to the performance on an individual sample, with horizontal bars indicating the mean performance across all samples. Points from the same samples are connected by grey dashed lines. P -values were calculated using a one-sided paired t -test. *, P -value < 0.05; **, P -value < 0.01; ***, P -value < 0.001. L2/3, cortical layer 2 and cortical layer 3; L4, cortical layer 4; L5, cortical layer 5; L6, cortical layer 6; Lesion, Lesion region; injured_STR, injured striatum region; injured_cortex, injured cortex region; injured_PAL, injured pallidum region; DG, dentate gyrus; FT, fiber tracts; HY, hypothalamus; OLF, olfactory areas; MH, medial habenula; CA, Ammon’s horn; STRd, striatum dorsal region; STRv, striatum ventral region; TH, thalamus; VS, ventricular systems; isl, islm, islands of Calleja and major island of Calleja; PAL, pallidum; RT, reticular nucleus of the thalamus; CAsp, field Ammon’s horn, pyramidal layer; LSX, lateral septal complex; sAMY, striatum-like amygdala nuclei; CTXsp, cortical subplate.

R3C5

5. Loss Threshold Parameter (β): The parameter β is introduced as a threshold for selecting structures based on graph loss but is not reflected in the mathematical formulations (Equations 7–9). The manuscript should be updated to incorporate β directly into the equations, with a formal explanation of its role and its effect on structure selection.

We thank the reviewer for this valuable feedback. In the original manuscript, the loss threshold parameter β was introduced as a heuristic strategy to mitigate the cluster collapse problem (i.e., all cells being assigned to a single cluster). In the revised version, we have **replaced this heuristic approach with a trainable entropy-based balance regularization term** to solve this issue, which is now directly incorporated into the optimization objective. The formulation is defined as:

$$L_{balance} = 1 - \frac{1}{M} \sum_{m=1}^M \frac{H_m^{(2)}}{\log c_2}$$

where M is the number of perturbations, c_2 is the number of coarse-grained TCs. $H_m^{(2)}$ is the entropy of the cluster assignment distribution for the m -th perturbation. The $H_m^{(2)}$ is computed as:

$$H_m^{(2)} = - \sum_{k=1}^{c_2} s_{m,k}^{(2)} \log s_{m,k}^{(2)}$$

where $s_{m,k}^{(2)}$ represents the assignment probability of coarse-grained TC k in the m -th perturbation. This regularization term encourages balanced cluster assignments by maximizing the entropy of the predicted distributions across clusters, thereby effectively preventing collapse of the coarse-grained TC assignments.

Accordingly, the original β threshold is no longer used. Its role has been fully replaced by the entropy-based regularization, which provides a **more rigorous and differentiable solution** to the same problem.

Full details of this entropy-based balance regularization term are provided in the **Methods** section under the title “**Robust hierarchical tissue structure identification**” (Equations 17-18).

R3C6

6. Details of Model Parameters: Please add a table summarizing the hyperparameters used for each dataset: number of GNN layers, MLP layers, values for α , β , and pruning thresholds. Without this information, reproducibility and understanding of model performance across datasets are limited.

We appreciate the reviewer’s constructive suggestion. In the revised manuscript, we have added a comprehensive table (**Supplementary Table 2**) summarizing the hyperparameters used in all datasets, and also described the key information below for clarity. Notably, in the revised version, the original fixed α that balanced the unsupervised losses of the fine-grained CN and the coarse-grained TC identification has been replaced by **an adaptive α scheduling strategy** (Methods), the previous orthogonality loss threshold β has been replaced by a **trainable balance regularization term** (Methods), and the fixed edge-pruning threshold has been replaced by an **adaptive edge-pruning strategy** (Methods).

Accordingly, the following parameters are included in the table:

- **#TC** and **#CN**, the numbers of TCs and CNs, respectively;
- **#K**, the number of neighbors K in the KNN graph (cell-cell proximity graph) construction;
- **#GNN layers** and **#MLP layers**, the numbers of the graph neural network and multilayer perceptron modules, respectively;
- **#Epochs**, the number of training epochs;
- **λ_{consis}** and **λ_{balance}** , the weights of consistency and balance regularization terms, respectively;

- **#Perturbations**, the number of perturbations on the cell-cell proximity graph;
- **DropNode rate**, the fraction of nodes whose features are masked during consistency regularization training;
- **Learning rate**, the learning rate of the training process.

Dataset	#TC	#CN	#K	#GNN layers	#MLP layers	#Epochs	λ_{consis}	$\lambda_{balance}$	#Perturbations	Drop Node rate	Learning rate
Human CRC Visium HD dataset	2	10	20	2	2	1500	1	1	5	0.5	0.0001
Human Breast Cancer Visium dataset	2	9	20								
Mouse ICH Stereo-seq dataset	3	20	50								
Mouse spleen CODEX dataset	2	7	50								
Mouse hippocampus Slide-seq V2 dataset	3	11	50								
Mouse hypothalamic preoptic region MERFISH dataset	2	12	100								
Human TNBC MIBI-TOF dataset	2	14	50								

Human Breast Cancer IMC dataset	2	7	20								
---	---	----	--	--	--	--	--	--	--	--

Supplementary Table 2. Hyperparameter settings of HRCHY-CytoCommunity used in all datasets.

This table details the configuration of the HRCHY-CytoCommunity model for each dataset, including the number of tissue compartments (TCs) and cellular neighborhoods (CNs), the value of K in the K -nearest neighbor graph, the number of GNN and MLP layers, total training epochs, regularization weights (α , λ_{consis} , and λ_{balance}), the number of perturbations on the cell-cell proximity graph, DropNode rate, and learning rate.

R3C7

7. Sensitivity Analysis: The study would benefit from conducting and reporting a sensitivity analysis for α , β , and edge pruning on at least one image-based and one sequencing-based dataset. This is critical to demonstrate model stability and the impact of key hyperparameters on performance.

We thank the reviewer for this valuable suggestion. In the revised manuscript, we have conducted comprehensive sensitivity analyses of key hyperparameters in HRCHY-CytoCommunity on both simulated and real datasets. Specifically, for coarse-grained TC identification, we performed sensitivity analysis on the **mouse spleen CODEX dataset (imaging-based)** and the **human CRC Visium HD dataset (sequencing-based)**. For fine-grained CN identification, we performed the analysis on the **mouse hypothalamic preoptic region MERFISH dataset (imaging-based)** and the **mouse ICH Stereo-seq dataset (sequencing-based)**.

As mentioned in **R3C6**, in the revised manuscript, the original fixed α , β , and edge pruning threshold have been replaced with an adaptive α scheduling strategy, a trainable entropy-based balance regularization term, and an adaptive edge-pruning strategy, respectively. Therefore, our sensitivity analysis focused on five key hyperparameters: (1) the **DropNode rate** (fraction of nodes whose features are masked during consistency regularization training), (2) **λ_{consis}** (weight of the consistency regularization term), (3) **λ_{balance}** (weight of the balance regularization term), (4) the **number of neighbors K** in the KNN graph (cell-cell proximity graph) construction, and (5) the **number of perturbations** used for generating perturbed cell-cell proximity graphs.

The results showed that:

1. On the simulated dataset, HRCHY-CytoCommunity exhibited stable performance across a wide range of hyperparameter values, with only a slight decline in accuracy as the DropNode rate increased (Response Fig. 28).

2. On real datasets,

- a) For coarse-grained TC identification, performance was moderately sensitive to λ_{consis} and λ_{balance} , reflecting their roles in maintaining structural consistency and preventing cluster collapse. Excessively high DropNode rates degraded performance, likely due to loss of informative features. In contrast, the adaptive edge-pruning strategy ensured robustness to changes in the number of K neighbors. Increasing the number of perturbations did not substantially alter results (Response Fig. 29).
- b) For fine-grained CN identification, performance remained highly stable across all hyperparameter settings (Response Fig. 30).

Taken together, these results demonstrate that HRCHY-CytoCommunity maintains **stable and accurate performance without requiring extensive hyperparameter tuning**.

Full details of the sensitivity analysis are provided in **Supplementary Note 4** and **Supplementary Figs. 16-18**.

Response Fig. 28 (Supplementary Fig. 16). Sensitivity analysis of hyperparameters in HRCHY-CytoCommunity using the simulated dataset. Hierarchical tissue structure identification under different model configurations. Each configuration was run 10 times with different random seeds. Boxplot elements are defined as: center line, median; box limits, upper and lower quartiles; whiskers, $1.5 \times$ interquartile range. Performances are quantified by Adjusted Mutual Information (AMI) and Macro-F1 scores. Each point corresponds to the performance of one run. This sensitivity analysis includes five hyperparameters: (1) the DropNode rate (fraction of nodes whose features are masked during consistency regularization training), (2) λ_{consis} (weight of the consistency regularization term), (3) λ_{balance} (weight of the balance regularization term), (4) the number of neighbors K in the KNN graph (cell-cell proximity graph) construction, and (5) the number of perturbations used for generating perturbed cell-cell proximity graphs.

Response Fig. 29 (Supplementary Fig. 17). Sensitivity analysis of hyperparameters in HRCHY-CytoCommunity on coarse-grained TC identification using real spatial omics datasets. Hierarchical tissue structure identification under different model configurations using (a) the mouse spleen CODEX dataset (imaging-based) and (b) the human colorectal cancer (CRC) Visium HD dataset (sequencing-based). Each run was run 10 times with different random seeds. Boxplot elements are defined as: center line, median; box limits, upper and lower quartiles; whiskers, $1.5\times$ interquartile range. Performances are quantified by Adjusted Mutual Information (AMI) and Macro-F1 scores. Each point corresponds to the performance of one run. This sensitivity analysis includes five hyperparameters: (1) the DropNode rate (fraction of nodes whose features are masked during consistency regularization training), (2) λ_{consis} (weight of the consistency regularization term), (3) λ_{balance} (weight of the balance regularization term), (4) the number of neighbors K in the KNN graph (cell-cell proximity graph) construction, and (5) the number of perturbations used for generating perturbed cell-cell proximity graphs.

Response Fig. 30 (Supplementary Fig. 18). Sensitivity analysis of hyperparameters in HRCHY-CytoCommunity on fine-grained CN identification using real spatial omics datasets. Hierarchical tissue structure identification under different model configurations using (a) the mouse hypothalamic preoptic region MERFISH dataset (imaging-based) and (b) the mouse intracerebral hemorrhage (ICH) Stereo-seq dataset (sequencing-based). Each configuration was run 10 times with different random seeds. Boxplot elements are defined as: center line, median; box limits, upper and lower quartiles; whiskers, $1.5\times$ interquartile range. Performances are quantified by Adjusted Mutual Information (AMI) and Macro-F1 scores. Each point corresponds to the performance of one run. This sensitivity analysis includes five hyperparameters: (1) the DropNode rate (fraction of nodes whose features are masked during consistency regularization training), (2) λ_{consis} (weight of the consistency regularization term), (3) λ_{balance} (weight of the balance regularization term), (4) the number of neighbors K in the KNN graph (cell-cell proximity

graph) construction, and (5) the number of perturbations used for generating perturbed cell-cell proximity graphs.

R3C8

8. Choice and Estimation of Number of Structures (c_1, c_2): The model requires the number of fine- and coarse-grained structures to be pre-defined. The manuscript should clarify whether these are learned or fixed, and how users should choose them. If large values are set, how does the model handle oversegmentation? A discussion or heuristic strategy would be helpful.

We appreciate the reviewer's constructive suggestions. In the revised manuscript, inspired by the state-of-the-art spatial domain detection method CellCharter¹, we have provided a **heuristic strategy to guide users in choosing the numbers of structures (c_1, c_2)**.

In this approach, users were first required to define a candidate search range for (c_1, c_2) and the number of repeated runs R . For each candidate pair (c_1, c_2), the model was trained R times independently. The stability of the resulting hierarchical structures (i.e., CNs and TCs) was then quantified using the FMI computed across runs. Specifically, the FMI was computed separately for the identified CNs and TCs across all pairs of runs for a given (c_1, c_2) configuration. This yielded a set of CN-level FMIs and a set of TC-level FMIs. For each level, the median FMI across all run pairs was computed to summarize the cluster stability at that level. The final stability score for a (c_1, c_2) configuration was defined as the average of the CN-level and TC-level median FMIs. The optimal (c_1, c_2) pair was selected as the one that yields the highest stability score. Finally, among the R models trained under the optimal (c_1, c_2) configuration, the model achieving the lowest training loss was chosen to output the final hierarchical tissue structures.

This procedure also addresses the reviewer's concern regarding potential over-segmentation when large values of (c_1, c_2) are used:

- When the number of clusters is too large, the clustering results tend to become unstable across repeated runs, which is reflected by lower FMI stability.
- Conversely, when the chosen (c_1, c_2) properly matches the underlying tissue organization, the model yields reproducible partitions with high FMI stability.

In summary, instead of requiring users to predefine (c_1, c_2), we introduce a **data-driven heuristic strategy** to determine appropriate values of (c_1, c_2) and prevent over-segmentation through stability analysis. This ensures that HRCHY-CytoCommunity can identify robust and stable hierarchical tissue structures.

Full details are provided in the **Methods** section under the title “**Determination of optimal numbers of hierarchical tissue structures based on cluster stability**”.

R3C9

9. Interpretability of Nested Structures: The manuscript suggests that assigning each fine-grained structure to a single coarse-grained one improves interpretability. However, biological systems often contain overlapping or ambiguous structures. For example, cells in a boundary region may legitimately belong to multiple domains. Instead of enforcing strict assignments, the model might benefit from allowing flexible memberships or highlighting ambiguity.

We thank the reviewer for this insightful comment. While HRCHY-CytoCommunity currently enforces strict hierarchical assignments for clear interpretability and consistency in downstream analyses, we fully agree that cells located in boundary regions may exhibit ambiguity and overlapping memberships in hierarchical structures. To address this concern, we provided an interface that allows users to visualize regions with potentially ambiguous hierarchical memberships. Specifically, we introduced an uncertainty measure based on the entropy of the soft assignment distribution for each cell. As shown in Response Fig. 31, cells with higher entropy scores are frequently located at CN boundaries or transition zones, reflecting potential ambiguity in structural assignments. This interface enables users to assess assignment uncertainty and highlight areas where nested or overlapping structures may exist. In future work, we plan to extend the framework by explicitly incorporating this uncertainty into probabilistic or soft-membership hierarchical models.

Full details of the ambiguous structure analysis are provided in Supplementary Note 9.

Response Fig. 31 (Supplementary Fig. 24). Ambiguous structures in the mouse Slide-seq V2 dataset. (a) Fine-grained CNs identified by HRCHY-CytoCommunity. (b) Uncertainty score distribution, with values near 0 indicating high confidence in tissue structure assignment and values near 1 reflecting high ambiguity.

R3C10

10. Survival Analysis Details: The survival analysis lacks details on the inclusion of clinical covariates. Without covariate adjustment, survival differences may be confounded. If covariates (e.g., age, sex, tumor grade) were not included, either re-run the analysis with appropriate adjustments or justify why this analysis is still valid.

We thank the reviewer for bringing this critical detail up and fully agree that covariate adjustment is essential for survival analysis. Accordingly, in the revised manuscript, we have replaced all previous results derived from log-rank tests with those obtained from **multivariable Cox proportional hazards models**.

In this analysis, we included three well-established prognostic covariates for breast cancer (**age, tumor grade, and tumor size**). Sex was not included as a covariate because all patients in this dataset were female. The results were reported as hazard ratios (HRs) and *P*-values (e.g., HR = 2.2, *P*-value = 0.031).

Overall, our results suggest the potential of hierarchical tissue structures identified by HRCHY-CytoCommunity serving as informative features for patient stratification, and reveal subgroups with independent prognostic significance.

Full implementation details of the survival analysis are provided in the **Methods** section under the title “**Survival analysis**”, and the corresponding results in the human breast cancer IMC dataset are provided in the **Results** section under the title “**Hierarchical tissue structure-based stratification of breast cancer patients**”.

R3C11

11. Validation with Simulated Data: Consider including a synthetic dataset with known hierarchical ground truth to benchmark model accuracy and interpretability under controlled conditions.

We thank the reviewer for this constructive suggestion. Following the suggestion, we constructed a **synthetic dataset with a known hierarchical ground truth** to benchmark HRCHY-CytoCommunity under controlled conditions. Specifically, inspired by the human TNBC MIBI-TOF dataset, we designed two TCs, one dominated by immune cells (cell type 1) and the other by tumor cells (cell type 8). Each TC contained multiple CNs with distinct compositional patterns, yielding a ground-truth structure of two TCs and eight CNs. The cell-type composition profiles for CNs in simulation data has been provided in **Supplementary Table 4**.

Our results showed that HRCHY-CytoCommunity accurately recovered the ground-truth hierarchical structures, achieving **AMI = 0.956 and Macro-F1 score = 0.994 for TCs, and AMI = 0.914 and Macro-F1 score = 0.959 for CNs** (Response Fig. 32). We further leveraged this dataset to perform **ablation and sensitivity analyses**. The ablation experiments confirmed that the consistency and balance regularization terms are most effective when applied together and also validated the effectiveness of the adaptive edge-pruning strategy in coarse-grained TC identification (Response Fig. 33). The sensitivity analysis demonstrated that model performance remains stable across a wide range of hyperparameter choices (Response Fig. 28).

Taken together, the experiments using the synthetic dataset verify that HRCHY-CytoCommunity can **accurately identify hierarchical tissue structures under controlled conditions** and further highlight the **effectiveness and robustness of the proposed design**.

Full details of the construction of the simulated dataset are provided in **Supplementary Note 7**, under the title “**Simulated dataset**”, and the results of HRCHY-CytoCommunity on this dataset are provided in **Supplementary Fig. 23**. Full details of the ablation and sensitivity analyses using this dataset are provided in **Supplementary Notes 3 and 4**.

Response Fig. 32 (Supplementary Fig. 23). Performance evaluation of HRCHY-CytoCommunity using the simulated dataset. (a) Spatial distribution of eight cell types in the simulated dataset. (b) Ground truth of the coarse-grained TCs. (c) Coarse-grained TCs identified by HRCHY-CytoCommunity (AMI = 0.956, Macro-F1 score = 0.994). (d) Ground truth of the fine-grained CNs. (e) Fine-grained CNs identified by HRCHY-CytoCommunity (AMI = 0.914, Macro-F1 score= 0.959).

Response Fig. 33 (Supplementary Fig. 13). Ablation study of key components in HRCHY-CytoCommunity using the simulated dataset. Hierarchical tissue structure identification under different model configurations. Each configuration was run 10 times with different random seeds. Boxplot elements are defined as: center line, median; box limits, upper and lower quartiles; whiskers, $1.5 \times$ interquartile range. Performances are quantified by Adjusted Mutual Information (AMI) and Macro-F1 scores. Each point corresponds to the performance of one run. Ablation studies were performed on (a) consistency and balance regularization terms, (b) the adaptive edge-pruning strategy, and (c) the adaptive α scheduling strategy.

R3C12

12. Multiple Sample Integration: The method currently runs HRCHY-CytoCommunity independently per sample, followed by clustering enrichment profiles. This strategy might not

fully capture shared spatial structure across samples. Consider discussing or exploring alternatives, such as joint embedding approaches or graph alignment techniques.

We thank the reviewer for this insightful comment. We agree that joint embedding or graph alignment strategies could better capture shared spatial structures across samples. To explore this, we implemented a straightforward multi-sample integration by merging individual sample-specific KNN graphs into a single large graph. In this integrated graph, each connected component corresponded to a sample-specific KNN graph. We then applied HRCHY-CytoCommunity directly to this integrated multi-sample graph.

The model consistently identified symmetric CNs that were highly concordant with the manually delineated nuclei (Response Fig. 34c). Quantitatively, the multi-sample strategy yielded a slightly lower AMI compared with the single-sample setting (two-sided paired t -test, P -value = 0.034; Response Fig. 34d, left panel) and a comparable Macro-F1 score (two-sided paired t -test, P -value = 0.55; Response Fig. 34d, right panel). These results suggested that this multi-sample integration is feasible, although sample-specific heterogeneity may slightly reduce accuracy. However, given that HRCHY-CytoCommunity is trained on the entire graph, scaling this integration to large cohorts would increase memory and computational requirements, thereby limiting its practicality for large-scale studies. Moreover, in the TNBC MIBI-TOF and the breast cancer IMC datasets, we have demonstrated that clustering cell-type enrichment profiles across samples can effectively capture both conserved and sample-specific TCs and CNs.

Therefore, we retained the single-sample implementation and used the clustering-based approach to perform multi-sample integration in this study, which offers a favorable trade-off between robustness and scalability. In future work, we plan to explore graph sampling and alignment-based techniques to develop a scalable cross-sample extension of HRCHY-CytoCommunity. This discussion has been incorporated into the revised **Discussion** section.

“Additionally, future extensions could incorporate graph sampling or alignment techniques to improve cross-sample integration within an end-to-end analytical workflow.”

Response Fig. 34. Performance evaluation of HRCHY-CytoCommunity with multi-sample integration on the mouse hypothalamic preoptic region MERFISH dataset. (a) Spatial maps of mouse hypothalamic preoptic region samples generated by MERFISH technology. Cells are colored by cell types (left) and

manually annotated hypothalamic nuclei (right), which were generated based on the outlines of hypothalamic nuclei (middle) and serve as the ground truth for fine-grained CNs. **(b)** Hierarchical tissue structure identified by the single-sample version of HRCHY-CytoCommunity, coarse-grained TCs (left) and fine-grained CNs (right). **(c)** Hierarchical tissue structure identified by the multi-sample version of HRCHY-CytoCommunity, coarse-grained TCs (left) and fine-grained CNs (right). **(d)** Adjusted Mutual Information (AMI) and Macro-F1 scores calculated using manually annotated hypothalamic nuclei. Each point corresponds to the performance on an individual sample, with horizontal bars indicating the mean performance across all samples. Points from the same sample are connected by grey dashed lines. *P*-values were calculated using one-sided paired *t*-tests. *, *P*-value < 0.05; **, *P*-value < 0.01; ***, *P*-value < 0.001.

Minor Comments

R3C1

1. Please add a legend to Figure 3a to clarify the visualized elements and improve readability.

We thank the reviewer for this helpful suggestion to improve the readability of our manuscript. In the revised version, we have added a clear cell-type legend below **Figure 3a** to clarify the visualized elements and enhance readability.

R3C2

2. The rationale for selecting the “3TC” annotation in the bregma mouse brain sample is not provided. Please justify its use or clarify its biological relevance.

We thank the reviewer for pointing this out. In the original manuscript, we empirically set the number of coarse-grained TCs in the mouse hypothalamic preoptic region MERFISH dataset to three. In the revised version, we employed the cluster stability-based heuristic strategy to determine the optimal number of coarse-grained TCs and fine-grained CNs in this dataset (see **R3C8**). For performance benchmarking, we fixed the number of fine-grained CNs to 12 (corresponding to the number of ground-truth CNs) and applied this strategy to determine the optimal number of coarse-grained TCs. The results showed that the optimal number of coarse-grained TCs is **two** (Response Fig. 35). Accordingly, we have updated the identified hierarchical structures under this configuration in the revised manuscript.

Response Fig. 35 (extracted from Supplementary Fig. 4). Cluster stability-based heuristic strategy to identify the optimal number of coarse-grained TCs and fine-grained CNs on the mouse hypothalamic preoptic region MERFISH dataset. (a) Cluster stability measure used by HRCHY-CytoCommunity to determine the optimal numbers of coarse-grained TCs and fine-grained CNs. Each point represents the Fowlkes-Mallows Index-based cluster stability for an individual sample. Horizontal bars indicate the mean across all samples. Mean values under each parameter setting are connected by grey solid lines. The red circle marks the selected stable resolution of tissue structures.

R3C3

3. For the brain example, it would be informative to overlay the results with reference annotations from the Allen Brain Atlas to validate spatial domains at both coarse and fine levels.

We thank the reviewer for this important suggestion. In the revised manuscript, we have validated the spatial domains identified by HRCHY-CytoCommunity against the reference anatomical annotations from the Allen Mouse Brain Atlas at both coarse- and fine-grained levels, using three representative brain datasets: the mouse hypothalamic preoptic region (MERFISH), the mouse hippocampus (Slide-seq V2), and the mouse ICH (Stereo-seq).

1. Coarse-grained validation:

In the Slide-seq V2 dataset, the identified coarse-grained TCs exhibited a strong correspondence with major anatomical regions in the Allen Mouse Brain Atlas, including the cortical region, hippocampus, and brain stem (Response Fig. 18c, middle panel). Similarly, in the Stereo-seq dataset, HRCHY-CytoCommunity consistently identified three biologically meaningful compartments: TC-0 (brain stem), TC-1 (cortex), and TC-2 (striatum), which closely matched the reference atlas (Response Fig. 21c, left column).

2. Fine-grained validation:

At the fine-grained level, HRCHY-CytoCommunity accurately resolved hippocampal

substructures in the Slide-seq V2 dataset, including the CA1-CA3 regions, dentate gyrus, corpus callosum, and cortical layers (Response Fig. 18d, middle panel). In the Stereo-seq dataset, it effectively delineated fine-grained CNs such as cortical layers, olfactory areas (OLF), meninges, and pallidum (PAL), all of which aligned well with the fine-level Allen Mouse Brain Atlas annotations (Response Fig. 21b, left column). Likewise, in the MERFISH dataset, the identified CNs corresponded to known hypothalamic nuclei (e.g., medial and lateral preoptic nuclei), displaying symmetric spatial organization consistent with the reference atlas (Response Fig. 26b, left column).

In summary, these results demonstrate that HRCHY-CytoCommunity not only achieves high quantitative accuracy but also recovers spatially and anatomically coherent domains that are highly concordant with the Allen Mouse Brain Atlas across multiple scales, thereby supporting the biological interpretability of the hierarchical tissue structures identified by HRCHY-CytoCommunity.

R3C4

4. In some examples, fine-grained domains appear more as transitional gradients than discrete clusters. This challenges the assumption of discrete, nested hierarchies and should be discussed.

We thank the reviewer for this insightful comment and fully agree that in certain biological contexts, fine-grained domains may exhibit continuous or transitional gradients rather than discrete cluster boundaries. In the revised manuscript, we have acknowledged this as a limitation of the current framework and clarified that HRCHY-CytoCommunity models tissue organization as a discrete hierarchy to enhance interpretability. Nevertheless, the use of soft probabilistic assignments allows the model to partially capture gradual transitions between adjacent regions (see R3C9, Response Fig. 31).

We also note that future extensions could incorporate trajectory-based representations to better characterize continuous spatial gradients. For example, recent approaches such as ONTraC²⁶ reconstruct spatially continuous niche trajectories and could complement discrete hierarchical modeling by providing a continuous perspective on tissue organization.

The corresponding statement has been added to the revised **Discussion** section:

“Moreover, while the use of discrete cell-type annotations enhances the method’s generalizability across platforms, it may not fully capture structures defined by continuous molecular gradients, transient states, or cell-cell communication signals. Future versions could incorporate multi-view learning to integrate such continuous features. In the meantime, methods like NicheCompass²⁰ (focused on cellular communication-driven structures) and ONTraC⁵⁵ (designed for spatially continuous niche trajectories) can complement HRCHY-CytoCommunity to provide a more comprehensive perspective on tissue organization.”

R3C5

5. While this method builds on Cytocommunity, the distinction lies primarily in the added CNN for hierarchical domain inference. Please clarify what conceptual or performance advancements are made beyond the original Cytocommunity approach.

We thank the reviewer for raising this important question. While HRCHY-CytoCommunity builds upon the conceptual foundation of CytoCommunity, it introduces **substantial conceptual, methodological, and performance advancements that enable hierarchical domain inference**, which is not supported by the original CytoCommunity framework. Specifically:

1. **Conceptual advancement:** CytoCommunity performs flat (non-hierarchical) tissue structure detection, identifying only a single-scale community structure. In contrast, HRCHY-CytoCommunity reformulates the problem as a **hierarchical community detection task**, jointly optimizing fine-grained CNs and coarse-grained TCs within a unified, fully differentiable framework. This design enables the model to capture multi-scale spatial organization that more faithfully reflects true tissue architecture.
2. **Methodological innovation:** HRCHY-CytoCommunity introduces a two-level differentiable pooling mechanism based on graph minimum-cut optimization, enabling learnable aggregation of fine-level CNs into coarse-level TCs. In addition, two new regularization strategies are incorporated:
 - a) **Consistency regularization**, which improves stability and reproducibility across stochastic perturbations.
 - b) **Entropy-based balance regularization**, which prevents cluster collapse and encourages balanced hierarchical assignments.

In summary, these conceptual and methodological advancements make HRCHY-CytoCommunity a **distinct framework** that advances the original CytoCommunity from flat clustering to hierarchical tissue structure identification.

R3C6

6. The claim that NeST is not applicable to low-feature datasets needs further explanation. Since NeST uses gene expression, it may still be applicable depending on the resolution. If excluded from comparisons, the manuscript should clearly justify why.

We thank the reviewer for this valuable comment and apologize for any confusion caused by our previous description of NeST's applicability to low-feature datasets. In the revised manuscript, we have clarified the description of NeST to more accurately reflect its scope and limitations:

“However, NeST relies on gene expression features, limiting its performance on data with sparse molecular measurements, such as spatial proteomics datasets”

Moreover, in the revised manuscript, we have expanded the performance benchmarking of HRCHY-CytoCommunity against NeST to include both imaging-based and sequencing-based datasets spanning a wide range of feature dimensionalities, as listed below:

1. Mouse spleen CODEX dataset (single-cell resolution, 30 proteins)
2. Human CRC Visium HD dataset (8 μm -bin resolution, 18085 genes)
3. Mouse hypothalamic preoptic region MERFISH dataset (single-cell resolution, 155 genes)
4. Mouse ICH Stereo-seq dataset (single-cell resolution, 23625 genes)
5. Mouse hippocampus Slide-seq V2 dataset (10 μm -spot resolution, 17733 genes)
6. Human breast cancer Visium V1 dataset (55 μm -spot resolution, 24923 genes)

Across all datasets, **HRCHY-CytoCommunity consistently achieved substantially superior performance compared to NeST**, both quantitatively (higher AMI and Macro-F1 scores) and qualitatively (more spatially coherent and biologically interpretable hierarchical structures) (Response Figs. 18-21, 24, 26).

Full details of performance benchmarking are provided in the **Results** section under titles: **“Performance evaluation on coarse-grained TC identification”**, **“Performance evaluation on fine-grained CN identification”**, and **“Cross-platform generalization of HRCHY-CytoCommunity to low-resolution spatial transcriptomics”**, as well as in **Supplementary Notes 1 and 2**.

R3C7

7. The use of manual annotations in unorganized organs (e.g., lung, kidney) is potentially problematic, as these organs lack strong spatial hierarchy. Please explain the rationale behind these choices and whether such annotations are used as training or just evaluation.

We thank the reviewer for this valuable comment. We fully agree that investigating hierarchical tissue structures in organs with unorganized or weak spatial hierarchies (e.g., lung or kidney) can be problematic. Accordingly, in this study, **HRCHY-CytoCommunity was applied only to organs with well-established multi-scale hierarchical organization** that have been consistently reported in previous studies, including **lymphoid organs**^{10,12} (e.g., spleen), **tumor tissues**^{15,17,30} (e.g., breast cancer and colorectal cancer), and the **brain** (e.g., hypothalamic preoptic region and hippocampus).

To avoid misunderstanding, we have **clarified the scope of our method** in the revised manuscript. The following revisions have been made in the revised **Discussion** section:

“As HRCHY-CytoCommunity explicitly models tissue organization as a hierarchy, it is particularly well-suited for organs with clearly stratified anatomical or functional layers. Its performance may be limited in tissues lacking strong hierarchical organization, such as liver, where metabolic zonation is continuously distributed along the porto-central axis⁵⁴.”

Reviewer #3 (Remarks on code availability):

The repository provided is well detailed and includes all information needed to reproduce the analysis presented and use the tool.

We thank the reviewer for the positive feedback and are glad to hear that the repository was found to be well documented and reproducible. The updated source code and documentation of HRCHY-CytoCommunity are available at <https://github.com/huBioinfo/HRCHY-CytoCommunity>.

Reviewer #4 (Remarks to the Author):

We thank the co-reviewer for their time and valuable contribution to the peer-review process.

References:

1. Varrone, M., Tavernari, D., Santamaria-Martínez, A., Walsh, L. A. & Ciriello, G. CellCharter reveals spatial cell niches associated with tissue remodeling and cell plasticity. *Nat Genet* **56**, 74–84 (2024).
2. Wang, Q. *et al.* The Allen Mouse Brain Common Coordinate Framework: A 3D Reference Atlas. *Cell* **181**, 936-953.e20 (2020).
3. Xu, H. *et al.* Unsupervised spatially embedded deep representation of spatial transcriptomics. *Genome Medicine* **16**, 12 (2024).
4. Feng, W. *et al.* Graph Random Neural Networks for Semi-Supervised Learning on Graphs.
5. Bai, Y. *et al.* SpaSEG: unsupervised deep learning for multi-task analysis of spatially resolved transcriptomics. *Genome Biol* **26**, 230 (2025).
6. Chen, J. G. *et al.* Giotto Suite: a multiscale and technology-agnostic spatial multiomics analysis ecosystem. *Nat Methods* 1–13 (2025) doi:10.1038/s41592-025-02817-w.
7. Long, Y. *et al.* Spatially informed clustering, integration, and deconvolution of spatial transcriptomics with GraphST. *Nat Commun* **14**, 1155 (2023).
8. Hu, J. *et al.* SpaGCN: Integrating gene expression, spatial location and histology to identify spatial domains and spatially variable genes by graph convolutional network. *Nat Methods* **18**, 1342–1351 (2021).
9. Birk, S. *et al.* Quantitative characterization of cell niches in spatially resolved omics data. *Nat Genet* <https://doi.org/10.1038/s41588-025-02120-6> (2025) doi:10.1038/s41588-025-02120-6.
10. Mebius, R. E. & Kraal, G. Structure and function of the spleen. *Nat Rev Immunol* **5**, 606–

616 (2005).

11. Cerutti, A., Cols, M. & Puga, I. Marginal zone B cells: virtues of innate-like antibody-producing lymphocytes. *Nat Rev Immunol* **13**, 118–132 (2013).
12. Goltsev, Y. *et al.* Deep Profiling of Mouse Splenic Architecture with CODEX Multiplexed Imaging. *Cell* **174**, 968-981.e15 (2018).
13. Lake, B. B. *et al.* An atlas of healthy and injured cell states and niches in the human kidney. *Nature* **619**, 585–594 (2023).
14. Fridman, W. H. *et al.* B cells and tertiary lymphoid structures as determinants of tumour immune contexture and clinical outcome. *Nat Rev Clin Oncol* **19**, 441–457 (2022).
15. Keren, L. *et al.* A Structured Tumor-Immune Microenvironment in Triple Negative Breast Cancer Revealed by Multiplexed Ion Beam Imaging. *Cell* **174**, 1373-1387.e19 (2018).
16. Ohara, K. *et al.* The evolution of metastatic upper tract urothelial carcinoma through genomic-transcriptomic and single-cell protein markers analysis. *Nat Commun* **15**, 2009 (2024).
17. Oliveira, M. F. D. *et al.* High-definition spatial transcriptomic profiling of immune cell populations in colorectal cancer. *Nat Genet* <https://doi.org/10.1038/s41588-025-02193-3> (2025) doi:10.1038/s41588-025-02193-3.
18. Arora, R. *et al.* Spatial transcriptomics reveals distinct and conserved tumor core and edge architectures that predict survival and targeted therapy response. *Nat Commun* **14**, 5029 (2023).
19. Yan, Y. *et al.* Multi-omic profiling highlights factors associated with resistance to immunotherapy in non-small-cell lung cancer. *Nat Genet* **57**, 126–139 (2025).

20. MacFawn, I. P. *et al.* The activity of tertiary lymphoid structures in high grade serous ovarian cancer is governed by site, stroma, and cellular interactions. *Cancer Cell* **42**, 1864-1881.e5 (2024).
21. Meylan, M. *et al.* Tertiary lymphoid structures generate and propagate anti-tumor antibody-producing plasma cells in renal cell cancer. *Immunity* **55**, 527-541.e5 (2022).
22. Tsutsumida, H. *et al.* A micropapillary pattern is predictive of a poor prognosis in lung adenocarcinoma, and reduced surfactant apoprotein A expression in the micropapillary pattern is an excellent indicator of a poor prognosis. *Mod Pathol* **20**, 638–647 (2007).
23. Sautès-Fridman, C., Petitprez, F., Calderaro, J. & Fridman, W. H. Tertiary lymphoid structures in the era of cancer immunotherapy. *Nat Rev Cancer* **19**, 307–325 (2019).
24. Schumacher, T. N. & Thommen, D. S. Tertiary lymphoid structures in cancer. *Science* <https://doi.org/10.1126/science.abf9419> (2022) doi:10.1126/science.abf9419.
25. Kietzmann, T. Metabolic zonation of the liver: The oxygen gradient revisited. *Redox Biol* **11**, 622–630 (2017).
26. Wang, W., Zheng, S., Shin, S. C., Chávez-Fuentes, J. C. & Yuan, G.-C. ONTraC characterizes spatially continuous variations of tissue microenvironment through niche trajectory analysis. *Genome Biology* **26**, 117 (2025).
27. Bhate, S. S., Barlow, G. L., Schürch, C. M. & Nolan, G. P. Tissue schematics map the specialization of immune tissue motifs and their appropriation by tumors. *Cell Syst* **13**, 109-130.e6 (2022).
28. Hickey, J. W. *et al.* Organization of the human intestine at single-cell resolution. *Nature* **619**, 572–584 (2023).

29. Nettekoven, C. *et al.* A hierarchical atlas of the human cerebellum for functional precision mapping. *Nat Commun* **15**, 8376 (2024).
30. Jackson, H. W. *et al.* The single-cell pathology landscape of breast cancer. *Nature* **578**, 615–620 (2020).

Response to reviewer comments

Reviewer #1 (Remarks to the Author):

Identifying multi-scale tissue structures from ST data is crucial for downstream biological discovery. This work provides a novel method to identify hierarchical structures.

This submission has made substantial revisions and addressed all my proposed questions. I recommend acceptance of this work.

Reviewer #2 (Remarks to the Author):

I am satisfied with the authors' response to my review comments. The revisions are detailed and directly address all major concerns raised in the initial review. Specifically, the authors have substantially strengthened the work by expanding the benchmarking, adding comprehensive ablation and robustness analyses, and most importantly replacing the post hoc majority-voting strategy with a principled, end-to-end regularization framework. These changes not only improve the technical rigor of the method but also enhance its transparency and practical relevance. I have no further concerns.

Reviewer #3 (Remarks to the Author):

The authors have substantially improved the manuscript in comparison to the first version. We are satisfied with the answers given to our concerns and with the modifications done in text and figures. We would like to highlight the level of detail provided in the answer's to reviewer's document, which better contextualize the benefits and limitations of this method in comparison to previous ones. We recommend the acceptance of the study.

Reviewer #4 (Remarks to the Author):

We sincerely thank all reviewers for the positive and encouraging comments regarding our revised manuscript. We are also grateful for their careful evaluation and constructive feedback throughout the entire review process, which have significantly strengthened our work.